# Machine Unlearning under Overparameterization

**Jacob L. Block**
UT Austin
jblock@utexas.edu

**Aryan Mokhtari**
UT Austin & Google Research
mokhtari@austin.utexas.edu

**Sanjay Shakkottai**
UT Austin
sanjay.shakkottai@utexas.edu

## Abstract

Machine unlearning algorithms aim to remove the influence of specific training samples, ideally recovering the model that would have resulted from training on the remaining data alone. We study unlearning in the overparameterized setting, where many models interpolate the data, and defining the solution as any loss minimizer over the retained set—as in prior work in the underparameterized setting—is inadequate, since the original model may already interpolate the retained data and satisfy this condition. In this regime, loss gradients vanish, rendering prior methods based on gradient perturbations ineffective, motivating both new unlearning definitions and algorithms. For this setting, we define the unlearning solution as the minimum-complexity interpolator over the retained data and propose a new algorithmic framework that only requires access to model gradients on the retained set at the original solution. We minimize a regularized objective over perturbations constrained to be orthogonal to these model gradients, a first-order relaxation of the interpolation condition. For different model classes, we provide exact and approximate unlearning guarantees and demonstrate that an implementation of our framework outperforms existing baselines across various unlearning experiments.

## 1 Introduction

The ability to remove the influence of specific samples from a trained model is essential to comply with privacy regulations such as the GDPR and CCPA [1, 2] and to correct mislabeling or bias that may compromise model integrity [3]. *Machine unlearning* [4] refers to algorithms that address these challenges by modifying a model trained on a dataset $\mathcal{D}$ to forget a subset of samples, termed the forget set $\mathcal{D}_f$, and produce a model that acts as if it had been trained only on the remaining data, denoted the retain set $\mathcal{D}_r = \mathcal{D} \setminus \mathcal{D}_f$. The ideal, yet costly, "gold standard" unlearning solution is to retrain the model from scratch on $\mathcal{D}_r$, which perfectly achieves the unlearning objective but is often infeasible due to high computational cost and potentially limited access to the original training data. The goal of an unlearning method is to efficiently approximate this outcome using the knowledge of the original training procedure, the samples to be forgotten, and potentially restricted side-information related to the retained data, aiming to recover a model that could result from training on $\mathcal{D}_r$ alone.

In the *underparameterized* regime, where the model class cannot fit all training data, the training loss admits a unique minimizer. Thus, the natural definition of exact unlearning is the unique minimizer to the loss on $\mathcal{D}_r$. When the loss is further strongly convex, prior work developed efficient unlearning approximations using influence functions, which estimate the effect of removing a sample via a single gradient ascent step over the loss on $\mathcal{D}_f$, preconditioned by the inverse loss Hessian on $\mathcal{D}$ [5–7].

39th Conference on Neural Information Processing Systems (NeurIPS 2025).

In contrast, this paper focuses on the *overparameterized* regime, where the model class contains many interpolating solutions. Crucially, the training loss no longer admits a unique minimizer, and defining the unlearning solution by loss optimality alone no longer suffices: the original model $\theta^*$ minimizes the loss over both $\mathcal{D}$ and $\mathcal{D}_r$, and $\theta^*$ clearly encodes information about $\mathcal{D}_f$, the data to be removed. Moreover, interpolation causes the loss gradients to vanish, rendering loss-gradient-based methods such as influence functions ineffective (Theorem 1). This fundamental shift necessitates both a new definition of unlearning and new algorithmic tools tailored to the overparameterized setting.

We begin by formalizing unlearning in the overparameterized setting. Specifically, we define the exact unlearning solution as the model which minimizes a model complexity measure $R$, subject to minimizing the loss over $\mathcal{D}_r$; see (2). For natural choices of $R$, such as the parameter norm, this definition ensures that the unlearned model reveals no information about the forgotten data and maintains strong generalization using only the retain set. Given this definition of unlearning, we propose a new algorithmic framework to compute the solution. We focus on settings where the loss is minimized by any interpolating model, so the constraint reduces to requiring interpolation over $\mathcal{D}_r$. To solve the resulting problem of minimizing $R$ subject to interpolation, we relax the constraint via a first-order Taylor expansion around $\theta^*$ and reparameterize as $\theta^* + \Delta$, where $\Delta$ is the drift. Since $\theta^*$ already interpolates $\mathcal{D}_r$, the linearized constraint requires $\Delta$ to be orthogonal to model gradients at $\theta^*$ on $\mathcal{D}_r$. This simplifies the problem, requiring only gradient access, and avoids the complex interpolation constraint. To mitigate error from this relaxation, we add a regularizer $\hat{R}(\Delta)$ to control the size and direction of the drift. The final objective minimizes $R(\theta^* + \Delta) + \hat{R}(\Delta)$ under the relaxed orthogonal gradient constraint, yielding updated parameters $\theta^* + \Delta$.

**Theoretical Contributions.** We prove for linear models and linear networks, there exists a regularizer $\hat{R}$ such that minimizing $R(\theta^* + \Delta) + \hat{R}(\Delta)$ over our constraint relaxation gives the exact unlearning solution when $R$ is the $\ell_2$-norm of either the effective regressor or the full parameter vector. For two-layer perceptrons with nonlinear activations, where $R$ measures network width, we prove that the right choice of $\hat{R}$ yields a solution to our relaxed problem which interpolates $\mathcal{D}_r$ and matches the best known upper bound on the number of neurons required to fit any dataset of a given size.

**Algorithmic Contributions.** We devise an iterative algorithm MinNorm-OG that accesses a subset of $\mathcal{D}_r$, aligning with data access assumptions in prior work [8–10], where OG refers to orthogonal gradient. MinNorm-OG alternates between two steps: solving for the minimizer of $R(\theta + \Delta) + \hat{R}(\Delta)$ over $\Delta$ satisfying the orthogonal gradient constraint, and descending on the loss over $\mathcal{D}_r$ (Algorithm 1). We take both $R$ and $\hat{R}$ as scaled squared $\ell_2$ norms, which apply broadly to parameterized models and yield a closed-form solution to the relaxed problem. We show strong performance of our method across three experimental settings: *Data Poisoning*, *Multi-Class Label Erasure*, and *Representation Collapse*, using natural and interpretable unlearning metrics to compare our method against existing baselines. Notably, the Multi-Class Label Erasure and Representation Collapse image-domain experiments introduce novel unlearning settings for effective evaluation.

**Related work**. Unlearning theory traces back to influence functions [11] which estimate the effect of down-weighting a sample on a learned function [5]. Extensions explored approximate unlearning via differential privacy [6, 7]. [6] analyzed the deletion capacity an unlearning method can tolerate while maintaining generalization. [12, 13] proposed joint learning-unlearning schemes that store information about data subsets during training for later unlearning. Several works proposed iterative unlearning methods for large-scale models, combining loss ascent, descent, and noise injection [14–18]. All these methods rely on loss gradient perturbations, which we show yield vacuous updates under overparameterization (Theorem 1). In practice, they also struggle to unlearn effectively [9], as loss ascent encourages misfitting $\mathcal{D}_f$ rather than forgetting it. Our framework enforces parameter perturbations to be orthogonal to the gradient over $\mathcal{D}_r$ to preserve loss optimality, an idea also used in continual learning contexts [19]. Recent unlearning methods use similar projections which mix loss ascent and descent, but their reliance on these objectives inherits prior limitations [20, 21].

## 2   Unlearning in Overparameterized Settings

We introduce notation (additional standard definitions in Appendix A) for our unlearning setting, highlighting the unique challenges of the overparameterized regime. We explain why loss optimality alone no longer suffices to define the ground truth unlearning solution and demonstrate why loss-gradient-based methods, originally designed for the underparameterized case, prove ineffective.

To formalize the unlearning problem, we now define the problem setting and notation, covering both the underparameterized and overparameterized regimes. We define the full training dataset $\mathcal{D} = \{(\boldsymbol{x}_i, \boldsymbol{y}_i)\}_{i=1}^{n}$, with sample inputs $\boldsymbol{x}_i \in \mathbb{R}^m$ and outputs $\boldsymbol{y}_i \in \mathbb{R}^l$ drawn from the data domain $\mathcal{Z} = \mathbb{R}^m \times \mathbb{R}^l$. Initially, training is performed on the full dataset $\mathcal{D}$ over the model class $\{f(\boldsymbol{\theta}, \cdot) \mid \boldsymbol{\theta} \in \mathbb{R}^d\}$ parameterized by $\boldsymbol{\theta} \in \mathbb{R}^d$, where $f : \mathbb{R}^{d+m} \to \mathbb{R}^l$ takes a parameter vector $\boldsymbol{\theta} \in \mathbb{R}^d$ and an input $\boldsymbol{x} \in \mathbb{R}^m$ and maps them to a prediction $f(\boldsymbol{\theta}, \boldsymbol{x})$ in the output space $\mathbb{R}^l$. We define the training procedure, also denoted the learning algorithm, as $\mathcal{A} : 2^{\mathcal{Z}} \to \mathbb{R}^d$, which takes in a dataset and returns the parameter vector $\boldsymbol{\theta}^*$ corresponding to the trained model. We make the minimal assumption that $\mathcal{A}$ is faithful to a known loss function $\mathcal{J}$, meaning $\mathcal{A}(\mathcal{D}) = \boldsymbol{\theta}^*$ is only guaranteed to be a minimizer of $\mathcal{J}$ over $\mathcal{D}$, where $\mathcal{J}$ is defined as the average of the sample-wise loss $\mathcal{L}$:

$$\mathcal{A}(\mathcal{D}) = \boldsymbol{\theta}^* \in \operatorname*{argmin}_{\boldsymbol{\theta}} \mathcal{J}(\boldsymbol{\theta}\,;\mathcal{D}) = \operatorname*{argmin}_{\boldsymbol{\theta}} \frac{1}{n} \sum_{(\boldsymbol{x},\boldsymbol{y}) \in \mathcal{D}} \mathcal{L}(\boldsymbol{\theta}\,;\boldsymbol{x},\boldsymbol{y})\,. \tag{1}$$

For our theoretical discussion, we consider sample-wise loss functions $\mathcal{L}(\boldsymbol{\theta}\,;\boldsymbol{x},\boldsymbol{y})$ which are minimized when $f(\boldsymbol{\theta},\boldsymbol{x}) = \boldsymbol{y}$, meaning that sample interpolation implies loss minimization. For example, this is the case for $\ell_p$-norm regression or classification with 0-1 loss.

With this training setup, we begin the unlearning process given a request for the model to forget a subset of the training data $\mathcal{D}_f \subseteq \mathcal{D}$. We then apply an unlearning algorithm $M(\mathcal{A}, \mathcal{I}_r, \mathcal{A}(\mathcal{D}), \mathcal{D}_f)$ which is given the learning algorithm $\mathcal{A}$, side information $\mathcal{I}_r$ (e.g., a subset of the samples, or the Hessian of the training loss over the retained data), initial solution $\mathcal{A}(\mathcal{D})$, and forget set $\mathcal{D}_f$, and which attempts to recover the desired unlearning solution, denoted by $\boldsymbol{\theta}_r^*$, where the subscript $r$ indicates that $\boldsymbol{\theta}_r^*$ is the parameter vector that would result from training only on the retain set $\mathcal{D}_r = \mathcal{D} \setminus \mathcal{D}_f$. To formally define $\boldsymbol{\theta}_r^*$, we must distinguish between underparameterized and overparameterized regimes, as the former's definition requires refinement to remain meaningful in the latter.

In the *underparameterized* setting, the loss function over both the full data set $\mathcal{J}(\boldsymbol{\theta}\,;\mathcal{D})$ as well as the retain set $\mathcal{J}(\boldsymbol{\theta}\,;\mathcal{D}_r)$ admits a unique minimizer. To ensure that the unlearning solution remains consistent with the training loss, the only valid choice is to define $\boldsymbol{\theta}_r^*$ as the unique minimizer of $\mathcal{J}(\boldsymbol{\theta}\,;\mathcal{D}_r)$. However, in the *overparameterized* setting this uniqueness property fails to hold, as both $\mathcal{J}(\boldsymbol{\theta}\,;\mathcal{D})$ and $\mathcal{J}(\boldsymbol{\theta}\,;\mathcal{D}_r)$ may admit multiple minimizers. In order to sidestep the non-uniqueness issue, one may be tempted to define *any* minimizer of $\mathcal{J}(\boldsymbol{\theta}\,;\mathcal{D}_r)$ as a valid unlearning solution, as presumably any minimizer to $\mathcal{J}(\boldsymbol{\theta}\,;\mathcal{D}_r)$ could be found from just training on $\mathcal{D}_r$ alone. However, following this rationale allows for seemingly valid unlearning solutions to leak information relating to $\mathcal{D}_f$. Specifically, the original solution $\boldsymbol{\theta}^*$ that interpolates all of $\mathcal{D}$ is itself a valid minimizer of the retain set loss $\mathcal{J}(\boldsymbol{\theta};\mathcal{D}_r)$, but $\boldsymbol{\theta}^*$ can reflect training dynamics influenced by $\mathcal{D}_f$, revealing information that cannot be inferred from $\mathcal{D}_r$ alone (see Appendix C for a concrete illustration).

## 2.1 Defining Unlearning Beyond Loss Optimality

As discussed above, the overparameterized setting requires a more fine-grained definition of the desired unlearning solution—one that goes beyond loss optimality. We define the unlearning solution in the overparameterized case to be the specific loss minimizer which minimizes an additional objective function $R(\boldsymbol{\theta})$, expressed as the output of a training algorithm $\mathcal{A}_R$:

$$\mathcal{A}_R(\mathcal{D}_r) = \boldsymbol{\theta}_r^* \in \operatorname*{argmin}_{\boldsymbol{\theta}} R(\boldsymbol{\theta}), \qquad \text{subject to} \quad \boldsymbol{\theta} \in \operatorname*{argmin}_{\boldsymbol{\theta}'} \mathcal{J}(\boldsymbol{\theta}';\mathcal{D}_r)\,. \tag{2}$$

This bilevel optimization problem searches for the model which minimizes the complexity measure $R$ among all models which minimize the retain set loss. Indeed, when $R$ admits a unique solution, this formulation overcomes the prior issues of non-uniqueness and the risk of revealing information from the forget set. While different choices of $R$ can address these issues, we ultimately want $R$ to promote desirable model properties. In our theoretical results, we focus on $R$ as a regularization function that penalizes model complexity. This way, the solution $\boldsymbol{\theta}_r^*$ to (2) corresponds to the simplest model that interpolates $\mathcal{D}_r$ – a particularly useful property in the overparameterized regime, where the simplest interpolating model is often associated with optimal generalization performance [22].

Then given the training algorithm $\mathcal{A}_R$, side information about the retain set $\mathcal{I}_r$, a minimizer to the original training loss $\mathcal{A}(\mathcal{D})$, and the forget set $\mathcal{D}_f$, an unlearning algorithm $M(\mathcal{A}_R, \mathcal{I}_r, \mathcal{A}(\mathcal{D}), \mathcal{D}_f)$ attempts to recover $\mathcal{A}_R(\mathcal{D}_r)$, the least complex loss minimizer over $\mathcal{D}_r$ as measured by $R$.

## 2.2 Loss Gradient Methods Deployed Under Overparameterization

For the characterization in (2) of the ground truth unlearning solution under overparameterization, we show that existing unlearning methods based on loss gradient perturbations fail to achieve meaningful unlearning updates. Prior theoretical works proposed gradient-ascent-style updates based on influence functions, a principled technique from robust statistics [5–7], while existing empirical unlearning methods perform combinations of loss ascent over $\mathcal{D}_f$, loss descent over $\mathcal{D}_r$, and parameter noising [14–16, 8]. We characterize these methods as *loss-gradient unlearning*, and show that they perform ineffective updates when deployed under overparameterization.

**Definition 1.** *Let $\boldsymbol{\theta}^* = \mathcal{A}(\mathcal{D})$. We say an unlearning algorithm $M$ performs loss-gradient unlearning if for any positive semi-definite $\boldsymbol{P}_r, \boldsymbol{P}_f \in \mathbb{R}^{d \times d}$ and zero-mean random variable $\boldsymbol{\xi} \in \mathbb{R}^d$,*

$$M\left(\mathcal{A}, \mathcal{I}_r, \mathcal{A}(\mathcal{D}), \mathcal{D}_f\right) = \boldsymbol{\theta}^* - \boldsymbol{P}_r \nabla_{\boldsymbol{\theta}} \mathcal{J}\left(\boldsymbol{\theta}^*; \mathcal{D}_r\right) + \boldsymbol{P}_f \nabla_{\boldsymbol{\theta}} \mathcal{J}\left(\boldsymbol{\theta}^*; \mathcal{D}_f\right) + \boldsymbol{\xi} \tag{3}$$

Although versions of loss-gradient unlearning have been theoretically motivated in the underparameterized setting [6, 7], we show they fail to unlearn in the overparameterized setting.

**Theorem 1.** *Let $f(\boldsymbol{\theta}^*, \cdot)$ interpolate $\mathcal{D}$, so $f(\boldsymbol{\theta}^*, \boldsymbol{x}) = \boldsymbol{y}$ for all $(\boldsymbol{x}, \boldsymbol{y}) \in \mathcal{D}$, and let $M_{LG}$ be any loss-gradient unlearning method. If the sample loss $\mathcal{L}(\boldsymbol{\theta}, \boldsymbol{x}, \boldsymbol{y})$ is minimized when $f(\boldsymbol{\theta}, \boldsymbol{x}) = \boldsymbol{y}$, then for all $\mathcal{D}_f \subseteq \mathcal{D}$, $M_{LG}$ simply noises $\boldsymbol{\theta}^*$ by some zero-mean random variable $\boldsymbol{\xi}$.*

$$M_{LG}\left(\mathcal{A}, \mathcal{I}_r, \mathcal{A}(\mathcal{D}), \mathcal{D}_f\right) = \boldsymbol{\theta}^* + \boldsymbol{\xi}$$

The recovered parameters $\boldsymbol{\theta}^*$ already minimize $\mathcal{J}\left(\boldsymbol{\theta}^*; \mathcal{D}_r\right)$, so the loss gradients vanish and $M_{LG}$ merely adds noise to $\boldsymbol{\theta}^*$. This shows the core issue with loss gradient updates in overparameterized unlearning: the loss gradient is uninformative, as both $\boldsymbol{\theta}^*$ and $\boldsymbol{\theta}_r^*$ minimize the loss on $\mathcal{D}_r$.

## 3 Our Proposed Framework

We present a new framework to efficiently address the desired unlearning goal in overparameterized settings without full retraining. A key assumption underlying our method is the richness of the function class, allowing for perfect fitting of the retain set. This means there exist several mappings $f(\boldsymbol{\theta}, \cdot)$ where $f(\boldsymbol{\theta}, \boldsymbol{x}_i) = \boldsymbol{y}_i$ for every $(\boldsymbol{x}_i, \boldsymbol{y}_i)$ in the retain set. This lets us replace the loss minimization in (2) with the hard constraint $f(\boldsymbol{\theta}, \boldsymbol{x}_i) = \boldsymbol{y}_i$, leading to the following formulation:

$$\boldsymbol{\theta}_r^* \in \underset{\boldsymbol{\theta}}{\arg\min}\, R(\boldsymbol{\theta}) \qquad \text{s.t. } f(\boldsymbol{\theta}, \boldsymbol{x}_i) = \boldsymbol{y}_i \quad \forall (\boldsymbol{x}_i, \boldsymbol{y}_i) \in \mathcal{D}_r \tag{4}$$

This problem can be independently solved, but this would be the equivalent of retraining on the retain set. The main goal of our proposed framework is to solve the above problem efficiently by starting from the model $\boldsymbol{\theta}^*$ which fits each sample and leveraging the feasibility of this model for the above optimization problem. To do so, we simplify the problem and replace the constraints in (4) with their linear approximation around $\boldsymbol{\theta}^*$. While the constraints $f(\boldsymbol{\theta}, \boldsymbol{x}_i) = \boldsymbol{y}_i$ in (4) can be highly nonconvex and difficult to satisfy in general, we demonstrate that using the proposed first-order approximation

$$f(\boldsymbol{\theta}^*, \boldsymbol{x}_i) + \nabla f(\boldsymbol{\theta}^*, \boldsymbol{x}_i)^\top (\boldsymbol{\theta} - \boldsymbol{\theta}^*) = \boldsymbol{y}_i \quad \Rightarrow \quad \nabla f(\boldsymbol{\theta}^*, \boldsymbol{x}_i)^\top (\boldsymbol{\theta} - \boldsymbol{\theta}^*) = 0, \tag{5}$$

renders it tractable as it leads to a set of linear constraints with respect to $\boldsymbol{\theta}$. Note that in the above simplification we used the fact that $\boldsymbol{\theta}^*$ perfectly fits the retain set, so $f(\boldsymbol{\theta}^*, \boldsymbol{x}_i) = \boldsymbol{y}_i$. Now if we apply this constraint relaxation the resulting optimization problem would be:

$$\min_{\boldsymbol{\Delta}} R(\boldsymbol{\theta}^* + \boldsymbol{\Delta}) \qquad \text{s.t.} \quad \nabla f(\boldsymbol{\theta}^*, \boldsymbol{x}_i)^\top \boldsymbol{\Delta} = 0 \quad \forall (\boldsymbol{x}_i, \boldsymbol{y}_i) \in \mathcal{D}_r, \tag{6}$$

where for notational convenience, we define the drift variable as $\boldsymbol{\Delta} = \boldsymbol{\theta} - \boldsymbol{\theta}^*$. While this relaxation is sensible, it presents a clear limitation: approximating a general function with its linearization is only locally accurate and thus valid when the drift term $\boldsymbol{\Delta}$ remains sufficiently small in some norm. To keep the surrogate solution close to that of the original problem in (4), we add a regularization term $\hat{R}(\boldsymbol{\Delta})$ to the loss to control the drift. The resulting objective function is $\tilde{R}(\boldsymbol{\theta}^* + \boldsymbol{\Delta}) := R(\boldsymbol{\theta}^* + \boldsymbol{\Delta}) + \hat{R}(\boldsymbol{\Delta})$. Consequently, the optimization problem we propose to solve instead of (4) is given by

$$\tilde{\boldsymbol{\Delta}} \in \underset{\boldsymbol{\Delta}}{\arg\min}\, \tilde{R}(\boldsymbol{\theta}^* + \boldsymbol{\Delta}) \qquad \text{s.t.} \quad \nabla f(\boldsymbol{\theta}^*, \boldsymbol{x}_i)^\top \boldsymbol{\Delta} = 0 \quad \forall (\boldsymbol{x}_i, \boldsymbol{y}_i) \in \mathcal{D}_r \tag{7}$$

Indeed, by finding $\tilde{\boldsymbol{\Delta}}$ the suggested unlearned model would be $\boldsymbol{\theta}^* + \tilde{\boldsymbol{\Delta}}$. Although (7) employs relaxed constraints, we will show that for various mapping functions $f$, there exists a function $\hat{R}$ such that the solution to (7) either (i) solves the original unlearning problem (4) exactly, or (ii) yields a model that both interpolates $\mathcal{D}_r$, remaining feasible for (4), and satisfies a tight upper bound on the complexity measure $R$. A key advantage of the formulation in (7), beyond simplifying the constraints, is its minimal information requirement: it only relies on the gradient of $f$ evaluated at the original trained model, i.e., the side information $\mathcal{I}_r = \{\nabla_{\boldsymbol{\theta}} f(\boldsymbol{\theta}^*, \boldsymbol{x})\}_{(\boldsymbol{x},\boldsymbol{y}) \in \mathcal{D}_r}$. This is significantly less restrictive than prior work, which requires access to the inverse Hessian of the loss over $\mathcal{D}_r$ [5–7], and makes our method substantially simpler than full retraining.

# 4 Theoretical Guarantees

This section provides theoretical guarantees for using our proposed relaxation (7) to solve the exact unlearning problem (4). Going forward, we denote Euclidean projection onto the set $\boldsymbol{S}$ as $\mathcal{P}_{\boldsymbol{S}}(\cdot)$ and define the penalty function $\delta_{\{a\}}$ which is $+\infty$ if condition $a$ is satisfied and $0$ otherwise.

## 4.1 Linear Model

Given a linear model $\boldsymbol{\theta}^*$ with $\boldsymbol{\theta}^{*\top} \boldsymbol{x}_i = y_i$ for all $(\boldsymbol{x}_i, y_i) \in \mathcal{D}$, we can easily solve the exact unlearning problem (4) for $R(\boldsymbol{\theta}) = \|\boldsymbol{\theta}\|_2$.

**Theorem 2.** *Let $\tilde{\boldsymbol{\Delta}}$ solve* (7) *for $f(\boldsymbol{\theta}, \boldsymbol{x}) = \boldsymbol{\theta}^\top \boldsymbol{x}$ and $\tilde{R}(\boldsymbol{\theta}) = \|\boldsymbol{\theta}\|_2$. Then the recovered solution $\tilde{\boldsymbol{\theta}} = \boldsymbol{\theta}^* + \tilde{\boldsymbol{\Delta}}$ solves the exact unlearning problem* (4) *for $R(\boldsymbol{\theta}) = \|\boldsymbol{\theta}\|_2$*

This result holds because, in the linear case, the surrogate and original constraints match exactly, and no approximation error is introduced. Thus, no additional regularizer (i.e., $\hat{R}(\cdot) = 0$) is needed.

## 4.2 $L$-Layer Linear Network

In this section, we extend our analysis to a more complex model: an $L$-layer linear network. Let the prediction function be $f(\boldsymbol{\theta}, \boldsymbol{x}) = \boldsymbol{c}^\top \boldsymbol{A}_{L-1} \cdots \boldsymbol{A}_1 \boldsymbol{x}$, where the parameter vector is partitioned $\boldsymbol{\theta} = [\boldsymbol{c} \,; \mathrm{vec}(\boldsymbol{A}_1) \,; \ldots \,; \mathrm{vec}(\boldsymbol{A}_{L-1})]$, with $\boldsymbol{A}_\ell \in \mathbb{R}^{h_\ell \times h_{\ell-1}}$ and $\boldsymbol{c} \in \mathbb{R}^{h_{L-1}}$ for $\ell = 1, \ldots, L-1$. The input dimension is $m = h_0$, and we assume $n < m$ to reflect the overparameterized regime. For clarity, define the effective linear predictor $\mathbf{w}(\boldsymbol{\theta}) = \boldsymbol{A}_1^\top \cdots \boldsymbol{A}_{L-1}^\top \boldsymbol{c}$, so that $f(\boldsymbol{\theta}, \boldsymbol{x}) = \mathbf{w}(\boldsymbol{\theta})^\top \boldsymbol{x}$. For this model class, we study two natural choices of regularizers in (4): (i) $R$ as the norm of the prediction function as a linear map, and (ii) $R$ as the norm of all model parameters.

### 4.2.1 Minimizing Predictor Norm

We first analyze when the $R$ measures the $\ell_2$-norm of the effective linear predictor: $R(\boldsymbol{\theta}) = \left\| \boldsymbol{A}_1^\top \cdots \boldsymbol{A}_{L-1}^\top \boldsymbol{c} \right\|_2 = \|\mathbf{w}(\boldsymbol{\theta})\|_2$. Given $\boldsymbol{\theta}^* = \left[ \boldsymbol{c}^* \,; \mathrm{vec}(\boldsymbol{A}_1^*) \,; \ldots \,; \mathrm{vec}(\boldsymbol{A}_{L-1}^*) \right]$ such that $\mathbf{w}(\boldsymbol{\theta}^*)^\top \boldsymbol{x} = y$ for all $(\boldsymbol{x}, y) \in \mathcal{D}$, we aim to solve (4) for this choice of $R$. In this case the mapping $f$ is non-linear with respect to $\boldsymbol{\theta}$. As a result, the first-order approximation for the constraints is not tight, so solving the surrogate problem in (7) does not necessarily give a solution for the problem in (4). However, we show that adding a suitable regularizer $\hat{R}$ to control model drift ensures the relaxed and original problems have the same solution. We first present an intermediate result showing the existence of a feasible perturbation $\tilde{\boldsymbol{\Delta}}$ that satisfies the relaxed linearized constraints and, when added to $\boldsymbol{\theta}^*$, yields an optimal solution to (4).

**Lemma 1.** *Denote the retain set input subspace by $\boldsymbol{S}_r = \mathrm{span}\{\boldsymbol{x} \mid (\boldsymbol{x}, y) \in \mathcal{D}_r\}$ and partition the perturbation as $\tilde{\boldsymbol{\Delta}} = \left[ \tilde{\boldsymbol{\Delta}}_{\boldsymbol{c}} \,; \mathrm{vec}(\tilde{\boldsymbol{\Delta}}_{\boldsymbol{A}_1}) \,; \ldots \,; \mathrm{vec}(\tilde{\boldsymbol{\Delta}}_{\boldsymbol{A}_{L-1}}) \right]$ in the same manner as $\boldsymbol{\theta}$. Set*

$$\tilde{\boldsymbol{\Delta}}_{\boldsymbol{A}_1} = - \left\| \boldsymbol{A}_2^{*\top} \cdots \boldsymbol{A}_{L-1}^{*\top} \boldsymbol{c}^* \right\|_2^{-2} \boldsymbol{A}_2^{*\top} \cdots \boldsymbol{A}_{L-1}^{*\top} \boldsymbol{c}^* \mathcal{P}_{\boldsymbol{S}_r^\perp} \left( \mathbf{w}(\boldsymbol{\theta}^*) \right)^\top \tag{8}$$

*and all other components of $\tilde{\boldsymbol{\Delta}}$ to zero. Then $\tilde{\boldsymbol{\Delta}}$ is orthogonal to the gradient of mapping $f(\boldsymbol{\theta}, \boldsymbol{x})$ evaluated at $\boldsymbol{\theta} = \boldsymbol{\theta}^*$ for each input $\boldsymbol{x}$ in the retain set and hence feasible for the relaxed problem* (7). *Moreover, $\boldsymbol{\theta}^* + \tilde{\boldsymbol{\Delta}}$ solves the exact unlearning problem* (4) *for $R(\boldsymbol{\theta}) = \|\mathbf{w}(\boldsymbol{\theta})\|_2$.*

The above result shows that the perturbation direction defined in (8) leads to an optimal solution for (4) once added to $\boldsymbol{\theta}^*$, while satisfying the relaxed linear constraints of the surrogate problem. That

said, it does not imply that solving (6), which only differs in the constraints from (4), would recover $\tilde{\boldsymbol{\Delta}}$. In fact, we can show that without adding a proper regularization term $\hat{R}$ to the loss, $\tilde{\boldsymbol{\Delta}}$ would not be a solution of the relaxed problem (see Appendix D.4.1). We next characterize the appropriate regularization $\hat{R}(\boldsymbol{\Delta})$ needed to ensure that $\tilde{\boldsymbol{\Delta}}$ is the optimal solution to the surrogate problem in (7).

**Theorem 3.** *The solution to the relaxed unlearning problem* (7) *with the following choice of* $\tilde{R}$ *solves the exact unlearning problem* (4) *for* $R(\boldsymbol{\theta}) = \|\mathbf{w}(\boldsymbol{\theta})\|_2$.

$$\tilde{R}(\boldsymbol{\theta} \,; \boldsymbol{\theta}^*) = \|\mathbf{w}(\boldsymbol{\theta})\|_2 + \delta_{\{\boldsymbol{c} \neq \boldsymbol{c}^*\}} + \sum_{\ell=2}^{L-1} \delta_{\{\boldsymbol{A}_\ell \neq \boldsymbol{A}_\ell^*\}} \tag{9}$$

### 4.2.2 Minimizing Parameter Norm

Next, we analyze when the unlearning solution is the loss minimizer with the smallest parameter norm, so $R(\boldsymbol{\theta}) = \|\boldsymbol{\theta}\|_2$. In this case, we can construct an exact unlearning solution from the exact unlearning solution to the previously analyzed case when $R(\boldsymbol{\theta}) = \|\mathbf{w}(\boldsymbol{\theta})\|_2$.

**Theorem 4.** *Let* $\hat{\boldsymbol{\theta}}_r^*$ *solve* (4) *for* $R(\boldsymbol{\theta}) = \|\mathbf{w}(\boldsymbol{\theta})\|_2$, *so* $\mathbf{w}(\hat{\boldsymbol{\theta}}_r^*)$ *is the min $\ell_2$-norm linear predictor over* $\mathcal{D}_r$. *Define* $\rho = \|\mathbf{w}(\hat{\boldsymbol{\theta}}_r^*)\|_2$ *and let* $\boldsymbol{v}_\ell \in \mathbb{R}^{h_\ell}$ *for* $\ell \in [L-1]$ *each satisfy* $\|\boldsymbol{v}_\ell\|_2 = 1$. *Set*

$$\tilde{\boldsymbol{A}}_1 = \rho^{\frac{1-L}{L}} \boldsymbol{v}_1 \mathbf{w}(\hat{\boldsymbol{\theta}}_r^*)^\top, \quad \tilde{\boldsymbol{A}}_\ell = \rho^{\frac{1}{L}} \boldsymbol{v}_\ell \boldsymbol{v}_{\ell-1}^\top \text{ for } \ell = 2, \ldots, L-1, \quad \tilde{\boldsymbol{c}} = \rho^{\frac{1}{L}} \boldsymbol{v}_{L-1}.$$

*Then* $\tilde{\boldsymbol{\theta}} = \left[\tilde{\boldsymbol{c}} \,; \mathrm{vec}(\tilde{\boldsymbol{A}}_1); \ldots; \mathrm{vec}(\tilde{\boldsymbol{A}}_{L-1})\right]$ *solves the exact unlearning problem* (4) *for* $R(\boldsymbol{\theta}) = \|\boldsymbol{\theta}\|_2$.

Thus, the solution to the minimum norm predictor problem gives the solution to minimum parameter norm problem, so we can apply the previous results to find a solution for (4) with $R(\boldsymbol{\theta}) = \|\mathbf{w}(\boldsymbol{\theta})\|_2$ using the constraint relaxation and then update the parameters as prescribed by Theorem 4.

### 4.3 2-Layer Perceptron

We lastly consider a 2-layer perceptron with a non-linear activation. Specifically, we define $f(\boldsymbol{\theta}, \boldsymbol{x}) = \boldsymbol{c}^\top \phi(\boldsymbol{A}\boldsymbol{x})$, where we use the partition $\boldsymbol{\theta} = [\boldsymbol{c} \,; \mathrm{vec}(\boldsymbol{A})]$ for $\boldsymbol{c} \in \mathbb{R}^h$, $\boldsymbol{A} \in \mathbb{R}^{h \times m}$. Here, $h$ is the total number of neurons and $\phi : \mathbb{R} \to \mathbb{R}$ is some activation function. We abuse notation and write $\phi(\boldsymbol{A}\boldsymbol{x})$ to denote the element-wise application of $\phi$ to $\boldsymbol{A}\boldsymbol{x}$. We analyze the case where $R$ measures the number of active neurons, i.e., the width of the network. Formally, we denote $\boldsymbol{a}_i^\top$ as the $i$th row of $\boldsymbol{A}$, and we set $R(\boldsymbol{\theta}) = \sum_{i=1}^h \mathbb{1}\{|c_i| \|\boldsymbol{a}_i\|_2 > 0\}$. With this choice of $R$, the unlearning solution promotes recovering a sparse network which fits $\mathcal{D}_r$, where $\mathcal{D}_r$ has $n_r = |\mathcal{D}_r|$ samples. Given that $\boldsymbol{c}^{*\top} \phi(\boldsymbol{A}^* \boldsymbol{x}) = y$ for all $(\boldsymbol{x}, y) \in \mathcal{D}$, we chase the minimum neuron interpolating solution to $\mathcal{D}_r$:

$$\boldsymbol{\theta}_r^* \in \operatorname*{argmin}_{\boldsymbol{\theta}} R(\boldsymbol{\theta}) \quad \text{s.t. } \boldsymbol{c}^\top \phi(\boldsymbol{A}\boldsymbol{x}) = y \quad \forall (\boldsymbol{x}, y) \in \mathcal{D}_r \tag{10}$$

While we aim to solve (10) for any retain set $\mathcal{D}_r$, the exact minimal-width solution remains unknown. Prior work shows that $n_r + 1$ neurons suffice for general activations [23], while for ReLU, some $n_r$-sample datasets need at least $n_r - 2$ neurons [24]. Here, we apply our framework to recover feasible unlearned networks with width at most $n_r$, improving the best known worst-case bound.

We begin by linearizing the constraints of problem (10) around $\boldsymbol{\theta}^*$, as directly solving this problem may be intractable due to the non-linear activation $\phi$, especially since we assume access to only the model gradients over $\mathcal{D}_r$, not the samples in $\mathcal{D}_r$ themselves. We define the drift as $\boldsymbol{\Delta} = [\boldsymbol{\Delta}_c \,; \mathrm{vec}(\boldsymbol{\Delta}_A)]$, yielding the specific instance of the linearized problem (6) for this model class:

$$\min_{\boldsymbol{\Delta}} R(\boldsymbol{\theta}^* + \boldsymbol{\Delta}) \text{ s.t. } \boldsymbol{\Delta}_c^\top \phi(\boldsymbol{A}^* \boldsymbol{x}_i) + \mathrm{tr}\big\{\boldsymbol{\Delta}_A^\top \left(\phi'(\boldsymbol{A}^* \boldsymbol{x}_i) \odot \boldsymbol{c}^*\right) \boldsymbol{x}_i^\top \big\} = 0 \ \ \forall (\boldsymbol{x}_i, \boldsymbol{y}_i) \in \mathcal{D}_r, \tag{11}$$

where $\odot$ denotes element-wise product. Due to the layered structure and non-linear activation $\phi$, solving (11) may not ensure feasibility for (10), as the relaxed constraints are loose. We first show that a feasible perturbation $\tilde{\boldsymbol{\Delta}}$, modifying only the last layer $\boldsymbol{c}^*$, exists and yields a network satisfying (10) with at most $n_r$ active neurons.

**Lemma 2.** *Assume the finite-width network* $f(\boldsymbol{\theta}^*, \boldsymbol{x}) = \boldsymbol{c}^{*\top} \phi(\boldsymbol{A}^* \boldsymbol{x})$ *interpolates* $\mathcal{D}_r$, *where* $n_r = |\mathcal{D}_r|$ *is the number of retain set samples. Let* $\dim\left(\mathrm{span}\{\phi(\boldsymbol{A}^* \boldsymbol{x})\}_{(\boldsymbol{x},y) \in \mathcal{D}_r}\right) = s \leq n_r$. *Then, there exists a feasible perturbation* $\tilde{\boldsymbol{\Delta}}$ *satisfying the linear constraints in* (11), *such that* $f(\boldsymbol{\theta}^* + \tilde{\boldsymbol{\Delta}}, \cdot)$ *interpolates* $\mathcal{D}_r$, $R(\boldsymbol{\theta}^* + \tilde{\boldsymbol{\Delta}}) \leq s$, *and* $\tilde{\boldsymbol{\Delta}}_A = \boldsymbol{0}$.

---

**Algorithm 1** MinNorm-OG

---

1: **Input:** $\boldsymbol{\theta}^*$, loss $\mathcal{J}(\boldsymbol{\theta})$, $\mathcal{D}'_r \subseteq \mathcal{D}_r$, step size $\eta_t$, regularization constant $\lambda_t \geq 0$, subsample batch size $n_{\text{pert}}$
2: Initialize $\boldsymbol{\theta} \leftarrow \boldsymbol{\theta}^*$
3: **for** $t = 1, \ldots, n_{\text{epochs}}$ **do**
4:     **for** each batch $\mathcal{B}$ from $\mathcal{D}'_r$ **do**
5:         **if** $\lambda_t < \infty$ **then**
6:             Compute function gradients $\boldsymbol{g}_i = \nabla_{\boldsymbol{\theta}} f(\boldsymbol{\theta}, \boldsymbol{x}_i)$ for $x_i \in \mathcal{B}$, $i = 1, \ldots, n_{\text{pert}}$
7:             Solve $\tilde{\boldsymbol{\Delta}} \leftarrow \operatorname{argmin}_{\boldsymbol{\Delta}} \|\boldsymbol{\theta} + \boldsymbol{\Delta}\|_2^2 + \lambda_t \|\boldsymbol{\Delta}\|_2^2$   s.t.   $\boldsymbol{\Delta} \perp \boldsymbol{g}_i$ for all $i \leq n_{\text{pert}}$
8:             Update $\boldsymbol{\theta} \leftarrow \boldsymbol{\theta} + \tilde{\boldsymbol{\Delta}}$
9:         Loss descent: $\boldsymbol{\theta} \leftarrow \boldsymbol{\theta} - \eta_t \nabla_{\boldsymbol{\theta}} \mathcal{J}(\boldsymbol{\theta}; \mathcal{B})$
10: **return** $\boldsymbol{\theta}$

---

While Lemma 2 provides a feasible point for (11), it is not the solution, as the relaxed problem linearizes the interpolation constraint without limiting drift size, potentially losing interpolation over $\mathcal{D}_r$. The following theorem shows that choosing $\hat{R}$ to restrict perturbations in $\boldsymbol{A}^*$ ensures that solving (7) yields a network feasible for (10) with at most $n_r$ active neurons.

**Theorem 5.** *For $R(\boldsymbol{\theta})$ which measures the number of active neurons of the network $f(\boldsymbol{\theta}, \cdot)$, define*

$$\tilde{R}(\boldsymbol{\theta}; \boldsymbol{\theta}^*) = R(\boldsymbol{\theta}) + \delta_{\{\boldsymbol{A} \neq \boldsymbol{A}^*\}} \tag{12}$$

*as the surrogate objective. Then the solution to the relaxed unlearning problem (7) with this choice of $\tilde{R}$ results in a network which interpolates $\mathcal{D}_r$, achieving feasibility for the exact unlearning problem (10), and admits at most $s = \dim\left(\operatorname{span}\{\phi(\boldsymbol{A}^*\boldsymbol{x})\}_{(\boldsymbol{x},y) \in \mathcal{D}_r}\right) \leq n_r$ active neurons, where $n_r = |\mathcal{D}_r|$.*

Theorem 5 shows that for general activation functions, linearizing the constraints to (10) and minimizing the sum of the complexity measure $R$ along the appropriate regularizer $\hat{R}$ for the drift term recovers a network that interpolates $\mathcal{D}_r$ with at most $s$ active neurons, where $s$ is the dimension of the span of the learned representations $\{\phi(\boldsymbol{A}^*\boldsymbol{x})\}_{(\boldsymbol{x},y) \in \mathcal{D}_r}$. Since $s$ can never exceed $n_r = |\mathcal{D}_r|$ our method guarantees a worst-case interpolation width of at most $n_r$, thereby improving the general bound of $n_r + 1$ implied by [23] for minimum width interpolation.

The drift regularizer $\hat{R}$ only allows perturbations to $\boldsymbol{c}^*$, so the solution to (7) reduces width via sparsity in the updated last layer $\boldsymbol{c}^* + \tilde{\boldsymbol{\Delta}}_{\boldsymbol{c}}$, while leaving the first layer $\boldsymbol{A}^*$ unchanged. Although $\boldsymbol{c}^* + \tilde{\boldsymbol{\Delta}}_{\boldsymbol{c}}$ relies on a small set of features, the feature map $\phi(\boldsymbol{A}^*\boldsymbol{x})$ still reflects representations learned from all of $\mathcal{D}$. We show, however, that the sparsity of $\boldsymbol{c}^* + \tilde{\boldsymbol{\Delta}}_{\boldsymbol{c}}$ can be propagated into $\boldsymbol{A}^*$, producing a network with a new, sparser feature map that is less expressive and no longer consistent with having been trained on the full dataset $\mathcal{D}$, yet still satisfies all unlearning guarantees in Theorem 5.

**Proposition 1.** *Let $\boldsymbol{\theta} = [\boldsymbol{c}; \operatorname{vec}(\boldsymbol{A})]$ be any parameter vector, and define $\hat{\boldsymbol{A}} = (\mathbf{1}_{\boldsymbol{c} \neq 0}, \mathbf{1}^\top) \odot \boldsymbol{A}$. Then the updated parameters $\hat{\boldsymbol{\theta}} = [\boldsymbol{c}; \operatorname{vec}(\hat{\boldsymbol{A}})]$ satisfy: (i) $f(\boldsymbol{\theta}, \boldsymbol{x}) = f(\hat{\boldsymbol{\theta}}, \boldsymbol{x})$ for all $\boldsymbol{x} \in \mathbb{R}^m$, (ii) $R(\boldsymbol{\theta}) = R(\hat{\boldsymbol{\theta}})$, and (iii) $\hat{\boldsymbol{A}}$ has at most $R(\hat{\boldsymbol{\theta}})$ number of nonzero rows.*

Thus, for any parameters $\boldsymbol{\theta}$, we can apply a simple update to recover new parameters $\hat{\boldsymbol{\theta}}$ which behave like an $R(\boldsymbol{\theta})$-neuron network in terms of both the function outputs and at the parameter level. We apply this result to the solution to the relaxed unlearning problem (7) in the following corollary.

**Corollary 1.** *Let $\tilde{\boldsymbol{\theta}} = [\tilde{\boldsymbol{c}}; \operatorname{vec}(\boldsymbol{A}^*)]$ solve (7) for $\tilde{R}$ defined in (12), and define the updated first layer as $\hat{\boldsymbol{A}} = (\mathbf{1}_{\tilde{\boldsymbol{c}} \neq 0} \mathbf{1}^\top) \odot \boldsymbol{A}^*$. Then network parameterized by $\hat{\boldsymbol{\theta}} = [\tilde{\boldsymbol{c}}; \operatorname{vec}(\hat{\boldsymbol{A}})]$ similarly interpolates $\mathcal{D}_r$, has the same number of active neurons $R(\tilde{\boldsymbol{\theta}}) = R(\hat{\boldsymbol{\theta}})$, and $\hat{\boldsymbol{A}}$ has at most $R(\hat{\boldsymbol{\theta}})$ non-zero rows.*

Thus, solving the relaxed problem (7) and updating $\boldsymbol{A}^*$ via Proposition 1 yields a network that reveals no trace of having been trained on the larger dataset $\mathcal{D} = \mathcal{D}_r \sqcup \mathcal{D}_f$, even at the representation level.

## 5 From Theory to Practice

We translate our framework into a practical algorithm MinNorm-OG (Algorithm 1). At epoch $t$, we alternate between solving a version of the relaxed unlearning problem (7) and descending the loss

Table 1: Data Poisoning experiment results, measured as the median sup-norm distance between the retain set trend $y = \sin(x)$ and the unlearned model outputs over 10 trials (smaller is better).

| Epochs | Retrain | MinNorm-OG | GD | GA | NGP | NGD | Ridge | $\ell_1$-Sparse |
|---|---|---|---|---|---|---|---|---|
| 10 | **1.50** | **1.50** | 3.23 | 2.56 | 2.50 | 2.73 | 2.27 | 3.28 |
| 100 | 1.36 | **1.08** | 2.76 | 23.8 | 2.62 | 2.85 | 2.13 | 1.79 |
| 1000 | 1.17 | **0.63** | 2.61 | 1400 | 2.75 | 2.45 | 1.96 | 1.37 |

on $\mathcal{D}_r$ to maintain feasibility for the exact unlearning problem (4), leveraging access to samples in $\mathcal{D}_r$. Steps 6-8 of Algorithm 1 denote solving (7) for $R(\boldsymbol{\theta}) = \|\boldsymbol{\theta}\|_2^2$ and $\hat{R}(\boldsymbol{\Delta}) = \lambda_t \|\boldsymbol{\Delta}\|_2^2$ where $\lambda_t \geq 0$ is a scaling parameter, and step 9 shows the loss descent step. To handle batched data and large models, we enforce the orthogonality constraint in (7) over a subsample of size $n_{\text{pert}}$ within each batch. For this $R$ and $\hat{R}$, the solution to (7) perturbs $\boldsymbol{\theta}$ toward its projection onto the span of model gradients over this subsample (see Appendix E), which can be interpreted as a proximal update under the orthogonal gradient constraint. The main overhead relative to gradient descent comes from solving for $\tilde{\boldsymbol{\Delta}}$ via a QR decomposition with complexity $\mathcal{O}(dn_{\text{pert}}^2)$, which is negligible compared to the $\mathcal{O}(dn_B)$ cost of gradient descent when $n_{\text{pert}} < \sqrt{n_B}$, where $n_B = |\mathcal{B}|$ is the batch size.

## 5.1 Experiments

We test our algorithm against the following baseline methods. Retrain erases the initial model and trains from scratch on $\mathcal{D}_r$. GD [14] runs gradient descent on $\mathcal{J}(\boldsymbol{\theta}; \mathcal{D}_r)$, while Noisy GD (NGD) [16] adds gradient noise to the GD steps. GA [15] runs gradient ascent on $\mathcal{J}(\boldsymbol{\theta}; \mathcal{D}_f)$. NegGrad+ (NGP) [8] minimizes a weighted combination of the GD and GA objectives. $\ell_1$-Sparse [25] runs GD with an $\ell_1$-norm regularizer. SCRUB [8] optimizes three objectives: minimizing $\mathcal{J}(\boldsymbol{\theta}; \mathcal{D}_r)$, minimizing KL divergence of model outputs on $\mathcal{D}_r$ relative to the original model, and maximizing KL divergence on $\mathcal{D}_f$. Negative Preference Optimization (NPO) [18] runs a form of gradient ascent over $\mathcal{J}(\boldsymbol{\theta}; \mathcal{D}_f)$ inspired by preference optimization. SalUn [26] minimizes $\mathcal{J}(\boldsymbol{\theta}; \mathcal{D}_r)$ while fitting random labels over $\mathcal{D}_f$, updating only the parameters that initially have large gradients with respect to $\mathcal{J}(\boldsymbol{\theta}; \mathcal{D}_f)$. We apply SCRUB, NPO, and SalUn only in our classification-based experiments, as they are not directly designed for general regression tasks. To highlight the performance of our algorithm, we also compare to ridge regression, which approximates our unlearning objective (4) for $R(\boldsymbol{\theta}) = \|\boldsymbol{\theta}\|_2$ by minimizing $\mathcal{J}(\boldsymbol{\theta}; \mathcal{D}_r) + \lambda_t \|\boldsymbol{\theta}\|_2^2$. The minimizer of this regularized objective converges to the minimum-$\ell_2$-norm loss minimizer in the ridgeless limit as $\lambda_t \to 0$ [22].

While recent work proposed various unlearning benchmarks, especially for LLMs [16, 27–29], they often rely on opaque metrics that emphasize suppressing forget-set generation. In contrast, we present the following experiments with interpretable quantitative metrics. In each experiment, we evaluate all methods over a fixed number of unlearning epochs. In the Data Poisoning experiment, each epoch corresponds to a single unlearning update over the full dataset, $\mathcal{D} = \mathcal{D}_f \sqcup \mathcal{D}_r$. In contrast, for the larger-scale Multi-Class Label Erasure and Representation Collapse experiments, each method processes batches from the entire forget set $\mathcal{D}_f$ and only a subset of the retain set, $\mathcal{D}'_r \subset \mathcal{D}_r$. Here, one epoch is defined as a full pass over the forget set batches, where each forget batch is paired with a retain batch of equal size sampled from $\mathcal{D}'_r$. Complete details of our experimental setup and full results with uncertainty estimates are provided in Appendix F.

**Data Poisoning.** We train a shallow neural network on retain samples $(x_r, y_r) \in \mathcal{D}_r$ with $y_r = \sin(x_r)$ and forget samples $(x_f, y_f) \in \mathcal{D}_f$ with $y_f = 1.5$, over input domain $\mathcal{X} = [-5\pi, 5\pi] \subseteq \mathbb{R}$. We evaluate the output $\boldsymbol{\theta}$ of each unlearning method by measuring the deviation from the retain set trend, given by $\sup_{\boldsymbol{x} \in \mathcal{X}} |f(\boldsymbol{\theta}, \boldsymbol{x}) - \sin(\boldsymbol{x})|$. Results are reported in Table 1 as the median over 10 trials with visualizations in Figure 1. Retrain fails to closely capture the retain set trend $y = \sin(x)$, while the (regularized) loss descent methods (GD, NGD, Ridge) struggle to simultaneously escape the influence of the forget set and fit to the retain set. GA is unstable and diverges from the retain set. Due to this instability, we found that NGP performed best when the loss-ascent coefficient was near zero, causing it to behave like GD.

**Multi-Class Label Erasure.** We use the CIFAR-10 [30] and Tiny ImageNet [31] datasets, creating red, green, and gray copies of each image. We train modified ResNet-18 and ResNet-50 models on

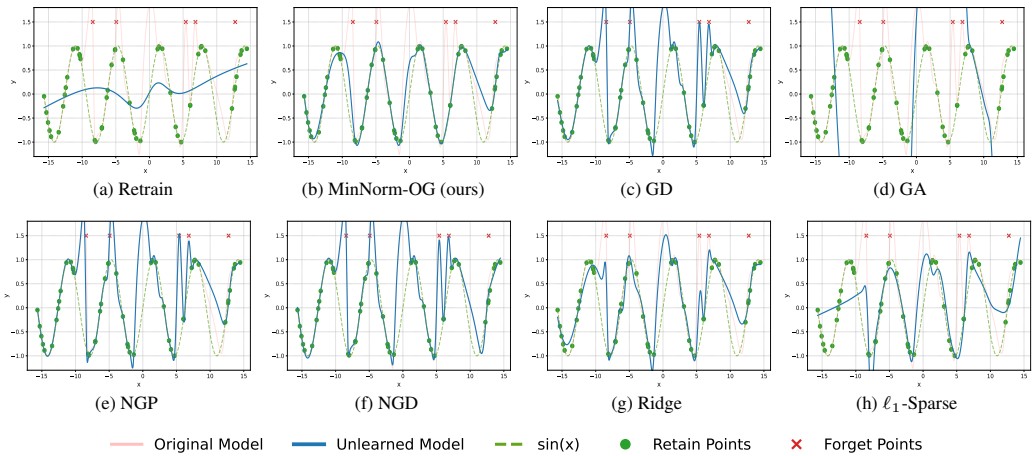

Figure 1: Example unlearned model fits when given 100 unlearning epochs for the Data Poisoning experiment, where the forget points distort the retain set trend $y = \sin(x)$.

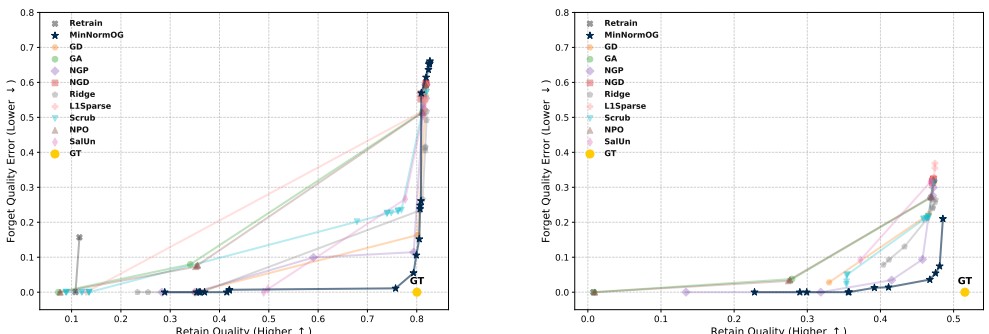

Figure 2: Pareto frontiers for each method across hyperparameter settings in the Multi-Class Label Erasure experiment on colored versions of CIFAR-10 (left) and Tiny ImageNet (right). Models predict color and content, but the retain set contains only gray images. The ground truth unlearned model (GT) performs well on gray inputs but always predicts gray with probability 1. The x-axis shows accuracy on gray test images (higher is better), and the y-axis shows mean squared error between predicted probability of gray on all inputs and the target of 1 (lower is better). MinNorm-OG (ours) best approaches the ground-truth unlearned model's performance relative to the other baselines.

CIFAR-10 and Tiny ImageNet, respectively, to jointly predict image class and color. The retain set $\mathcal{D}_r$ contains all image content classes only in gray, while the forget set $\mathcal{D}_f$ contains all colors. The ground truth unlearned model predicts gray content well and always predicts gray color with probability 1, regardless of input. We evaluate retain quality by accuracy on gray-colored test samples, and we measure forget quality error as the mean squared error between the model's predicted probability of an image being gray and the target value of 1 for all colored inputs.

Figure 2 presents the Pareto frontier of the mean performance across 5 trials for each method under different hyperparameter settings. The optimal point $(1, 0)$ indicates perfect retain quality and zero forget quality error. The ground truth unlearned model is labeled GT. Our method, MinNorm-OG, performs best across both datasets, as the other methods struggle to unlearn the influence of the colored forget set samples without harming accurate classification of gray test images. Notably, retraining from scratch yields a model with nearly perfect forget quality error, as it is trained only on gray images, but performs no better than random guessing on gray test images due to the limited training epochs and restricted retain set access.

**Representation Collapse.** We again use the CIFAR-10 dataset where each of the 10 classes is assigned a unique color. The retain set $\mathcal{D}_r$ contains the CIFAR-10 images colored according to their unique color, while the forget set $\mathcal{D}_f$ comprises randomly colored images. The ground truth

Table 2: Representation Collapse experiment results across constraints on the number of unlearning epochs and percentage of accessible retain set samples. Models are trained on colored images where color perfectly predicts the label in the retain set but not in the full dataset $\mathcal{D}$. Evaluation is measured as the mean accuracy (%) on test images labeled by color over 5 trials (higher is better).

| Retain % | Epochs | Retrain | MinNorm-OG | GD | GA | NGP | NGD | Ridge | $\ell_1$-Sparse | Scrub | NPO | SalUn |
|---|---|---|---|---|---|---|---|---|---|---|---|---|
| | 5 | 79.1 | **49.1** | 24.0 | 34.5 | 45.9 | 12.6 | 20.9 | 12.4 | 37.3 | 40.6 | 29.4 |
| 0.1 | 10 | 95.5 | **78.7** | 55.2 | 38.3 | 73.7 | 33.0 | 55.2 | 23.0 | 61.9 | 43.2 | 53.4 |
| | 15 | 96.3 | **93.2** | 78.2 | 39.8 | 81.5 | 46.9 | 78.2 | 25.1 | 74.1 | 44.6 | 76.3 |
| 1 | 5 | 92.8 | **59.5** | 40.7 | 34.2 | 58.3 | 32.9 | 33.5 | 12.5 | 47.8 | 42.5 | 42.5 |
| | 10 | 98.5 | **94.7** | 73.6 | 38.2 | 92.4 | 70.5 | 63.4 | 31.4 | 75.8 | 43.5 | 68.7 |

Table 3: Average time in seconds to perform a single unlearning epoch on the Multi-Class Label Erasure experiment on Tiny ImageNet using ResNet-50, averaged over 5 trials.

| Method | Retrain | MinNorm-OG | GD | GA | NGP | NGD | Ridge | $\ell_1$-Sparse | Scrub | NPO | SalUn |
|---|---|---|---|---|---|---|---|---|---|---|---|
| Time (s) | 0.90 | 0.78 | 0.67 | 0.68 | 0.74 | 0.69 | 0.68 | 0.69 | 0.96 | 0.94 | 1.09 |

unlearned model predicts from color alone, as the color feature completely determines the class label and is easier to learn than the image content features. In contrast, models trained on the full dataset $\mathcal{D} = \mathcal{D}_r \sqcup \mathcal{D}_f$ must predict based on content, since color is no longer fully predictive of the label. For evaluation, we label heldout test images by color and assess unlearning via color-label accuracy, testing if the unlearning methods can collapse the original model into just a color classifier.

Table 2 presents mean results over 5 trials. We observe that Retrain achieves much higher color classification accuracy than any unlearning method, as image color is an easy feature to learn from scratch. However, this result is specific to this setup: since the target function over the retain set is so simple, it is easier to learn from a freshly initialized model than by adapting the original model, which has learned to rely on complex image content features. Our prior experiments demonstrate that Retrain is not a viable unlearning strategy in general, so we report its results separately in the leftmost column. We bold the MinNorm-OG results, as they achieve the highest accuracy for each combination of constraints on the percentage of accessible retain set samples and number of unlearning epochs.

## 5.2 Runtime Comparison

In the above experiments, we compare all unlearning methods under a fixed number of epochs. To assess per-epoch computational efficiency, we measure the runtime required by each method to complete a single unlearning epoch in the Multi-Class Label Erasure experiment using Tiny ImageNet with ResNet-50, our largest-scale setting. The results are reported in Table 3.

We observe that methods which compute loss gradients with simple regularizers on a single data split (GD, GA, NGD, Ridge, $\ell_1$-Sparse) are the fastest. NGP is slower, as it requires forward and backward passes over both retain and forget set batches. Our method, MinNorm-OG, operates only on the retain set but solves an additional subproblem on top of the loss descent step, while maintaining a runtime comparable to NGP. Scrub and NPO are slower, as they require extra forward passes using the original model to evaluate their loss functions, while Retrain is slowed by the need to initialize a new model from scratch. Lastly, SalUn is the slowest due to an initial pass over the entire dataset to compute the threshold for deciding which parameters to freeze during unlearning.

## 6 Conclusion

We proposed a new unlearning framework under overparameterization by seeking the simplest solution consistent with the retain set. We proved guarantees on solving the exact unlearning problem through a tractable relaxed formulation. A practical implementation of our framework outperformed baselines, as the simplest solution aligns with unlearning goals and removes artifacts unrelated to the retain set. While our theoretical guarantees open the door for unlearning analysis beyond the underparameterized setting, we focused on model classes like linear networks and two-layer perceptrons. We naturally aim to analyze unlearning in more complex settings like deep networks in future work, as well as experiment within broader domains at larger scale.

## Acknowledgments

This work was supported in part by NSF Grants 2019844, 2107037, and 2112471, ONR Grant N00014-19-1-2566, the Machine Learning Lab (MLL) at UT Austin, the NSF AI Institute for Foundations of Machine Learning (IFML), and the Wireless Networking and Communications Group (WNCG) Industrial Affiliates Program. We are grateful for computing support on the Vista GPU Cluster through the Center for Generative AI (CGAI) and the Texas Advanced Computing Center (TACC) at the University of Texas at Austin.

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

# A  General Notation

Vectors and matrices are in bold, with vectors lowercase and matrices uppercase. For sets $A, B$, $A \sqcup B$ denotes disjoint union. $2^A$ is the power set. For a proposition $a$, $\mathbb{1}\{a\}$ is 1 if true and 0 otherwise; $\delta_{\{a\}}$ is $+\infty$ if true and 0 otherwise. For $\boldsymbol{x} \in \mathbb{R}^d$ and $A \subseteq \mathbb{R}^d$, $\mathcal{P}_A(\boldsymbol{x})$ is the Euclidean projection onto $A$. For $\boldsymbol{Z} \in \mathbb{R}^{m \times n}$, $\mathrm{vec}(\boldsymbol{Z}) \in \mathbb{R}^{mn}$ is the columnwise vectorization. $\mathrm{im}(\boldsymbol{Z})$, $\ker(\boldsymbol{Z})$, and $\mathrm{row}(\boldsymbol{Z})$ denote the image, kernel, and rowspace. $\|\boldsymbol{Z}\|_F$ is the Frobenius norm, $\|\boldsymbol{Z}\|_*$ is the nuclear norm, and $\|\boldsymbol{Z}\|_2$ is the spectral norm. For $Y \in \mathbb{R}^{m \times n}$, $\langle \boldsymbol{Z}, \boldsymbol{Y} \rangle$ is the Frobenius inner product and $\boldsymbol{Z} \odot \boldsymbol{Y}$ is the element-wise product. $\mathrm{tr}\{\cdot\}$ is the trace. For $\boldsymbol{x} \in \mathbb{R}^{d_x}$ and $\boldsymbol{y} \in \mathbb{R}^{d_y}$, $[\boldsymbol{x}; \boldsymbol{y}] \in \mathbb{R}^{d_x + d_y}$ stacks $\boldsymbol{x}$ and $\boldsymbol{y}$. $\|\boldsymbol{x}\|_p$ is the $\ell_p$ norm. $[n] = \{1, \ldots, n\}$. For $x \in \mathbb{R}$, $(x)_+ = \max\{x, 0\}$ is the ReLU. Let $\boldsymbol{0}$ and $\boldsymbol{1}$ denote the vectors with each entry equal to 0 and 1 respectively. Further, for $\boldsymbol{x} \in \mathbb{R}^d$ and $c \in \mathbb{R}$, let $\boldsymbol{1}_{\boldsymbol{x} \neq c}$ denote the vector which is 1 in each entry of $\boldsymbol{x}$ which is not equal to $c$ and 0 otherwise.

# B  Minimum Norm Solutions to Linear Regression

Here we prove various properties of minimum norm solutions to linear regression problems which we later use for our unlearning results. Following the notation in Section 2, we consider the full $n$-sample dataset $\mathcal{D} = \{(\boldsymbol{x}_i, y_i)\}_{i=1}^n$ with sample inputs $\boldsymbol{x}_i \in \mathbb{R}^m$ and outputs $y_i \in \mathbb{R}$. We consider training a linear model $f(\boldsymbol{\theta}, \boldsymbol{x}) = \boldsymbol{\theta}^\top \boldsymbol{x}$ parameterized by $\boldsymbol{\theta} \in \mathbb{R}^m$. We work within the overparameterized setting, so we assume $m > n$. Define the span of the input vectors $\boldsymbol{S} = \mathrm{span}\{\boldsymbol{x} \mid (\boldsymbol{x}, y) \in \mathcal{D}\}$, and assume $\dim(\boldsymbol{S}) = n$ so the regression problem is realizable. Consider solving the following problem for finding the linear regression solution with minimum $\ell_2$ norm:

$$\boldsymbol{\theta}^* = \operatorname*{argmin}_{\boldsymbol{\theta}} \|\boldsymbol{\theta}\|_2 \ \text{ s.t. } f(\boldsymbol{\theta}, \boldsymbol{x}) = y \quad \forall (\boldsymbol{x}, y) \in \mathcal{D}$$

Let $\boldsymbol{X} \in \mathbb{R}^{n \times m}$ be the wide matrix whose $i$th row is equal to $\boldsymbol{x}_i^\top$, and let $\boldsymbol{y} \in \mathbb{R}^n$ be the vector whose $i$th element is $y_i$. Then, we can write an equivalent problem in matrix form.

$$\boldsymbol{\theta}^* = \operatorname*{argmin}_{\boldsymbol{\theta}} \frac{1}{2} \|\boldsymbol{\theta}\|_2^2 \ \text{ s.t. } \boldsymbol{y} = \boldsymbol{X}\boldsymbol{\theta} \tag{13}$$

We can then characterize the solution to the above problem relative to the constraint set.

**Lemma 3.** $\boldsymbol{\theta}^*$ *is the unique vector in* $\mathrm{row}(\boldsymbol{X})$ *which is feasible for* (13)

*Proof.* The objective (13) is a convex objective with linear constraints which is bounded from below by 0 and has a non-empty feasible set. Thus, the KKT conditions are necessary and sufficient for optimality. We now derive the solution $\boldsymbol{\lambda}^* \in \mathbb{R}^n$ to the dual problem.

$$
\begin{aligned}
\min_{\boldsymbol{\theta}} \frac{1}{2} \|\boldsymbol{\theta}\|_2^2 \ \text{ s.t. } \ \boldsymbol{y} = \boldsymbol{X}\boldsymbol{\theta} \quad &= \min_{\boldsymbol{\theta}} \max_{\boldsymbol{\lambda} \in \mathbb{R}^n} \frac{1}{2} \|\boldsymbol{\theta}\|_2^2 + \boldsymbol{\lambda}^\top (\boldsymbol{y} - \boldsymbol{X}\boldsymbol{\theta}) \\
&= \max_{\boldsymbol{\lambda}} \min_{\boldsymbol{\theta}} \frac{1}{2} \|\boldsymbol{\theta}\|_2^2 + \boldsymbol{\lambda}^\top (\boldsymbol{y} - \boldsymbol{X}\boldsymbol{\theta}) \\
&= \max_{\boldsymbol{\lambda}} \frac{1}{2} \|\boldsymbol{X}^\top \boldsymbol{\lambda}\|_2^2 + \boldsymbol{\lambda}^\top (\boldsymbol{y} - \boldsymbol{X}\boldsymbol{X}^\top \boldsymbol{\lambda}) \quad \text{ s.t. } \ \boldsymbol{\theta} = \boldsymbol{X}^\top \boldsymbol{\lambda} \\
&= \max_{\boldsymbol{\lambda}} -\frac{1}{2} \|\boldsymbol{X}^\top \boldsymbol{\lambda}\|_2^2 + \boldsymbol{\lambda}^\top \boldsymbol{y} \quad \text{ s.t. } \ \boldsymbol{\theta} = \boldsymbol{X}^\top \boldsymbol{\lambda} \\
&\implies \boldsymbol{X}\boldsymbol{X}^\top \boldsymbol{\lambda}^* = \boldsymbol{y} \text{ and } \boldsymbol{\theta}^* = \boldsymbol{X}^\top \boldsymbol{\lambda}^* \tag{14}
\end{aligned}
$$

Thus the primal solution $\boldsymbol{\theta}^*$ must be of the form $\boldsymbol{X}^\top \boldsymbol{\lambda}^* \in \mathrm{row}(\boldsymbol{X})$. To show uniqueness, consider $\boldsymbol{\theta}_1^*, \boldsymbol{\theta}_2^* \in \mathrm{row}(\boldsymbol{X})$ that are both feasible for (13). Then,

$$\boldsymbol{y} = \boldsymbol{X}\boldsymbol{\theta}_1^* = \boldsymbol{X}\boldsymbol{\theta}_2^* \implies \boldsymbol{X}(\boldsymbol{\theta}_1^* - \boldsymbol{\theta}_2^*) = \boldsymbol{0} \implies \boldsymbol{\theta}_1^* - \boldsymbol{\theta}_2^* \in \ker(\boldsymbol{X}).$$

But, since $\mathrm{row}(\boldsymbol{X})$ is a subspace, $\boldsymbol{\theta}_1^*, \boldsymbol{\theta}_2^* \in \mathrm{row}(\boldsymbol{X})$ implies $\boldsymbol{\theta}_1^* - \boldsymbol{\theta}_2^* \in \mathrm{row}(\boldsymbol{X})$. Further, $\mathrm{row}(\boldsymbol{X}) = \ker(\boldsymbol{X})^\perp$. Thus,

$$\boldsymbol{\theta}_1^* - \boldsymbol{\theta}_2^* \in \ker(\boldsymbol{X}) \cap \ker(\boldsymbol{X})^\perp = \{\boldsymbol{0}\} \implies \boldsymbol{\theta}_1^* = \boldsymbol{\theta}_2^*$$

$\square$

Using the same analysis, we can characterize the entire feasible set in terms of $\boldsymbol{\theta}^*$.

**Lemma 4.** *The feasible set to* (13) $\{\boldsymbol{\theta} \mid \boldsymbol{y} = \boldsymbol{X}\boldsymbol{\theta}\} = \boldsymbol{\theta}^* + \ker(\boldsymbol{X})$.

*Proof.* Let $\boldsymbol{\theta}'$ satisfy $\boldsymbol{y} = \boldsymbol{X}\boldsymbol{\theta}'$. Then, $\boldsymbol{X}(\boldsymbol{\theta}' - \boldsymbol{\theta}^*) = \boldsymbol{0}$ so $\boldsymbol{\theta}' - \boldsymbol{\theta} \in \ker(\boldsymbol{X})$.

To show the converse, take any $\boldsymbol{z} \in \ker(\boldsymbol{X})$. Then $\boldsymbol{X}(\boldsymbol{\theta}^* + \boldsymbol{z}) = \boldsymbol{X}\boldsymbol{\theta}^* + \boldsymbol{X}\boldsymbol{z} = \boldsymbol{X}\boldsymbol{\theta}^* = \boldsymbol{y}$. □

Using this characterization of $\boldsymbol{\theta}^*$ and the feasible set, we can cleanly understand how to achieve minimum norm solutions over just a subset of the constraints given a feasible point. This is central to our unlearning setup in later sections.

**Lemma 5.** *Consider any subset $\mathcal{D}_r \subseteq \mathcal{D}$, and define $\boldsymbol{\theta}_r^*$ as the linear regression solution over just $\mathcal{D}_r$ with minimum norm:*

$$\boldsymbol{\theta}_r^* = \operatorname*{argmin}_{\boldsymbol{\theta}} \|\boldsymbol{\theta}\|_2 \ \text{s.t.} \ f(\boldsymbol{\theta}, \boldsymbol{x}) = y \quad \forall (\boldsymbol{x}, y) \in \mathcal{D}_r \tag{15}$$

*Let $\boldsymbol{S}_r = \operatorname{span}\{\boldsymbol{x} \mid (\boldsymbol{x}, y) \in \mathcal{D}_r\}$. Then $\boldsymbol{\theta}_r^* = \mathcal{P}_{\boldsymbol{S}_r}(\boldsymbol{\theta}^*)$.*

*Proof.* $\boldsymbol{\theta}^*$ already satisfies the feasibility constraint over the whole dataset $\mathcal{D}$, so it must be feasible for (15). Applying Lemmas 3 and 4 to the minimum norm problem over just $\mathcal{D}_r$ (15), we must have that $\boldsymbol{\theta}_r^* \in \boldsymbol{S}_r$ and $\boldsymbol{\theta}^* = \boldsymbol{\theta}_r^* + \boldsymbol{z}$ for some $\boldsymbol{z} \in \boldsymbol{S}_r^\perp$. Then,

$$\mathcal{P}_{\boldsymbol{S}_r}(\boldsymbol{\theta}^*) = \mathcal{P}_{\boldsymbol{S}_r}(\boldsymbol{\theta}_r^* + \boldsymbol{z}) = \mathcal{P}_{\boldsymbol{S}_r}(\boldsymbol{\theta}_r^*) = \boldsymbol{\theta}_r^*.$$

□

## C  Loss Minimization Does not Protect Against Data Leakage

The following example concretely demonstrates how certain minimizers of the retain set loss do not align with the intended goals of unlearning.

Recall the unlearning problem for linear regression discussed in Section 4.1. In this case, we use the linear model $f(\boldsymbol{\theta}, \boldsymbol{x}) = \boldsymbol{\theta}^\top \boldsymbol{x}$ parameterized by $\boldsymbol{\theta} \in \mathbb{R}^m$. Further suppose the original dataset $\mathcal{D} = \{(\boldsymbol{x}_i, y_i)\}_{i=1}^n$ has $n$ samples with $\boldsymbol{x}_i \in \mathbb{R}^m, y_i \in \mathbb{R}$. Denote the subspace $\boldsymbol{S} = \operatorname{span}\{\boldsymbol{x} \mid (\boldsymbol{x}, y) \in \mathcal{D}\}$, and assume $\dim(\boldsymbol{S}) = n$ so the problem is realizable. We work in the overparameterized setting where $m > n$ and the objective function is defined as the mean squared error denoted by

$$\mathcal{J}(\boldsymbol{\theta}; \mathcal{D}) = \frac{1}{n} \sum_{(\boldsymbol{x}, y) \in \mathcal{D}} \left(y - \boldsymbol{\theta}^\top \boldsymbol{x}\right)^2$$

Consider when the learning algorithm $\mathcal{A}$ runs gradient descent on the loss, initialized at $\boldsymbol{0}$. Due to the overparameterization, $\mathcal{J}$ has an infinite number of minimizers which each achieve $0$ loss. However, $\mathcal{A}$ is biased towards a specific minimizer which is the unique minimizer to the loss on the span of the input samples, denoted as the subspace $\boldsymbol{S}$.

**Proposition 2.** *Let $\mathcal{A}^k(\mathcal{D})$ be a learning algorithm which runs $k$ steps of gradient descent on $\mathcal{J}(\boldsymbol{\theta}; \mathcal{D})$ initialized at $\boldsymbol{0}$, and define $\boldsymbol{S} = \operatorname{span}\{\boldsymbol{x} \mid (\boldsymbol{x}, y) \in \mathcal{D}\}$. If $\lim_{k \to \infty} \mathcal{A}^k(\mathcal{D})$ converges to some $\boldsymbol{\theta}^*$, then*

$$\{\boldsymbol{\theta}^*\} = \boldsymbol{S} \cap \operatorname{argmin} \mathcal{J}(\boldsymbol{\theta}; \mathcal{D})$$

*Proof.* We write the loss function $\mathcal{J}(\boldsymbol{\theta}; \mathcal{D})$ in vector form $\mathcal{J}(\boldsymbol{\theta}; \mathcal{D}) = \frac{1}{n}\|\boldsymbol{y} - \boldsymbol{X}\boldsymbol{\theta}\|_2^2$, where the $i$th entry of $\boldsymbol{y} \in \mathbb{R}^n$ is $y_i$ and the $i$th row of $\boldsymbol{X} \in \mathbb{R}^{n \times m}$ is $\boldsymbol{x}_i^\top$. Note that the gradient of the loss for any value of $\boldsymbol{\theta}$ is contained the subspace $\boldsymbol{S}$, as $\nabla_{\boldsymbol{\theta}} \mathcal{J}(\boldsymbol{\theta}; \mathcal{D}) = \frac{2}{n}\boldsymbol{X}^\top(\boldsymbol{X}\boldsymbol{\theta} - \boldsymbol{y})$ and $\operatorname{im}(\boldsymbol{X}^\top) = \boldsymbol{S}$. Further, the initial iterate of $\mathcal{A}^k$ is $\boldsymbol{0} \in \boldsymbol{S}$. Since subspaces are closed under addition, every iterate of gradient descent on $\mathcal{J}(\boldsymbol{\theta}; \mathcal{D})$ starting from $\boldsymbol{0}$ must be contained in $\boldsymbol{S}$. Thus if $\mathcal{A}^k(\mathcal{D})$ converges, it must converge to a zero of the gradient of the loss, and this point must also be in $\boldsymbol{S}$. Since the loss is convex, this point must be a loss minimizer. □

In this case, the original training solution $\boldsymbol{\theta}^*$ which results from simply performing gradient descent interpolates all of $\mathcal{D}$ and lies on $\boldsymbol{S}$, the span of the input samples in $\mathcal{D}$. Then, given an unlearning request to forget any subset $\mathcal{D}_f$ from $\mathcal{D}$, $\boldsymbol{\theta}^*$ itself is a minimizer to the loss on the resulting retain set $\mathcal{D}_r = \mathcal{D} \setminus \mathcal{D}_f$. However, since $\boldsymbol{\theta}^* \in \boldsymbol{S}$ reveals information about all the input samples in $\mathcal{D}$, it necessarily leaks information about the samples in $\mathcal{D}_f$. Thus, even though $\boldsymbol{\theta}^*$ is a valid minimizer of $\mathcal{J}(\boldsymbol{\theta}; \mathcal{D}_r)$, it is not an acceptable unlearning solution.

## D Proofs

### D.1 Proof of Theorem 1

We assume $f(\boldsymbol{\theta}^*, \cdot)$ interpolates all of $\mathcal{D}$, so $f(\boldsymbol{\theta}^*, \boldsymbol{x}) = y$ for all $(\boldsymbol{x}, \boldsymbol{y}) \in \mathcal{D}$, and that the sample-wise loss $\mathcal{L}(\boldsymbol{\theta}, \boldsymbol{x}, \boldsymbol{y})$ is minimized when $f(\boldsymbol{\theta}, \boldsymbol{x}) = y$. Thus, $\boldsymbol{\theta}^*$ must minimize each of the sample-wise losses $\mathcal{L}(\boldsymbol{\theta}, \boldsymbol{x}, \boldsymbol{y})$ for all $(\boldsymbol{x}, \boldsymbol{y}) \in \mathcal{D}$. Therefore, $\nabla_{\boldsymbol{\theta}} \mathcal{L}(\boldsymbol{\theta}^*, \boldsymbol{x}, \boldsymbol{y}) = \mathbf{0}$ for all $(\boldsymbol{x}, \boldsymbol{y}) \in \mathcal{D}$.

Since $\mathcal{J}(\boldsymbol{\theta}^*; \mathcal{D}_r) = \frac{1}{|\mathcal{D}_r|} \sum_{(\boldsymbol{x}, \boldsymbol{y}) \in \mathcal{D}_r} \mathcal{L}(\boldsymbol{\theta}, \boldsymbol{x}, \boldsymbol{y})$ and $\mathcal{J}(\boldsymbol{\theta}^*; \mathcal{D}_f) = \frac{1}{|\mathcal{D}_f|} \sum_{(\boldsymbol{x}, \boldsymbol{y}) \in \mathcal{D}_f} \mathcal{L}(\boldsymbol{\theta}, \boldsymbol{x}, \boldsymbol{y})$, we must have that $\nabla_{\boldsymbol{\theta}} \mathcal{J}(\boldsymbol{\theta}^*; \mathcal{D}_r) = \nabla_{\boldsymbol{\theta}} \mathcal{J}(\boldsymbol{\theta}^*; \mathcal{D}_f) = \mathbf{0}$.

Then, if $M_{\text{LG}}$ is any loss-gradient unlearning method, the update rule must be of the form

$$M(\mathcal{A}, \mathcal{I}_r, \mathcal{A}(\mathcal{D}), \mathcal{D}_f) = \boldsymbol{\theta}^* - \boldsymbol{P}_r \nabla_{\boldsymbol{\theta}} \mathcal{J}(\boldsymbol{\theta}^*; \mathcal{D}_r) + \boldsymbol{P}_f \nabla_{\boldsymbol{\theta}} \mathcal{J}(\boldsymbol{\theta}^*; \mathcal{D}_f) + \boldsymbol{\xi},$$

where $\boldsymbol{P}_r$ and $\boldsymbol{P}_f$ are positive semi-definite matrices and $\boldsymbol{\xi}$ is a zero-mean random variable. Applying the fact that $\nabla_{\boldsymbol{\theta}} \mathcal{J}(\boldsymbol{\theta}^*; \mathcal{D}_r) = \nabla_{\boldsymbol{\theta}} \mathcal{J}(\boldsymbol{\theta}^*; \mathcal{D}_f) = \mathbf{0}$ to the update of $M_{\text{LG}}$ gives the desired result:

$$M_{\text{LG}}(\mathcal{A}, \mathcal{I}_r, \mathcal{A}(\mathcal{D}), \mathcal{D}_f) = \boldsymbol{\theta}^* + \boldsymbol{\xi}$$

### D.2 Proof of Theorem 2

Recall we have a feasible vector $\boldsymbol{\theta}^*$ such that $\boldsymbol{\theta}^{*\top} \boldsymbol{x} = y$ for all $(\boldsymbol{x}, y) \in \mathcal{D}$, and we want to recover $\boldsymbol{\theta}_r^*$, the minimum $\ell_2$ norm solution over just a subset $\mathcal{D}_r \subseteq \mathcal{D}$:

$$\boldsymbol{\theta}_r^* = \operatorname*{argmin}_{\boldsymbol{\theta}} \|\boldsymbol{\theta}\|_2 \quad \text{s.t. } \boldsymbol{\theta}^\top \boldsymbol{x} = y \quad \forall (\boldsymbol{x}, y) \in \mathcal{D}_r \tag{16}$$

Consider solving the relaxed unlearning problem (7) for $\tilde{R}(\boldsymbol{\theta}) = \|\boldsymbol{\theta}\|_2$:

$$\tilde{\boldsymbol{\Delta}} = \operatorname*{argmin}_{\boldsymbol{\Delta}} \|\boldsymbol{\theta}^* + \boldsymbol{\Delta}\|_2 \text{ s.t. } \boldsymbol{\Delta} \perp \boldsymbol{x} \quad \forall (\boldsymbol{x}, y) \in \mathcal{D}_r$$

Define $\boldsymbol{S}_r = \operatorname{span}\{\boldsymbol{x} \mid (\boldsymbol{x}, y) \in \mathcal{D}_r\}$ and write the equivalent problem:

$$\tilde{\boldsymbol{\Delta}} = \operatorname*{argmin}_{\boldsymbol{\Delta} \in \boldsymbol{S}_r^\perp} \frac{1}{2} \|\boldsymbol{\theta}^* + \boldsymbol{\Delta}\|_2^2$$

By first order optimality, $\boldsymbol{\theta}^* + \tilde{\boldsymbol{\Delta}} \in \boldsymbol{S}_r$, so we must have that

$$\tilde{\boldsymbol{\Delta}} = -\mathcal{P}_{\boldsymbol{S}_r^\perp}(\boldsymbol{\theta}^*)$$

Thus the updated unlearned vector is

$$\boldsymbol{\theta}^* + \tilde{\boldsymbol{\Delta}} = \boldsymbol{\theta}^* - \mathcal{P}_{\boldsymbol{S}_r^\perp}(\boldsymbol{\theta}^*) = \mathcal{P}_{\boldsymbol{S}_r}(\boldsymbol{\theta}^*).$$

Then, $\mathcal{P}_{\boldsymbol{S}_r}(\boldsymbol{\theta}^*) = \boldsymbol{\theta}_r^*$ by Lemma 5.

### D.3 Proof of Lemma 1

Recall that in this case we are interested in minimizing $R(\boldsymbol{\theta}) = \|\mathbf{w}(\boldsymbol{\theta})\|_2$, where $\mathbf{w}(\boldsymbol{\theta}) = \boldsymbol{A}_1^\top \cdots \boldsymbol{A}_{L-1}^\top \boldsymbol{c}$ returns the effective linear predictor parameterized by $\boldsymbol{\theta}$.

We first show that $\tilde{\boldsymbol{\Delta}}$ is feasible for the relaxed problem (7). Firstly, $\tilde{\boldsymbol{\Delta}}$ is zero in all entries except those corresponding to the perturbation of $\boldsymbol{A}_1$, so we only need to ensure that $\tilde{\boldsymbol{\Delta}}_{\boldsymbol{A}_1}$ is orthogonal to $\nabla_{\boldsymbol{A}_1} f(\boldsymbol{\theta}^*, \boldsymbol{x})$ for each $(\boldsymbol{x}, y) \in \mathcal{D}_r$. Recall we denote the retain set input space as $\boldsymbol{S}_r = \mathrm{span}\{\, \boldsymbol{x} \mid (\boldsymbol{x}, y) \in \mathcal{D}_r \,\}$, and $\tilde{\boldsymbol{\Delta}}_{\boldsymbol{A}_1}$ is defined as

$$\tilde{\boldsymbol{\Delta}}_{\boldsymbol{A}_1} = -\left\| \boldsymbol{A}_2^{*\top} \cdots \boldsymbol{A}_{L-1}^{*\top} \boldsymbol{c}^* \right\|_2^{-2} \boldsymbol{A}_2^{*\top} \cdots \boldsymbol{A}_{L-1}^{*\top} \boldsymbol{c}^* \mathcal{P}_{\boldsymbol{S}_r^\perp} \left( \mathbf{w}(\boldsymbol{\theta}^*) \right)^\top.$$

Further, the gradients are computed as

$$\nabla_{\boldsymbol{A}_1} f(\boldsymbol{\theta}^*, \boldsymbol{x}) = \boldsymbol{A}_2^{*\top} \cdots \boldsymbol{A}_{L-1}^{*\top} \boldsymbol{c}^* \boldsymbol{x}^\top$$

Then for any $(\boldsymbol{x}, y) \in \mathcal{D}_r$,

$$
\begin{aligned}
\langle \tilde{\boldsymbol{\Delta}}_{\boldsymbol{A}_1}, \nabla_{\boldsymbol{A}_1} f(\boldsymbol{\theta}^*, \boldsymbol{x}) \rangle &= \mathrm{tr}\left\{ \left( \tilde{\boldsymbol{\Delta}}_{\boldsymbol{A}_1} \right)^\top \nabla_{\boldsymbol{A}_1} f(\boldsymbol{\theta}^*, \boldsymbol{x}) \right\} \\
&= \mathrm{tr}\left\{ \nabla_{\boldsymbol{A}_1} f(\boldsymbol{\theta}^*, \boldsymbol{x}) \left( \tilde{\boldsymbol{\Delta}}_{\boldsymbol{A}_1} \right)^\top \right\} \\
&= -\left\| \boldsymbol{A}_2^{*\top} \cdots \boldsymbol{A}_{L-1}^{*\top} \boldsymbol{c}^* \right\|_2^{-2} \mathrm{tr}\left\{ \boldsymbol{A}_2^{*\top} \cdots \boldsymbol{A}_{L-1}^{*\top} \boldsymbol{c}^* \boldsymbol{x}^\top \mathcal{P}_{\boldsymbol{S}_r^\perp} \left( \mathbf{w}(\boldsymbol{\theta}^*) \right) \boldsymbol{c}^{*\top} \boldsymbol{A}_{L-1}^* \cdots \boldsymbol{A}_2^* \right\} \\
&= 0,
\end{aligned}
$$

where the last step follows from the fact that the inner term $\boldsymbol{x}^\top \mathcal{P}_{\boldsymbol{S}_r^\perp} \left( \mathbf{w}(\boldsymbol{\theta}^*) \right) = 0$ since $\boldsymbol{x} \in \mathcal{D}_r$ implies $\boldsymbol{x} \in \boldsymbol{S}_r$ by definition.

We now show that $\boldsymbol{\theta}^* + \tilde{\boldsymbol{\Delta}}$ achieves the optimal unlearning solution $\boldsymbol{\theta}^*$. By construction of $\tilde{\boldsymbol{\Delta}}$, the only entries of $\boldsymbol{\theta}^*$ that are perturbed are those which correspond to $\boldsymbol{A}_1$. Thus, we compute the effective linear predictor after the perturbation:

$$
\begin{aligned}
\mathbf{w}(\boldsymbol{\theta}^* + \tilde{\boldsymbol{\Delta}}) &= \mathbf{w}(\boldsymbol{\theta}^*) + \tilde{\boldsymbol{\Delta}}_{\boldsymbol{A}_1}^\top \boldsymbol{A}_2^{*\top} \cdots \boldsymbol{A}_{L-1}^{*\top} \boldsymbol{c}^* \\
&= \mathbf{w}(\boldsymbol{\theta}^*) - \left\| \boldsymbol{A}_2^{*\top} \cdots \boldsymbol{A}_{L-1}^{*\top} \boldsymbol{c}^* \right\|_2^{-2} \mathcal{P}_{\boldsymbol{S}_r^\perp} \left( \mathbf{w}(\boldsymbol{\theta}^*) \right) \boldsymbol{c}^{*\top} \boldsymbol{A}_{L-1}^* \cdots \boldsymbol{A}_2^* \boldsymbol{A}_2^{*\top} \cdots \boldsymbol{A}_{L-1}^{*\top} \boldsymbol{c}^* \\
&= \mathbf{w}(\boldsymbol{\theta}^*) - \left\| \boldsymbol{A}_2^{*\top} \cdots \boldsymbol{A}_{L-1}^{*\top} \boldsymbol{c}^* \right\|_2^{-2} \mathcal{P}_{\boldsymbol{S}_r^\perp} \left( \mathbf{w}(\boldsymbol{\theta}^*) \right) \left( \boldsymbol{A}_2^{*\top} \cdots \boldsymbol{A}_{L-1}^{*\top} \boldsymbol{c}^* \right)^\top \boldsymbol{A}_2^{*\top} \cdots \boldsymbol{A}_{L-1}^{*\top} \boldsymbol{c}^* \\
&= \mathbf{w}(\boldsymbol{\theta}^*) - \mathcal{P}_{\boldsymbol{S}_r^\perp} \left( \mathbf{w}(\boldsymbol{\theta}^*) \right) \\
&= \mathcal{P}_{\boldsymbol{S}_r} \left( \mathbf{w}(\boldsymbol{\theta}^*) \right)
\end{aligned}
$$

Since the linear predictor $\mathbf{w}(\boldsymbol{\theta}^*)$ already interpolated $\mathcal{D}$, $\mathcal{P}_{\boldsymbol{S}_r} \left( \mathbf{w}(\boldsymbol{\theta}^*) \right)$ must be the minimum norm linear predictor over $\mathcal{D}_r$ by Lemma 5. Thus, the effective predictor of the perturbed parameters $\mathbf{w}(\boldsymbol{\theta}^* + \tilde{\boldsymbol{\Delta}})$ solves the exact unlearning problem (4) when $R(\boldsymbol{\theta}) = \|\mathbf{w}(\boldsymbol{\theta})\|_2$, so $\boldsymbol{\theta}^* + \tilde{\boldsymbol{\Delta}}$ achieves the optimal unlearning solution.

### D.4 Proof of Theorem 3

Recall for this theorem we analyze $R(\boldsymbol{\theta}) = \|\mathbf{w}(\boldsymbol{\theta})\|_2$. Let $\tilde{\boldsymbol{\Delta}}$ be the perturbation which satisfies the conditions in Lemma 1. Then, $\tilde{\boldsymbol{\Delta}}$ is feasible for the relaxed problem (7), and further $\boldsymbol{\theta}^* + \tilde{\boldsymbol{\Delta}}$ solves the exact unlearning problem (4).

Now, let $\boldsymbol{\Delta}^*$ minimize the relaxed problem (7) for this $\tilde{R}$ defined in (9). Then because $\tilde{R}$ ensures that all elements of $\boldsymbol{\Delta}^*$ which do not correspond to $\boldsymbol{A}_1$ are zero, we must have that for any $(\boldsymbol{x}, y) \in \mathcal{D}_r$:

$$
\begin{aligned}
\mathbf{w}(\boldsymbol{\theta}^* + \boldsymbol{\Delta}^*)^\top \boldsymbol{x} &= \boldsymbol{c}^{*\top} \boldsymbol{A}_{L-1}^* \cdots \boldsymbol{A}_2^* \left( \boldsymbol{A}_1^* + \boldsymbol{\Delta}_{\boldsymbol{A}_1}^* \right) \boldsymbol{x} \\
&= y + \boldsymbol{c}^{*\top} \boldsymbol{A}_{L-1}^* \cdots \boldsymbol{A}_2^* \boldsymbol{\Delta}_{\boldsymbol{A}_1}^* \boldsymbol{x} \\
&= y + \langle \boldsymbol{\Delta}^*, \nabla_{\boldsymbol{\theta}} f(\boldsymbol{\theta}^*, \boldsymbol{x}) \rangle \\
&= y,
\end{aligned}
$$

where the last equality follows from the feasibility of $\boldsymbol{\Delta}^*$ to (7). Thus, $\boldsymbol{\theta}^* + \boldsymbol{\Delta}^*$ interpolates $\mathcal{D}_r$, so $\boldsymbol{\theta}^* + \boldsymbol{\Delta}^*$ is feasible for the exact unlearning problem (4). We now show this point is also optimal for (4).

Since $\boldsymbol{\theta}^* + \tilde{\boldsymbol{\Delta}}$ solves the exact unlearning problem (4) and $\boldsymbol{\theta}^* + \boldsymbol{\Delta}^*$ is another feasible point, we must have that

$$R(\boldsymbol{\theta}^* + \tilde{\boldsymbol{\Delta}}) \leq R(\boldsymbol{\theta}^* + \boldsymbol{\Delta}^*).$$

Further, both $\tilde{\boldsymbol{\Delta}}$ and $\boldsymbol{\Delta}^*$ are feasible for (7) and $\boldsymbol{\Delta}^*$ is defined as the solution to (7), so we must have that

$$\tilde{R}(\boldsymbol{\theta}^* + \boldsymbol{\Delta}^*) \leq \tilde{R}(\boldsymbol{\theta}^* + \tilde{\boldsymbol{\Delta}}).$$

But, since both $\tilde{\boldsymbol{\Delta}}$ and $\boldsymbol{\Delta}^*$ are non-zero only in the entries corresponding to $\boldsymbol{A}_1$, applying $R$ and $\tilde{R}$ yields the same value:

$$R(\boldsymbol{\theta}^* + \tilde{\boldsymbol{\Delta}}) = \tilde{R}(\boldsymbol{\theta}^* + \tilde{\boldsymbol{\Delta}}) \quad \text{and} \quad R(\boldsymbol{\theta}^* + \boldsymbol{\Delta}^*) = \tilde{R}(\boldsymbol{\theta}^* + \boldsymbol{\Delta}^*)$$

Thus, $R(\boldsymbol{\theta}^* + \boldsymbol{\Delta}^*) = R(\boldsymbol{\theta}^* + \tilde{\boldsymbol{\Delta}})$, so $\boldsymbol{\theta}^* + \boldsymbol{\Delta}^*$ achieves the optimal objective value of (4). Since we established feasibility and optimality, $\boldsymbol{\theta}^* + \boldsymbol{\Delta}^*$ must solve (4).

### D.4.1 Necessity of Additional Regularizer $\hat{R}$ for Theorem 3

In this section, we show that minimizing just $R$ over the relaxed constraints, i.e. solving (6), for $R$ which measures the linear network predictor norm does not solve the exact unlearning solution. Because there is no control the size and direction of the perturbation $\boldsymbol{\Delta}$, we can construct a simple example where $\boldsymbol{\Delta}$ satisfies just the linearization of the data interpolation constraints but the updated network $\boldsymbol{\theta}^* + \boldsymbol{\Delta}$ no longer interpolates $\mathcal{D}_r$.

Consider a dataset of two samples $\mathcal{D} = \{(\boldsymbol{e}_1, 1), (\boldsymbol{e}_2, 1)\}$, where $\boldsymbol{e}_i \in \mathbb{R}^m$ is the $i$th standard basis vector for any $m \geq 3$. Consider the original 2-layer interpolating network trained on $\mathcal{D}$ defined by parameters $\boldsymbol{\theta}^* = [\boldsymbol{c}^* ; \text{vec}(\boldsymbol{A}^*)]$, where $\boldsymbol{c}^* = \boldsymbol{e}_1 + \boldsymbol{e}_2 \in \mathbb{R}^m$ and $\boldsymbol{A}^*$ is the $m \times m$ identity matrix $\boldsymbol{A}^* = \boldsymbol{I}_m$, so $f(\boldsymbol{\theta}^*, \boldsymbol{x}) = \boldsymbol{c}^{*\top} \boldsymbol{A}^* \boldsymbol{x} = (\boldsymbol{e}_1 + \boldsymbol{e}_2)^\top \boldsymbol{x}$.

We set $\mathcal{D}_r = \{(\boldsymbol{e}_1, 1)\}$ and $\mathcal{D}_f = \{(\boldsymbol{e}_2, 1)\}$, and define the perturbation variable $\boldsymbol{\Delta} = [\boldsymbol{\Delta}_c ; \text{vec}(\boldsymbol{\Delta}_A)]$. Translating the constraints of (6) to this specific problem instance, we have that

$$\boldsymbol{\Delta}_c^\top \boldsymbol{e}_1 + \text{tr}\{\boldsymbol{\Delta}_A^\top (\boldsymbol{e}_1 + \boldsymbol{e}_2)\boldsymbol{e}_1^\top\} = 0$$

We then select the values $\boldsymbol{\Delta}_c = -\boldsymbol{e}_3$ and $\boldsymbol{\Delta}_A = \boldsymbol{e}_3 \boldsymbol{e}_1^\top - \boldsymbol{e}_2 \boldsymbol{e}_2^\top - \boldsymbol{e}_3 \boldsymbol{e}_3^\top$. It is easy to see that these choices satisfy the above constraint. Further, they achieve exact minimization of (6). We show below that the resulting network's predictor $(\boldsymbol{A}^* + \boldsymbol{\Delta}_A)^\top (\boldsymbol{c}^* + \boldsymbol{\Delta}_c) = \boldsymbol{0}$.

$$
\begin{aligned}
R(\boldsymbol{\theta}^* + \boldsymbol{\Delta}) &= \left\| (\boldsymbol{A}^* + \boldsymbol{\Delta}_A)^\top (\boldsymbol{c}^* + \boldsymbol{\Delta}_c) \right\|_2 \\
&= \left\| (\boldsymbol{I} + \boldsymbol{e}_3 \boldsymbol{e}_1^\top - \boldsymbol{e}_2 \boldsymbol{e}_2^\top - \boldsymbol{e}_3 \boldsymbol{e}_3^\top)^\top (\boldsymbol{e}_1 + \boldsymbol{e}_2 - \boldsymbol{e}_3) \right\|_2 \\
&= \left\| (\boldsymbol{I} + \boldsymbol{e}_1 \boldsymbol{e}_3^\top - \boldsymbol{e}_2 \boldsymbol{e}_2^\top - \boldsymbol{e}_3 \boldsymbol{e}_3^\top)(\boldsymbol{e}_1 + \boldsymbol{e}_2 - \boldsymbol{e}_3) \right\|_2 \\
&= \left\| \boldsymbol{e}_1 + \boldsymbol{e}_2 - \boldsymbol{e}_3 - \boldsymbol{e}_2 - \boldsymbol{e}_1 + \boldsymbol{e}_3 \right\|_2 \\
&= \| \boldsymbol{0} \|_2 = 0
\end{aligned}
$$

Thus, the updated network which solves (6) predicts the constant function at $\boldsymbol{0}$ for all inputs $\boldsymbol{x}$, as $f(\boldsymbol{\theta}^* + \boldsymbol{\Delta}, \boldsymbol{x}) = \left( (\boldsymbol{A}^* + \boldsymbol{\Delta}_A)^\top (\boldsymbol{c}^* + \boldsymbol{\Delta}_c) \right)^\top \boldsymbol{x} = \boldsymbol{0}^\top \boldsymbol{x} = 0$.

This clearly does not interpolate $\mathcal{D}_r$, and this example as a whole demonstrates that failing to control the size and direction of the drift term $\boldsymbol{\Delta}$ beyond just the linearized constraints does not lead to the exact unlearning solution.

### D.5 Proof of Theorem 4

Denote the minimum $\ell_2$ norm solution $\mathbf{w}(\hat{\boldsymbol{\theta}}_r^*)$ to $\boldsymbol{y} = \boldsymbol{X}\mathbf{w}$ as just $\mathbf{w}_r^*$ for brevity. Using $\mathbf{w}_r^*$, we construct a solution to the exact unlearning problem (4) for $R(\boldsymbol{\theta}) = \|\boldsymbol{\theta}\|_2$, which we restate below:

$$\underset{\boldsymbol{\theta}}{\arg\min} \|\boldsymbol{\theta}\|_2 \text{ s.t. } \mathbf{w}(\boldsymbol{\theta})^\top \boldsymbol{x} = y \quad \forall (\boldsymbol{x}, y) \in \mathcal{D}_r$$

Expanding $\boldsymbol{\theta} = [\boldsymbol{c}\,;\,\mathrm{vec}(\boldsymbol{A}_1)\,;\,\ldots\,;\,\mathrm{vec}(\boldsymbol{A}_{L-1})]$ into the sub-parameters, squaring the objective, and organizing $(\boldsymbol{x}, y) \in \mathcal{D}_r$ into input data matrix $\boldsymbol{X}_r \in \mathbb{R}^{|\mathcal{D}_r| \times d}$ and output vector $\boldsymbol{y}_r \in \mathbb{R}^{|\mathcal{D}_r|}$ gives an equivalent problem:

$$\operatorname*{argmin}_{\boldsymbol{c}, \boldsymbol{A}_1, \ldots, \boldsymbol{A}_{L-1}} \|\boldsymbol{c}\|_2^2 + \sum_{\ell=1}^{L-1} \|\boldsymbol{A}_\ell\|_F^2 \text{ s.t. } \boldsymbol{y}_r = \boldsymbol{X}_r \boldsymbol{A}_1^\top \ldots \boldsymbol{A}_{L-1}^\top \boldsymbol{c} \tag{17}$$

Let $\boldsymbol{c}^*, \boldsymbol{A}_1^*, \ldots, \boldsymbol{A}_{L-1}^*$ be a solution to (17). Then, $\boldsymbol{A}_1^{*\top} \ldots \boldsymbol{A}_{L-1}^{*\top} \boldsymbol{c}^*$ interpolates $\mathcal{D}_r$, so $\boldsymbol{A}_1^{*\top} \ldots \boldsymbol{A}_{L-1}^{*\top} \boldsymbol{c}^* = \mathbf{w}_r^* + \boldsymbol{z}$ where $\mathbf{w}_r^* \in \mathrm{row}(\boldsymbol{X}_r)$ and $\boldsymbol{z} \in \ker(\boldsymbol{X}_r)$ by Lemma 4.

Let $\boldsymbol{P}_{\mathbf{w}_r^*} = \frac{1}{\|\mathbf{w}_r^*\|_2^2} \mathbf{w}_r^* \mathbf{w}_r^{*\top}$ be the projection matrix onto $\mathrm{span}(\mathbf{w}_r^*)$. Then replacing $\boldsymbol{A}_1^*$ with $\boldsymbol{A}_1^* \boldsymbol{P}_{\mathbf{w}_r^*}$ maintains feasibility since $\boldsymbol{P}_{\mathbf{w}_r^*}^\top \boldsymbol{A}_1^{*\top} \ldots \boldsymbol{A}_{L-1}^{*\top} \boldsymbol{c}^* = \boldsymbol{P}_{\mathbf{w}_r^*}(\mathbf{w}_r^* + \boldsymbol{z}) = \mathbf{w}_r^*$ which is feasible by definition. Further, $\boldsymbol{A}_1^* \boldsymbol{P}_{\mathbf{w}_r^*}$ achieves smaller objective function value since

$$\left\|\boldsymbol{A}_1^* \boldsymbol{P}_{\mathbf{w}_r^*}\right\|_F^2 = \mathrm{tr}\{\boldsymbol{A}_1^* \boldsymbol{P}_{\mathbf{w}_r^*} \boldsymbol{P}_{\mathbf{w}_r^*} \boldsymbol{A}_1^{*\top}\} = \mathrm{tr}\{\boldsymbol{P}_{\mathbf{w}_r^*} \boldsymbol{A}_1^{*\top} \boldsymbol{A}_1^*\} \leq \left\|\boldsymbol{P}_{\mathbf{w}_r^*}\right\|_2 \left\|\boldsymbol{A}_1^{*\top} \boldsymbol{A}_1^*\right\|_* = \|\boldsymbol{A}_1^*\|_F^2.$$

The second equality follows from the cyclic property of trace and the fact that $\boldsymbol{P}_{\mathbf{w}_r^*}$ is both symmetric and idempotent, and the inequality is a generalized Hölder's inequality for matrices.

Thus, replacing $\boldsymbol{A}_1^*$ with the rank-1 matrix $\boldsymbol{A}_1^* \boldsymbol{P}_{\mathbf{w}_r^*}$ must preserve optimality of any solution that contains $\boldsymbol{A}_1^*$. Write $\boldsymbol{A}_1^* \boldsymbol{P}_{\mathbf{w}_r^*} = \lambda_1 \boldsymbol{v}_1 \mathbf{w}_r^{*\top}$ for some $\lambda_1 \in \mathbb{R}$, $\boldsymbol{v}_1 \in \mathbb{R}^{h_\ell}$ with $\|\boldsymbol{v}_1\|_2 = 1$.

We can apply an analogous argument with the matrix $\boldsymbol{P}_{\boldsymbol{v}_1}$, which projects its input onto $\mathrm{span}(\boldsymbol{v}_1)$, to show that any solution that contains $\boldsymbol{A}_2^*$ must remain optimal with $\boldsymbol{A}_2^*$ replaced by the rank-1 matrix $\boldsymbol{A}_2^* \boldsymbol{P}_{\boldsymbol{v}_1}$. Continuing this argument for each $\boldsymbol{A}_\ell^*$, $\ell = 3, \ldots, L-1$ as well as for $\boldsymbol{c}^*$ shows that we can search for solution over a much smaller space. Specifically, for some $\lambda_\ell \in \mathbb{R}$ and $\boldsymbol{v} \in \mathbb{R}^{h_\ell}$, we can decompose $\boldsymbol{c}^*$ and each $\boldsymbol{A}_\ell^*$ as

$$\boldsymbol{A}_1^* = \lambda_1 \boldsymbol{v}_1 \mathbf{w}_r^{*\top} \qquad \boldsymbol{A}_\ell^* = \lambda_\ell \boldsymbol{v}_\ell \boldsymbol{v}_{\ell-1}^\top \text{ for } \ell = 2, \ldots, L-1 \qquad \boldsymbol{c}^* = \lambda_L \boldsymbol{v}_{L-1}$$

Then, (17) reduces to

$$\min_{\lambda_i, \boldsymbol{v}_\ell} \|\lambda_L \boldsymbol{v}_{L-1}\|_2^2 + \left\|\lambda_1 \boldsymbol{v}_1 \mathbf{w}_r^{*\top}\right\|_F^2 + \sum_{\ell=2}^{L-1} \left\|\lambda_\ell \boldsymbol{v}_\ell \boldsymbol{v}_{\ell-1}^\top\right\|_F^2$$

$$\text{s.t.} \quad (\lambda_1 \mathbf{w}_r^* \boldsymbol{v}_1^\top)(\lambda_2 \boldsymbol{v}_1 \boldsymbol{v}_2^\top) \cdots (\lambda_{L-1} \boldsymbol{v}_{L-2} \boldsymbol{v}_{L-1}^\top)(\lambda_L \boldsymbol{v}_{L-1}) = \mathbf{w}_r^* \text{ and } \|\boldsymbol{v}_\ell\|_2 = 1$$

$$= \min_{\lambda_i} \|\mathbf{w}_r^*\|_2^2 \lambda_1^2 + \sum_{\ell=2}^{L} \lambda_\ell^2 \quad \text{s.t.} \quad \lambda_1 \lambda_2 \cdots \lambda_L = 1 \tag{18}$$

We perform a change of variables setting $\gamma_i = \lambda_i^2$ and enforcing $\gamma_i > 0$.

$$\min_{\gamma_i > 0} \|\mathbf{w}_r^*\|_2^2 \gamma_1 + \sum_{\ell=2}^{L} \gamma_\ell \quad \text{s.t.} \quad \gamma_1 \gamma_2 \cdots \gamma_L = 1 \tag{19}$$

Define $\boldsymbol{\gamma} = (\gamma_1, \ldots, \gamma_L)$, objective function $g(\boldsymbol{\gamma}) = \|\mathbf{w}_r^*\|_2^2 \gamma_1 + \sum_{\ell=2}^{L} \gamma_\ell$, and constraint $h(\boldsymbol{\gamma}) = \gamma_1 \gamma_2 \cdots \gamma_L - 1 = 0$. By the AM-GM inequality, we have that for any feasible $\boldsymbol{\gamma}$

$$g(\boldsymbol{\gamma}) \geq L \left(\|\mathbf{w}_r^*\|_2^2 \gamma_1 \cdots \gamma_L\right)^{\frac{1}{L}} = L \|\mathbf{w}_r^*\|_2^{2/L},$$

where the last equality follows from the constraint $h(\boldsymbol{\gamma}) = 0$. Define feasible point $\boldsymbol{\gamma}^*$ such that

$$\boldsymbol{\gamma}^* = \left(\|\mathbf{w}_r^*\|_2^{\frac{2(1-L)}{L}}, \|\mathbf{w}_r^*\|_2^{\frac{2}{L}}, \ldots, \|\mathbf{w}_r^*\|_2^{\frac{2}{L}}\right).$$

Then $g(\boldsymbol{\gamma}^*) = \|\mathbf{w}_r^*\|_2^{2/L}$ achieves the lower bound, so it must solve (19). Thus, the optimal values $\lambda_1^*, \ldots, \lambda_L^*$ to (18) result from taking square roots of $\boldsymbol{\gamma}_\ell^*$. Then, the following values for the network parameters must be optimal for (17):

$$\boldsymbol{A}_1^* = \|\mathbf{w}_r^*\|_2^{\frac{(1-L)}{L}} \boldsymbol{v}_1 \mathbf{w}_r^{*\top} \qquad \boldsymbol{A}_\ell^* = \|\mathbf{w}_r^*\|_2^{\frac{1}{L}} \boldsymbol{v}_\ell \boldsymbol{v}_{\ell-1}^\top \text{ for } \ell = 2, \ldots, L-1 \qquad \boldsymbol{c}^* = \|\mathbf{w}_r^*\|_2^{\frac{1}{L}} \boldsymbol{v}_{L-1}.$$

### D.6 Proof of Theorem 5

We prove the theorem using the following lemma. See the end of the section for a proof.

**Lemma 6.** *For $\boldsymbol{c} \in \mathbb{R}^h$ and subspace $\mathcal{G} \subseteq \mathbb{R}^h$ such that $\dim(\mathcal{G}) = s$, there exists $\boldsymbol{\Delta}_c \in \mathcal{G}_r^\perp$ such that $\|\boldsymbol{c} + \boldsymbol{\Delta}_c\|_0 \leq s$, where the $\ell_0$-"norm" $\|\cdot\|_0$ counts the number of non-zero elements.*

Because $\hat{R}$ does not allow any perturbation of $\boldsymbol{A}^*$, any solution to (12) must only perturb $\boldsymbol{\theta}^*$ in the entries corresponding to $\boldsymbol{c}^*$.

Let $s = \dim(\text{span}\{\phi(\boldsymbol{A}^*\boldsymbol{x})\}_{(\boldsymbol{x},y)\in\mathcal{D}_r})$. Note that by definition we have that $s \leq |\mathcal{D}_r|$. Apply the lemma to $\boldsymbol{c}^*$ and $\text{span}\{\phi(\boldsymbol{A}^*\boldsymbol{x})\}_{(\boldsymbol{x},y)\in\mathcal{D}_r}$ so that there exists $\tilde{\boldsymbol{\Delta}}_c \in \text{span}\left(\{\phi(\boldsymbol{A}^*\boldsymbol{x})\}_{(\boldsymbol{x},y)\in\mathcal{D}_r}\right)^\perp$ such that $\|\boldsymbol{c}^* + \tilde{\boldsymbol{\Delta}}_c\|_0 \leq s$. Define $\tilde{\boldsymbol{\Delta}} = \left[\tilde{\boldsymbol{\Delta}}_c \,;\, \boldsymbol{0}\right]$.

Then the network defined by $\boldsymbol{\theta}^* + \tilde{\boldsymbol{\Delta}}$ has at most $s$ active neurons since any zero element of $\boldsymbol{c}^* + \tilde{\boldsymbol{\Delta}}_c$ cannot contribute an active neuron. Further, $\{\phi(\boldsymbol{A}^*\boldsymbol{x})\}_{(\boldsymbol{x},y)\in\mathcal{D}_r} = \{\nabla_{\boldsymbol{c}} f(\boldsymbol{\theta}^*, \boldsymbol{x})\}_{\boldsymbol{x},y\in\mathcal{D}_r}$, so the perturbation $\tilde{\boldsymbol{\Delta}}$ is feasible for the relaxed problem (7). But, $f$ is linear in $\boldsymbol{c}$, so this perturbation must preserve function value on $\mathcal{D}_r$, since the constraints of the relaxed problem are tight when just perturbing $\boldsymbol{c}^*$. Thus, the resulting network defined by $\boldsymbol{\theta}^* + \tilde{\boldsymbol{\Delta}}$ both interpolates $\mathcal{D}_r$ and has at most $s = \dim(\text{span}\{\phi(\boldsymbol{A}^*\boldsymbol{x})\}_{(\boldsymbol{x},y)\in\mathcal{D}_r})$ active neurons.

Note that this construction of $\tilde{\boldsymbol{\Delta}}$ satisfies the conditions of Lemma 2, so we do not include a separate proof of the lemma since it is contained within the larger proof of the theorem.

*Proof of Lemma 6*:

Let the columns of some $\boldsymbol{P} \in \mathbb{R}^{h\times(h-s)}$ form a basis for $\mathcal{G}^\perp$ so that $\text{im}(\boldsymbol{P}) = \mathcal{G}^\perp$. Consider the reduced column echelon form of $\boldsymbol{P}$ denoted $\text{rcef}(\boldsymbol{P}) = \tilde{\boldsymbol{P}}$. By definition, $\text{im}(\tilde{\boldsymbol{P}}) = \text{im}(\boldsymbol{P}) = \mathcal{G}^\perp$, so $\text{rank}(\tilde{\boldsymbol{P}}) = h - s$ and thus each of the $h - s$ columns of $\tilde{\boldsymbol{P}}$ has a leading one. Let $\tilde{\boldsymbol{p}}_i$ be the $i$th column of $\tilde{\boldsymbol{P}}$ and let $j_i$ denote the index of the leading one in $\tilde{\boldsymbol{p}}_i$ for all $i \in [h - s]$.

Let $(\tilde{\boldsymbol{p}}_i)_k$ denote the $k$th element of $\tilde{\boldsymbol{p}}_i$. By definition of the reduced column echelon form, we have that $(\tilde{\boldsymbol{p}}_i)_k = 0$ for all $k < j_i$. Define

$$\boldsymbol{\Delta}_c = \sum_{i=1}^{h-s} \gamma_i \tilde{\boldsymbol{p}}_i$$

for coefficients $\gamma_i \in \mathbb{R}$ defined as

$$\gamma_i = -\left(\boldsymbol{c}^* + \sum_{k=1}^{i-1} \tilde{\boldsymbol{p}}_k\right)_{j_i}$$

Since each $\tilde{\boldsymbol{p}}_i$ is only non-zero in the indices $j_i$ to $h$, we must have that $(\boldsymbol{c}^* + \boldsymbol{\Delta}_c)_{j_i} = 0$ for all $i \in [h - s]$, so $\|\boldsymbol{c}^* + \boldsymbol{\Delta}_c\|_0 \leq s$.

### D.7 Proof of Proposition 1

Consider any parameter vector $\boldsymbol{\theta} = [\boldsymbol{c}\,;\,\text{vec}(\boldsymbol{A})]$. Then for any input $\boldsymbol{x}$, we can write $f(\boldsymbol{\theta}, \boldsymbol{x}) = \sum_{i=1}^h c_i\phi(\boldsymbol{a}_i^\top\boldsymbol{x})$ where $c_i$ is the $i$th element of $\boldsymbol{c}$ and $\boldsymbol{a}_i^\top$ is the $i$th row of $\boldsymbol{A}$. Consider the updated parameters $\hat{\boldsymbol{\theta}} = [\boldsymbol{c}\,;\,\text{vec}(\hat{\boldsymbol{A}})]$ for $\hat{\boldsymbol{A}} = (\mathbf{1}_{\boldsymbol{c}\neq 0}, \mathbf{1}^\top) \odot \boldsymbol{A}$. Then,

$$f(\boldsymbol{\theta}, \boldsymbol{x}) = \sum_{i=1}^h c_i\phi(\boldsymbol{a}_i^\top\boldsymbol{x}) = \sum_{i=1}^h c_i\phi(\mathbb{1}\{c_i \neq 0\}\boldsymbol{a}_i^\top\boldsymbol{x}) = f(\hat{\boldsymbol{\theta}}, \boldsymbol{x}),$$

where the second equality follows from the fact that we can set $\boldsymbol{a}_i$ to be zero whenever $\boldsymbol{c}_i = 0$ since that neuron does not contribute to the function output whenever $\boldsymbol{c}_i = 0$. Further, changing $\boldsymbol{a}_i$ for any $i$ where $\boldsymbol{c}_i = 0$ does not change the number of neurons, since if for the $i$th neuron we have $\boldsymbol{c}_i = 0$, then this neuron can never be active no matter the value of $\boldsymbol{a}_i$:

$$R(\boldsymbol{\theta}) = \sum_{i=1}^{h} \mathbb{1}\{|\boldsymbol{c}_i|\,\|\boldsymbol{a}_i\|_2 > 0\} = \sum_{i\,:\,\boldsymbol{c}_i \neq 0} \mathbb{1}\{\boldsymbol{a}_i \neq \mathbf{0}\} = \sum_{i\,:\,\boldsymbol{c}_i \neq 0} \mathbb{1}\{\hat{\boldsymbol{a}}_i \neq \mathbf{0}\} = R(\hat{\boldsymbol{\theta}}),$$

where $\hat{\boldsymbol{a}}_i^\top$ is the $i$th row of $\hat{\boldsymbol{A}}$. Lastly, since $\hat{\boldsymbol{a}}_i$ is always equal to $\mathbf{0}$ when $\boldsymbol{c}_i = 0$, we must have that $\hat{\boldsymbol{A}}$ has at most $R(\hat{\boldsymbol{\theta}})$ number of nonzero rows.

## E   MinNorm-OG Algorithm

We derive the closed form solution of (7) for the specific choice $\tilde{R}(\boldsymbol{\theta} + \boldsymbol{\Delta}) = \|\boldsymbol{\theta} + \boldsymbol{\Delta}\|_2^2 + \lambda\,\|\boldsymbol{\Delta}\|_2^2$.

Define the span of the model gradients over $\mathcal{D}_r$ as the subspace $\mathcal{G}_r = \mathrm{span}\{\nabla_{\boldsymbol{\theta}} f(\boldsymbol{\theta}, \boldsymbol{x})\}_{(\boldsymbol{x},\boldsymbol{y}) \in \mathcal{D}_r}$ and consider any $\lambda \geq 0$. We then solve the following problem:

$$\tilde{\boldsymbol{\Delta}} = \operatorname*{argmin}_{\boldsymbol{\Delta}} \|\boldsymbol{\theta} + \boldsymbol{\Delta}\|_2^2 + \lambda\,\|\boldsymbol{\Delta}\|_2^2 \quad \text{s.t. } \boldsymbol{\Delta} \in \mathcal{G}_r^\perp. \tag{20}$$

This is a strongly convex problem over a linear constraint, so its solution $\tilde{\boldsymbol{\Delta}}$ is the unique point which satisfies the following condition for first order optimality:

$$(1 + \lambda)\tilde{\boldsymbol{\Delta}} + \boldsymbol{\theta} \in \mathcal{G}_r.$$

Note that this is satisfied by the projection

$$\tilde{\boldsymbol{\Delta}} = -\frac{1}{1 + \lambda} \mathcal{P}_{\mathcal{G}_r^\perp}(\boldsymbol{\theta}),$$

which must then be the unique solution to (20).

## F   Experiments

We first standardize the notation for each algorithm. Throughout our experiments, we sweep over hyperparameters and report the best results for each algorithm, and we sweep related hyperparameters for each algorithm through the same set of values. For example, every algorithm has a learning rate which is selected from searching over the same set of values. We first define the hyperparameter names we use along with the algorithms they apply to.

Table 4: Hyperparameter definitions and their associated methods.

| Symbol | Methods | Description |
|---|---|---|
| $T$ | All | Number of epochs |
| $\eta$ | All | Learning rate |
| $\lambda_{\mathrm{GA}}$ | NGP, Scrub, SalUn | Loss ascent coefficient |
| $\lambda_{\mathrm{reg}}$ | NPO, Scrub, MinNorm-OG, Ridge, $\ell_1$-Sparse | Regularization coefficient |
| $\sigma$ | NGD | Gradient noise standard deviation |
| $T_{\mathrm{GD}}$ | Scrub, MinNorm-OG | Number of final descent epochs on retain set |
| $\gamma_{\mathrm{reg}}$ | MinNorm-OG, Ridge | Regularization coefficient decay rate |
| $T_{\mathrm{Proj}}$ | MinNorm-OG | Projection period |
| $n_{\mathrm{pert}}$ | MinNorm-OG | Subsample size to compute gradient space |

### F.1   Implementations

We now define the exact implementation of each method. Consider a batch of retain samples $\mathcal{B}_r$ and forget samples $\mathcal{B}_f$, along with loss function $\mathcal{J}$. For each method, we use the AdamW optimizer with learning rate $\eta$ on different effective loss functions. We express the loss functions below.

### F.1.1 Retrain and GD

Retrain and GD share the same loss function, but Retrain initializes a new model from scratch before optimizing this loss.

$$\mathcal{J}_{\text{Retrain}}(\boldsymbol{\theta}\,;\mathcal{B}_r) = \mathcal{J}_{\text{GD}}(\boldsymbol{\theta}\,;\mathcal{B}_r) = \mathcal{J}\,(\boldsymbol{\theta}\,;\mathcal{B}_r)$$

### F.1.2 GA

$$\mathcal{J}_{\text{GA}}(\boldsymbol{\theta}\,;\mathcal{B}_f) = -\mathcal{J}\,(\boldsymbol{\theta}\,;\mathcal{B}_f)$$

### F.1.3 NGD

$$\mathcal{J}_{\text{NGD}}(\boldsymbol{\theta}\,;\mathcal{B}_r) = \mathcal{J}\,(\boldsymbol{\theta}\,;\mathcal{B}_r) + \boldsymbol{\theta}^\top\boldsymbol{\xi},$$

where $\boldsymbol{\xi} \sim \mathcal{N}(\mathbf{0}, \sigma^2\boldsymbol{I})$ is a zero-mean Gaussian random vector.

### F.1.4 NGP

$$\mathcal{J}_{\text{NGP}}(\boldsymbol{\theta}\,;\mathcal{B}_r, \mathcal{B}_f) = \mathcal{J}\,(\boldsymbol{\theta}\,;\mathcal{B}_r) - \lambda_{\text{GA}}\mathcal{J}\,(\boldsymbol{\theta}\,;\mathcal{B}_f)$$

### F.1.5 Ridge

We store a regularization weighting $\lambda$ which we initialize to $\lambda = \lambda_{\text{reg}}$. We define the Ridge loss as

$$\mathcal{J}_{\text{Ridge}}(\boldsymbol{\theta}\,;\mathcal{B}_r) = \mathcal{J}\,(\boldsymbol{\theta}\,;\mathcal{B}_r) + \lambda\,\|\boldsymbol{\theta}\|_2^2\,.$$

After updating the parameter vector using this loss on each batch, we update $\lambda$ as

$$\lambda \leftarrow \gamma_{\text{reg}}\lambda.$$

Note that $\gamma_{\text{reg}}$ is always set within the range $(0, 1)$, so the update to $\lambda$ approximates the limit as $\lambda$ goes to 0 as we iterate through the epochs. This attempts to recover the minimum-norm training loss minimizer.

### F.1.6 $\ell_1$-Sparse

For each epoch $t = 1, \ldots, T$, we define the $\ell_1$-Sparse loss as

$$\mathcal{J}_{\ell_1\text{-Sparse}}(\boldsymbol{\theta}\,;\mathcal{B}_r) = \mathcal{J}\,(\boldsymbol{\theta}\,;\mathcal{B}_r) + 2(1 - \frac{t-1}{T})\lambda_{\text{reg}}\,\|\boldsymbol{\theta}\|_1\,.$$

This follows the linearly decaying regularization schedule proposed in [25].

### F.1.7 Scrub

The Scrub loss decomposes into different terms depending on the epoch. Let $\pi_{\boldsymbol{\theta}}(\boldsymbol{y}\mid\boldsymbol{x})$ denote the model's predicted distribution over classes $\boldsymbol{y}$ for input $\boldsymbol{x}$ for parameter vector $\boldsymbol{\theta}$, and define $\text{KL}(\cdot\,\|\,\cdot)$ as the Kullback-Leiber divergence. Recall $\boldsymbol{\theta}^*$ denotes the initial trained model parameters, and denote the current epoch $t \in \{0, \ldots, T-1\}$. Then the Scrub loss $\mathcal{J}_{\text{Scrub}}(\boldsymbol{\theta}\,;\mathcal{B}_r, \mathcal{B}_f, \lambda_{\text{reg}}, \lambda_{\text{GA}}, t)$ is defined as:

$$\mathcal{J}_{\text{Scrub}}(\boldsymbol{\theta}\,;\mathcal{B}_r, \mathcal{B}_f, \lambda_{\text{reg}}, \lambda_{\text{GA}}, t) =$$
$$\begin{cases} \mathcal{J}\,(\boldsymbol{\theta}\,;\mathcal{B}_r) + \frac{\lambda_{\text{reg}}}{|\mathcal{B}_r|}\displaystyle\sum_{(\boldsymbol{x}_r, y_r)\in\mathcal{B}_r} \text{KL}(\pi_{\boldsymbol{\theta}^*}(\boldsymbol{y}\mid\boldsymbol{x}_r)\,\|\,\pi_{\boldsymbol{\theta}}(\boldsymbol{y}\mid\boldsymbol{x}_r)) & \text{if } t \text{ even or } t \geq T - T_{\text{GD}} \\ -\frac{\lambda_{\text{GA}}}{|\mathcal{B}_f|}\displaystyle\sum_{(\boldsymbol{x}_f, y_f)\in\mathcal{B}_f} \text{KL}(\pi_{\boldsymbol{\theta}^*}(\boldsymbol{y}\mid\boldsymbol{x}_f)\,\|\,\pi_{\boldsymbol{\theta}}(\boldsymbol{y}\mid\boldsymbol{x}_f)) & \text{otherwise} \end{cases}$$

### F.1.8   NPO

Recall that $\boldsymbol{\theta}^*$ denotes the initial trained model parameters. Then, the NPO loss is

$$\mathcal{J}_{\text{NPO}}\left(\boldsymbol{\theta}\,;\mathcal{B}_f,\lambda_{\text{GA}}\right) = \frac{1}{|\mathcal{B}|}\sum_{(\boldsymbol{x}_f,y_f)\in\mathcal{B}_f}\frac{2}{\lambda_{\text{GA}}}\log\left(1 + \frac{\pi_{\boldsymbol{\theta}}(y_f\mid\boldsymbol{x}_f)}{\pi_{\boldsymbol{\theta}^*}(y_f\mid\boldsymbol{x}_f)}\right)^{\lambda_{\text{GA}}},$$

where $\pi_{\boldsymbol{\theta}}(y_f\mid\boldsymbol{x}_f)$ denotes the model's predicted probability of class $y_f$ for input $\boldsymbol{x}_f$ for parameter vector $\boldsymbol{\theta}$. Note that this is equivalent to setting the parameter $\beta$ in [18] to $\lambda_{\text{reg}}$.

### F.1.9   SalUn

Before performing any unlearning, SalUn computes the median of the absolute values of the elements of the model gradient with respect to the loss over the forget set at the original parameter vector $\boldsymbol{\theta}^*$. Formally, define
$$\boldsymbol{g}_f = \text{abs}(\nabla_{\boldsymbol{\theta}}\mathcal{J}\left(\boldsymbol{\theta}^*\,;\mathcal{D}_f\right)),$$
where $\text{abs}(\cdot)$ denotes the element-wise absolute value. This step requires an initial pass over the forget set which we do not count toward the number of unlearning epochs $T$.

The "saliency mask" $\boldsymbol{m}_S$ is then defined as the binary vector that selects parameters whose corresponding gradient magnitudes exceed the median of $\boldsymbol{g}_f$, denoted $\text{median}(\boldsymbol{g}_f)$:

$$\boldsymbol{m}_S = \mathbb{1}\{\boldsymbol{g}_f > \text{median}(\boldsymbol{g}_f)\}.$$

After computing $\boldsymbol{m}_S$, SalUn minimizes the loss

$$\mathcal{J}_{\text{SalUn}}(\boldsymbol{\theta}\,;\mathcal{B}_r,\mathcal{B}_f) = \mathcal{J}\left(\boldsymbol{\theta}\,;\mathcal{B}_r\right) + \lambda_{\text{GA}}\mathcal{J}\left(\boldsymbol{\theta}\,;\mathcal{B}_f'\right),$$

where $\mathcal{B}_f'$ is a modified version of the forget set batch $\mathcal{B}_f$ in which each true label is replaced with a random incorrect label. During unlearning, SALUN treats only the parameters where $\boldsymbol{m}_S = 1$ as trainable and freezes the others. Thus, the gradient update is applied only to $\boldsymbol{m}_S \odot \boldsymbol{\theta}$.

### F.1.10   MinNorm-OG

For each batch $\mathcal{B}_r$, we always perform a loss descent step:

$$\mathcal{J}_{\text{MinNorm-OG}}(\boldsymbol{\theta}\,;\mathcal{B}_r) = \mathcal{J}\left(\boldsymbol{\theta}\,;\mathcal{B}_r\right)$$

Following the AdamW update for this loss, we then (depending on the epoch) perform the model update corresponding to solving the relaxed unlearning problem (7) for $\tilde{R}(\boldsymbol{\theta}+\boldsymbol{\Delta}) = \|\boldsymbol{\theta}+\boldsymbol{\Delta}\|_2^2 + \lambda\|\boldsymbol{\Delta}\|_2^2$, where $\lambda$ is a saved parameter of the algorithm. We use the parameters $T_{\text{Proj}}$ and $T_{\text{GD}}$ to determine which epochs to perform the unlearning update. For the $T_{\text{GD}}$ last epochs, we only perform the descent step and skip the unlearning update, similar to Scrub. In the first $T - T_{\text{GD}}$ epochs, we perform the unlearning update every $T_{\text{Proj}}$ epochs.

We initialize $\lambda = \frac{1}{\lambda_{\text{reg}}} - 1$, and each time we perform the unlearning update, we grow the value of $\lambda$ through the update $\lambda \leftarrow \frac{\lambda+1}{\gamma_{\text{reg}}} - 1$ using the decay factor $\gamma_{\text{reg}} \in [0,1]$. For our algorithm we only use values of $\lambda_{\text{reg}}$ such that $\lambda_{\text{reg}} \leq 1$. The update for $\lambda$ leads to solutions to the relaxed unlearning problem which result in smaller, more conservative perturbations.

To interpret these values, first recall that we solve the relaxed unlearning problem over a subsample of each batch $\mathcal{B}_r' \subseteq \mathcal{B}_r$ where $|\mathcal{B}_r'| = n_{\text{pert}}$. For convenience, define the gradient subspace $\mathcal{G}_r' = \text{span}\{\nabla_{\boldsymbol{\theta}}f(\boldsymbol{\theta},\boldsymbol{x})\}_{(\boldsymbol{x},\boldsymbol{y})\in\mathcal{B}_r'}$. As we showed in Appendix E, for any value of $\lambda$, the optimal perturbation is then $\tilde{\boldsymbol{\Delta}} = -\frac{1}{1+\lambda}\mathcal{P}_{\mathcal{G}_r'^{\perp}}(\boldsymbol{\theta})$. Thus, the initial value $\lambda = \frac{1}{\lambda_{\text{reg}}} - 1$ leads to the perturbation $\tilde{\boldsymbol{\Delta}} = -\lambda_{\text{reg}}\mathcal{P}_{\mathcal{G}_r'^{\perp}}(\boldsymbol{\theta})$. Further, the coefficient update $\lambda = \frac{\lambda'+1}{\gamma_{\text{reg}}} - 1$ leads to a more conservative unlearning perturbation $\tilde{\boldsymbol{\Delta}} = -\gamma_{\text{reg}}\frac{1}{1+\lambda'}\mathcal{P}_{\mathcal{G}_r'^{\perp}}(\boldsymbol{\theta})$, as it is down-weighted by $\gamma_{\text{reg}}$. Thus, $\lambda_{\text{reg}}$ is the initial strength of the perturbation and $\gamma_{\text{reg}}$ represents a multiplicative decay of this strength through each update to $\lambda$.

Table 5: Data Poisoning experiment results showing the median sup-norm distance between the retain set trend $y = \sin(x)$ and the unlearned model outputs across 10 trials (smaller is better). The parentheses indicate the central range of values over the 10 trials.

| Epochs | Retrain | MinNorm-OG | GD | GA | NGP | NGD | Ridge | $\ell_1$-Sparse |
|---|---|---|---|---|---|---|---|---|
| 10 | **1.50** (1.34, 1.85) | **1.50** (1.45, 2.96) | 3.23 (2.52, 6.65) | 2.56 (2.19, 3.45) | 2.50 (2.20, 3.94) | 2.73 (2.26, 7.23) | 2.27 (1.90, 3.02) | 3.28 (2.52, 3.62) |
| 100 | 1.36 (1.27, 1.44) | **1.08** (1.00, 1.63) | 2.76 (2.48, 3.54) | 23.8 (18.1, 30.0) | 2.62 (2.31, 7.87) | 2.85 (2.47, 3.57) | 2.13 (1.75, 3.12) | 1.79 (1.54, 2.48) |
| 1000 | 1.17 (1.08, 1.26) | **0.63** (0.42, 0.96) | 2.61 (2.39, 3.38) | 1400 (1025, 1869) | 2.75 (2.07, 9.80) | 2.45 (1.96, 3.67) | 1.96 (1.61, 3.84) | 1.37 (1.19, 1.55) |

We formally write the unlearning update at epoch $t$ as follows, where $\boldsymbol{\theta}_0$ is the current parameter vector, $\boldsymbol{\theta}_{\text{new}}$ is the updated vector, and $\mathrm{mod}$ denotes the modulo operation.

**if** $t \bmod T_{\text{Proj}} \neq 0$ or $t \geq T - T_{\text{GD}}$
$$\boldsymbol{\theta}_{\text{new}} = \boldsymbol{\theta}_0$$
**else**
$$\tilde{\boldsymbol{\Delta}} = \operatorname*{argmin}_{\boldsymbol{\Delta} \in \mathcal{G}_r'^{\perp}} \|\boldsymbol{\theta}_0 + \boldsymbol{\Delta}\|_2^2 + \lambda \|\boldsymbol{\Delta}\|_2^2$$
$$\boldsymbol{\theta}_{\text{new}} = \boldsymbol{\theta}_0 + \tilde{\boldsymbol{\Delta}}$$
$$\lambda \leftarrow \frac{\lambda + 1}{\gamma_{\text{reg}}} - 1$$

**Gradients for Classification.** We make a special note of how we compute the gradient subspace $\mathcal{G}_r'$ for classification tasks. At the parameter value $\boldsymbol{\theta}_0$, the model prediction is $f(\boldsymbol{\theta}_0, \boldsymbol{x}) = \operatorname{argmax} \boldsymbol{z}_{\boldsymbol{\theta}_0}(\boldsymbol{y} \mid \boldsymbol{x})$ where $\boldsymbol{z}_{\boldsymbol{\theta}}(\boldsymbol{y} \mid \boldsymbol{x})$ denotes the model's unnormalized logits over the classes $\boldsymbol{y}$ for input $\boldsymbol{x}$ for parameter vector $\boldsymbol{\theta}$. This is not a continuous function of $\boldsymbol{\theta}$, so we cannot compute its gradient directly. However, following prior works [19], we use the gradient $\nabla_{\boldsymbol{\theta}} \left( \boldsymbol{z}_{\boldsymbol{\theta}_0}(\boldsymbol{y} \mid \boldsymbol{x}) \right)_j$, where $j = f(\boldsymbol{\theta}_0, \boldsymbol{x})$ is the model's predicted class for input $\boldsymbol{x}$. In other words, we take the gradient of the the unnormalized logits at the index of the maximum value, where we do not treat the index as a function of $\boldsymbol{\theta}$.

### F.2 Data Poisoning

We train a 3-layer multilayer perceptron with a hidden dimension of 300 using the sigmoid linear unit (SiLU) activation function. For each seed, we randomly sample 50 retain set points $(x_r, y_r) \in \mathcal{D}_r$ with $y_r = \sin(x_r)$ and 5 forget set points $(x_f, y_f) \in \mathcal{D}_f$ with $y_f = 1.5$, over the input domain $\mathcal{X} = [-5\pi, 5\pi] \subseteq \mathbb{R}$. We initially train the poisoned model on all the samples using the AdamW optimizer with a learning rate of $10^{-3}$ over 100,000 epochs.

Given these poisoned models, we apply each of the unlearning algorithms over a sweep of hyperparameters and evaluate the output $\boldsymbol{\theta}$ of each unlearning method by measuring the deviation from the retain set trend, given by $\sup_{\boldsymbol{x} \in \mathcal{X}} |f(\boldsymbol{\theta}, \boldsymbol{x}) - \sin(\boldsymbol{x})|$. We fix the number of epochs for each algorithm and allow full data access, so each method has access to all of $\mathcal{D}_r$ during unlearning. We repeat the entire process over 10 trials. For the number of unlearning epochs $T \in \{10, 100, 1000\}$, we report the best performance of each algorithm in Table 5 along with the central range of each method's performance over the 10 trials, discarding the two best and worst trials. We report the best hyperparameters for each method in Table 6, and the corresponding search spaces in Table 7. We also include visualizations of the recovered models from each unlearning method in Figures 3, 4, and 5. All experiments were run on either a single NVIDIA A40 GPU or a single NVIDIA H200 GPU.

Table 6: Hyperparameter settings for each entry in Table 5. Blank entries indicate that the hyperparameter is not applicable to the corresponding method.

| Epochs | Method | $\eta$ | $\lambda_{\mathrm{GA}}$ | $\lambda_{\mathrm{reg}}$ | $\sigma$ | $T_{\mathrm{GD}}$ | $\gamma_{\mathrm{reg}}$ | $T_{\mathrm{Proj}}$ | $n_{\mathrm{pert}}$ |
|---|---|---|---|---|---|---|---|---|---|
| 10 | Retrain | 1e-4 | | | | | | | |
| | MinNorm-OG | 1e-3 | | | .1 | 2 | .9 | 1 | 50 |
| | GD | 1e-4 | | | | | | | |
| | GA | 1e-4 | | | | | | | |
| | NGP | 1e-4 | 1.0 | | | | | | |
| | NGD | 1e-2 | | | .5 | | | | |
| | Ridge | 1e-2 | | 3.0 | | | .6 | | |
| | $\ell_1$-Sparse | 1e-2 | | .1 | | | | | |
| 100 | Retrain | 1e-4 | | | | | | | |
| | MinNorm-OG | 1e-3 | | | .1 | 50 | .9 | 2 | 50 |
| | GD | 1e-3 | | | | | | | |
| | GA | 1e-4 | | | | | | | |
| | NGP | 5e-4 | .01 | | | | | | |
| | NGD | 1e-3 | | | .1 | | | | |
| | Ridge | 1e-3 | | 3.0 | | | .9 | | |
| | $\ell_1$-Sparse | 1e-3 | | .1 | | | | | |
| 1000 | Retrain | 1e-4 | | | | | | | |
| | MinNorm-OG | 1e-3 | | | .1 | 500 | .9 | 10 | 50 |
| | GD | 1e-3 | | | | | | | |
| | GA | 1e-4 | | | | | | | |
| | NGP | 1e-3 | .001 | | | | | | |
| | NGD | 1e-3 | | | 1.0 | | | | |
| | Ridge | 1e-3 | | 3.0 | | | .9 | | |
| | $\ell_1$-Sparse | 1e-3 | | .1 | | | | | |

Table 7: Hyperparameter values tested in the experiments corresponding to Table 5.

| Epochs | $\eta$ | $\sigma$ | $\lambda_{\mathrm{reg}}$ | $\gamma_{\mathrm{reg}}$ | $\lambda_{\mathrm{GA}}$ | $T_{\mathrm{GD}}$ | $T_{\mathrm{Proj}}$ | $n_{\mathrm{pert}}$ |
|---|---|---|---|---|---|---|---|---|
| 10 | {1e-2, 1e-3, 1e-4} | {0.1, 0.5, 1.0} | {0.1, 1.0, 3.0} | {0.3, 0.6, 0.9} | {0.001, 0.01, 0.1, 1.0} | {1, 2, 5} | {1, 2, 5} | {50} |
| 100 | {1e-3, 5e-4, 1e-4} | {0.1, 0.5, 1.0} | {0.1, 1.0, 3.0} | {0.3, 0.6, 0.9} | {0.001, 0.01, 0.1, 1.0} | {25, 50} | {1, 2} | {50} |
| 1000 | {1e-3, 5e-4, 1e-4} | {0.1, 0.5, 1.0} | {0.1, 1.0, 3.0} | {0.3, 0.6, 0.9} | {0.001, 0.01, 0.1, 1.0} | {100, 200, 500} | {10, 50, 100} | {50} |

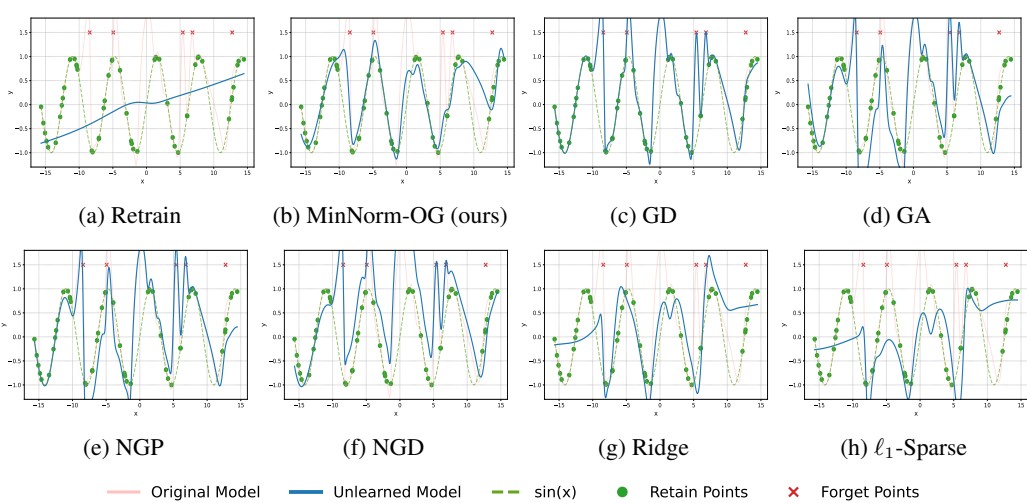

(a) Retrain    (b) MinNorm-OG (ours)    (c) GD    (d) GA

(e) NGP    (f) NGD    (g) Ridge    (h) $\ell_1$-Sparse

—— Original Model    —— Unlearned Model    - - sin(x)    ● Retain Points    ✕ Forget Points

Figure 3: Example unlearned model fits when given 10 unlearning epochs for the Data Poisoning experiment, where the forget points distort the retain set trend $y = \sin(x)$.

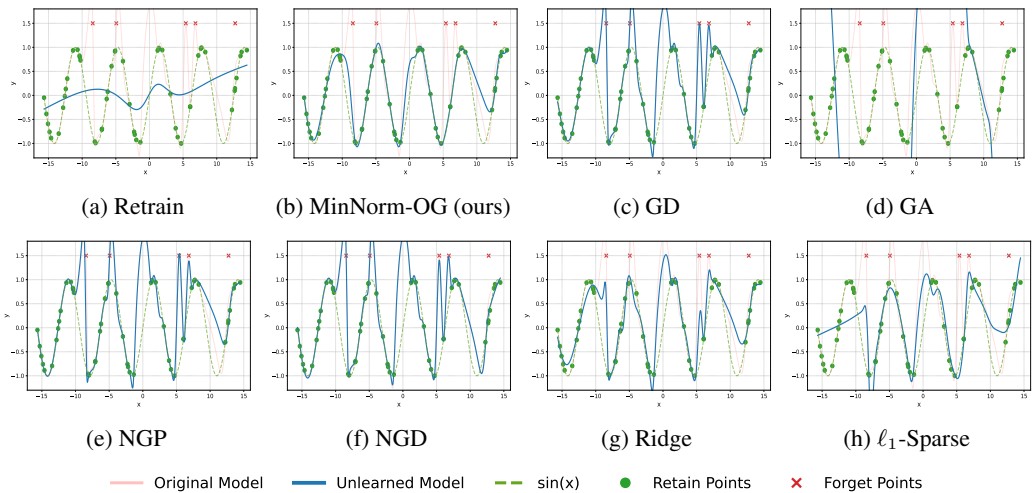

Figure 4: Example unlearned model fits when given 100 unlearning epochs for the Data Poisoning experiment, where the forget points distort the retain set trend $y = \sin(x)$.

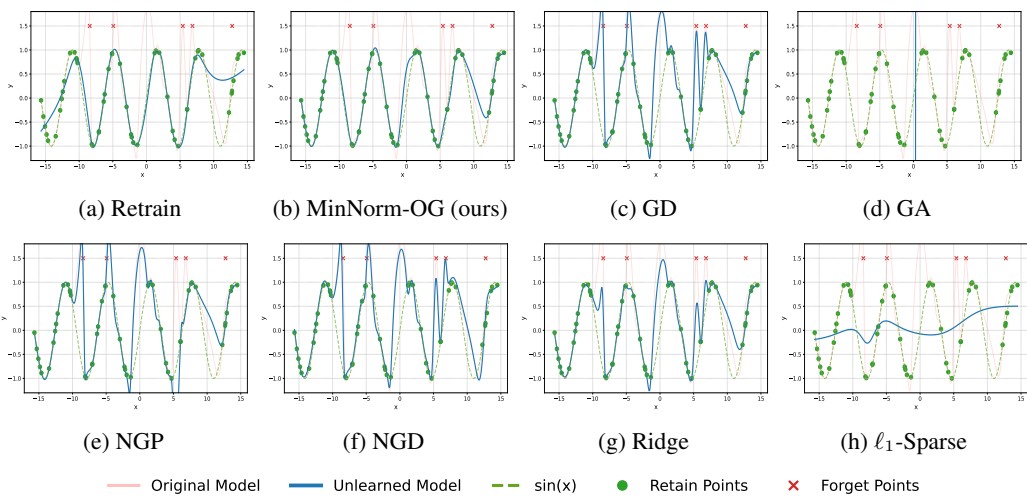

Figure 5: Example unlearned model fits when given 1000 unlearning epochs for the Data Poisoning experiment, where the forget points distort the retain set trend $y = \sin(x)$.

### F.3 Multi-Class Label Erasure

We use the CIFAR-10 [30] and Tiny ImageNet [31] datasets, creating red, green, and gray copies of each image in the training sets. The retain set consists of all gray images, while the forget set is formed by randomly sampling a proportion $p_{\text{color}} \in [0, 1]$ of the red and green copies. We then train modified ResNet-18 and ResNet-50 models [32] on CIFAR-10 and Tiny ImageNet, respectively, to jointly predict image class and color. Each model includes two separate prediction heads, one for image class and one for color.

For CIFAR-10, we train for 100 epochs using the SGD optimizer with an initial learning rate of $3 \times 10^{-2}$, weight decay of $5 \times 10^{-4}$, momentum of 0.9, and batch size of 256. The learning rate is reduced to $3 \times 10^{-3}$ at epoch 50. The ground-truth unlearned model is trained on the gray images alone using the same parameters.

For Tiny ImageNet, we initialize the ResNet-50 architecture with ImageNet-pretrained weights [33] from the `torchvision` library. We apply standard data augmentations to reduce overfitting and train for 100 epochs using a batch size of 512, initial learning rate of 0.1, weight decay of $10^{-4}$, and momentum of 0.9. The learning rate is decayed by a factor of 0.3 every 25 epochs. During the first 10 epochs, we update the model only using the class prediction loss to adapt the pretrained weights to the colored-image domain. For the remaining epochs, we optimize both the class and color prediction losses. The ground-truth unlearned model is trained on the gray images alone using the same parameters.

We then apply each of the unlearning algorithms over different constraints on the number of unlearning epochs and the amount of available retain data. We define $p_{\text{retain}} \in [0, 1]$ as the proportion of $\mathcal{D}_r$ available during unlearning. For each of the 5 trials, we train a new initial model and sample $p_{\text{retain}}$ proportion of $\mathcal{D}_r$ to serve as the available retain data. During each unlearning epoch, the algorithms iterate over batches from the forget set. For every forget set batch, a corresponding batch of the same size is sampled from the available retained data. The epoch ends once all forget set batches have been processed, regardless of whether there are unused retain set samples remaining. Any unused retain batches are not discarded—they will be sampled in subsequent epochs. Once all available retain set batches have been used at least once, the sampling process begins again from the start of the available retain set samples.

The ground truth unlearned model is only trained on gray samples, so it achieves strong accuracy on gray-colored inputs and always predicts the input image to be gray, no matter the input image color. We thus measure retain quality as accuracy on gray-colored test samples, and forget quality error by the mean squared error between the predicted gray probability and the ideal value of 1 across all colored inputs. For each method, we sweep hyperparameters and plot the corresponding Pareto frontier across the two metrics, where the optimal point at $(1, 0)$ which indicates perfect retain quality and zero forget quality error. Each point in the frontier for a given method represents the mean results over 5 trials of a single hyperparameter combination, with error bars representing the standard error for each metric. We label the performance of the ground truth unlearned model as GT. All training and parameter searches were performed on a cluster of NVIDIA GH200 GPUs.

Table 8: Hyperparameter values tested for the results in Figure 6 running the Multi-Label Class Erasure experiment on CIFAR-10 with $p_{\text{color}} = .01$, $p_{\text{retain}} = .01$ and $T = 2$.

| Hyperparameter | Sweep Values |
|---|---|
| $\eta$ | $\{10^{-3}, 10^{-4}, 10^{-5}, 10^{-6}\}$ |
| $\lambda_{\text{GA}}$ | $\{10^{-3}, 10^{-2}, 10^{-1}\}$ |
| $\lambda_{\text{reg}}$ | $\{10^{-3}, 10^{-2}, 5 \times 10^{-2}, 10^{-1}\}$ |
| $\sigma$ | $\{0.1, 1.0, 10.0\}$ |
| $T_{\text{GD}}$ | $\{0\}$ |
| $\gamma_{\text{reg}}$ | $\{0.2, 0.4, 0.8\}$ |
| $T_{\text{Proj}}$ | $\{1\}$ |
| $n_{\text{pert}}$ | $\{50\}$ |

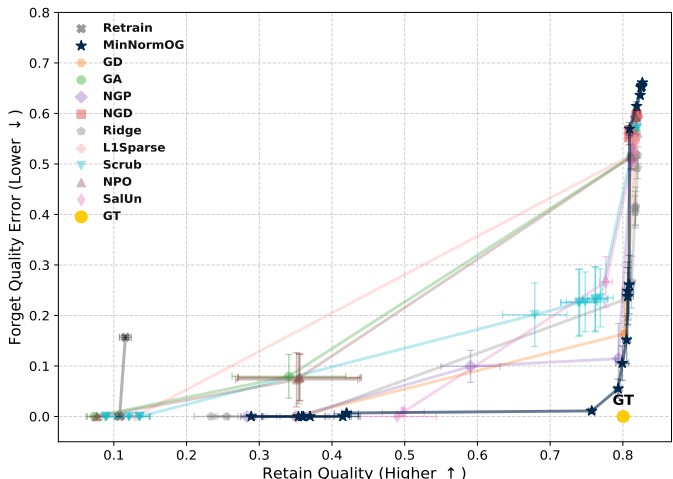

Figure 6: Pareto frontiers for each method across hyperparameter settings in the Multi-Class Label Erasure task on CIFAR-10 with $p_{\text{color}} = .01$, $p_{\text{retain}} = .01$ and $T = 2$. This is an enlarged version of the left subfigure in Figure 2 with added error bars.

Table 9: Hyperparameter values tested for the results in Figure 7 running the Multi-Label Class Erasure experiment on Tiny ImageNet with $p_{\text{color}} = .01$, $p_{\text{retain}} = .01$ and $T = 5$.

| Hyperparameter | Sweep Values |
| --- | --- |
| $\eta$ | $\{5 \times 10^{-4}, 10^{-4}, 10^{-5}, 10^{-6}\}$ |
| $\lambda_{\text{GA}}$ | $\{10^{-3}, 10^{-2}, 10^{-1}\}$ |
| $\lambda_{\text{reg}}$ | $\{10^{-3}, 10^{-2}, 5 \times 10^{-2}, 10^{-1}\}$ |
| $\sigma$ | $\{0.1, 1.0, 10.0\}$ |
| $T_{\text{GD}}$ | $\{2, 3\}$ |
| $\gamma_{\text{reg}}$ | $\{0.2, 0.4, 0.8, 0.9\}$ |
| $T_{\text{Proj}}$ | $\{1\}$ |
| $n_{\text{pert}}$ | $\{50\}$ |

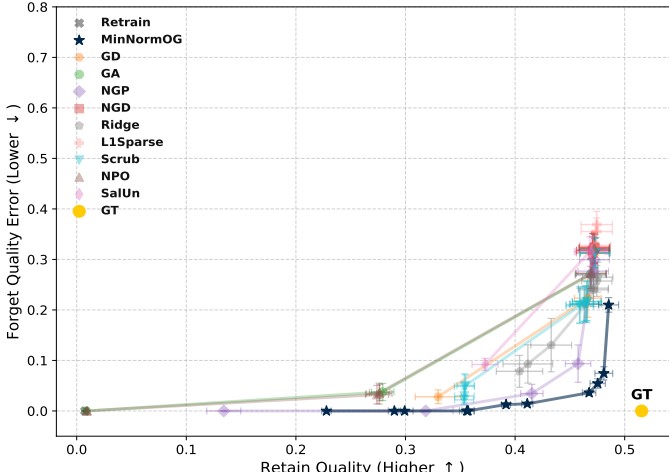

Figure 7: Pareto frontiers for each method across hyperparameter settings in the Multi-Class Label Erasure task on Tiny ImageNet with $p_{\text{color}} = .01$, $p_{\text{retain}} = .01$ and $T = 5$. This is an enlarged version of the right subfigure in Figure 2 with added error bars.

Table 10: Hyperparameter values tested for the results in Figure 8 running the Multi-Label Class Erasure experiment on CIFAR-10 with $p_{\text{color}} = .01$, $p_{\text{retain}} = .01$ and $T = 1$.

| Hyperparameter | Sweep Values |
|---|---|
| $\eta$ | $\{10^{-3}, 10^{-4}, 10^{-5}, 10^{-6}\}$ |
| $\lambda_{\text{GA}}$ | $\{10^{-3}, 10^{-2}, 10^{-1}\}$ |
| $\lambda_{\text{reg}}$ | $\{10^{-3}, 10^{-2}, 5 \times 10^{-2}, 10^{-1}\}$ |
| $\sigma$ | $\{0.1, 1.0, 10.0\}$ |
| $T_{\text{GD}}$ | $\{0\}$ |
| $\gamma_{\text{reg}}$ | $\{0.2, 0.4, 0.8\}$ |
| $T_{\text{Proj}}$ | $\{1\}$ |
| $n_{\text{pert}}$ | $\{50\}$ |

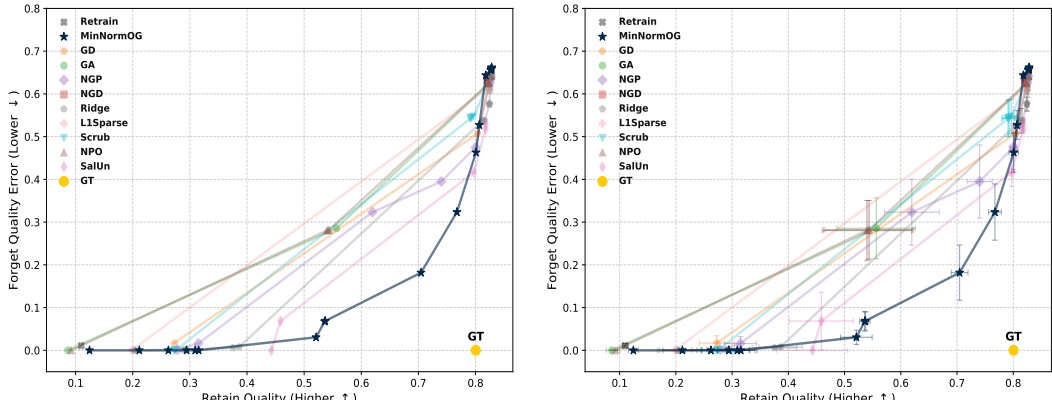

Figure 8: Pareto frontiers for each method across hyperparameter settings in the Multi-Class Label Erasure task on CIFAR-10 with $p_{\text{color}} = .01$, $p_{\text{retain}} = .01$ and $T = 1$. The left panel omits error bars for visual clarity.

Table 11: Hyperparameter values tested for the results in Figure 9 running the Multi-Label Class Erasure experiment on CIFAR-10 with $p_{\text{color}} = .01$, $p_{\text{retain}} = .001$ and $T = 2$.

| Hyperparameter | Sweep Values |
|---|---|
| $\eta$ | $\{10^{-3}, 10^{-4}, 10^{-5}, 10^{-6}\}$ |
| $\lambda_{\text{GA}}$ | $\{10^{-3}, 10^{-2}, 10^{-1}\}$ |
| $\lambda_{\text{reg}}$ | $\{10^{-3}, 10^{-2}, 5 \times 10^{-2}, 10^{-1}\}$ |
| $\sigma$ | $\{0.1, 1.0, 10.0\}$ |
| $T_{\text{GD}}$ | $\{0\}$ |
| $\gamma_{\text{reg}}$ | $\{0.2, 0.4, 0.8\}$ |
| $T_{\text{Proj}}$ | $\{1\}$ |
| $n_{\text{pert}}$ | $\{50\}$ |

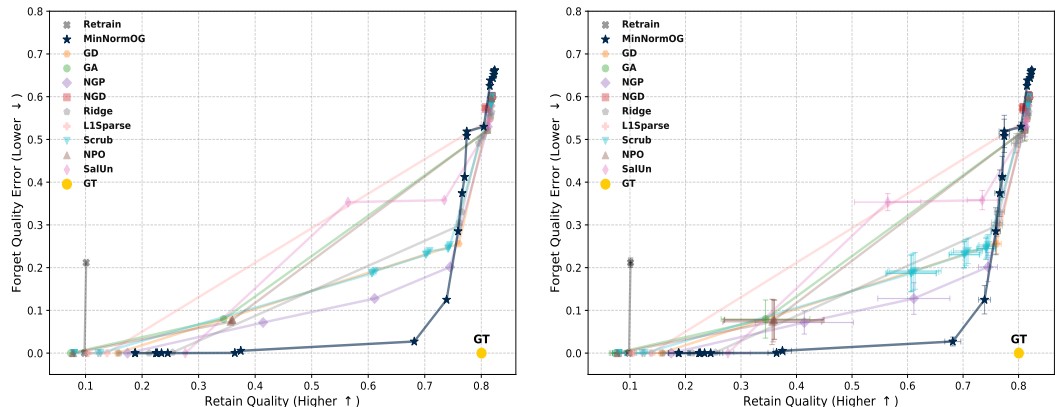

Figure 9: Pareto frontiers for each method across hyperparameter settings in the Multi-Class Label Erasure task on CIFAR-10 with $p_{\text{color}} = .01$, $p_{\text{retain}} = .001$ and $T = 2$. The left panel omits error bars for visual clarity.

Table 12: Hyperparameter values tested for the results in Figure 10 running the Multi-Label Class Erasure experiment on CIFAR-10 with $p_{\text{color}} = .01$, $p_{\text{retain}} = .001$ and $T = 5$.

| Hyperparameter | Sweep Values |
|---|---|
| $\eta$ | $\{10^{-3}, 5 \times 10^{-4}, 10^{-4}, 5 \times 10^{-5}, 10^{-5}, 10^{-6}\}$ |
| $\lambda_{\text{GA}}$ | $\{10^{-3}, 10^{-2}, 10^{-1}\}$ |
| $\lambda_{\text{reg}}$ | $\{10^{-3}, 10^{-2}, 5 \times 10^{-2}, 10^{-1}\}$ |
| $\sigma$ | $\{0.1, 1.0, 10.0\}$ |
| $T_{\text{GD}}$ | $\{1, 2, 3\}$ |
| $\gamma_{\text{reg}}$ | $\{0.2, 0.4, 0.8\}$ |
| $T_{\text{Proj}}$ | $\{1\}$ |
| $n_{\text{pert}}$ | $\{50\}$ |

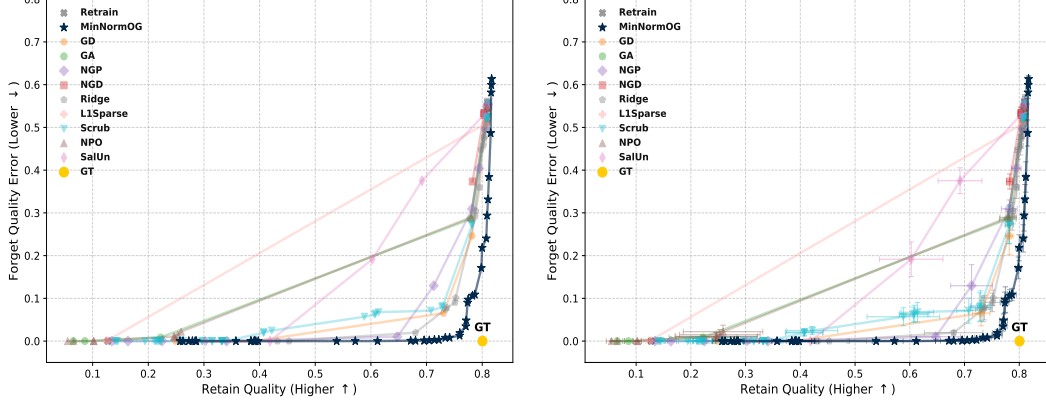

Figure 10: Pareto frontiers for each method across hyperparameter settings in the Multi-Class Label Erasure task on CIFAR-10 with $p_{\text{color}} = .01$, $p_{\text{retain}} = .001$ and $T = 5$. The left panel omits error bars for visual clarity.

Table 13: Hyperparameter values tested for the results in Figure 11 running the Multi-Label Class Erasure experiment on CIFAR-10 with $p_{\text{color}} = .001$, $p_{\text{retain}} = .001$ and $T = 10$.

| Hyperparameter | Sweep Values |
|---|---|
| $\eta$ | $\{10^{-3}, 10^{-4}, 10^{-5}, 10^{-6}\}$ |
| $\lambda_{\text{GA}}$ | $\{10^{-3}, 10^{-2}, 10^{-1}\}$ |
| $\lambda_{\text{reg}}$ | $\{10^{-3}, 10^{-2}, 5 \times 10^{-2}, 10^{-1}\}$ |
| $\sigma$ | $\{0.1, 1.0, 10.0\}$ |
| $T_{\text{GD}}$ | $\{0, 1, 2, 5\}$ |
| $\gamma_{\text{reg}}$ | $\{0.2, 0.4, 0.8\}$ |
| $T_{\text{Proj}}$ | $\{1\}$ |
| $n_{\text{pert}}$ | $\{50\}$ |

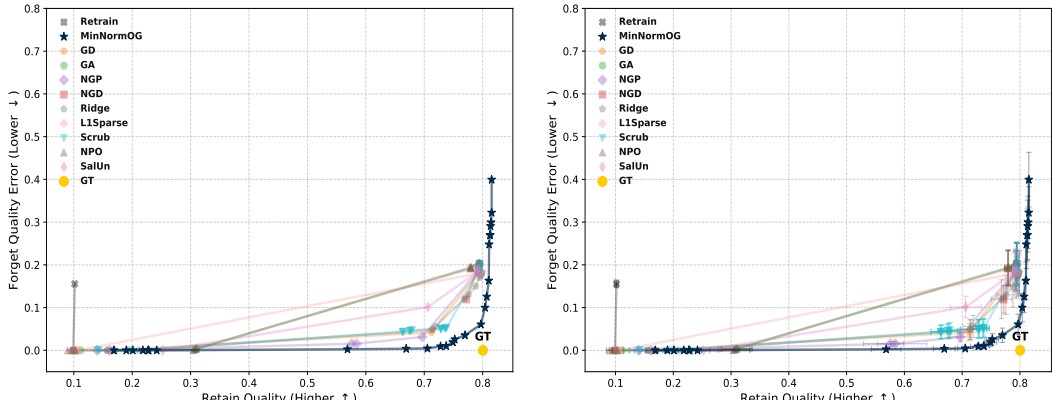

Figure 11: Pareto frontiers for each method across hyperparameter settings in the Multi-Class Label Erasure task on CIFAR-10 with $p_{\text{color}} = .001$, $p_{\text{retain}} = .001$, and $T = 10$. The left panel omits error bars for visual clarity.

Table 14: Hyperparameter values tested for the results in Figure 12 running the Multi-Label Class Erasure experiment on CIFAR-10 with $p_{\text{color}} = .001$, $p_{\text{retain}} = .01$ and $T = 10$.

| Hyperparameter | Sweep Values |
|---|---|
| $\eta$ | $\{10^{-3}, 10^{-4}, 10^{-5}, 10^{-6}\}$ |
| $\lambda_{\text{GA}}$ | $\{10^{-3}, 10^{-2}, 10^{-1}\}$ |
| $\lambda_{\text{reg}}$ | $\{10^{-3}, 10^{-2}, 5 \times 10^{-2}, 10^{-1}\}$ |
| $\sigma$ | $\{0.1, 1.0, 10.0\}$ |
| $T_{\text{GD}}$ | $\{0, 1, 2, 5\}$ |
| $\gamma_{\text{reg}}$ | $\{0.2, 0.4, 0.8\}$ |
| $T_{\text{Proj}}$ | $\{1\}$ |
| $n_{\text{pert}}$ | $\{50\}$ |

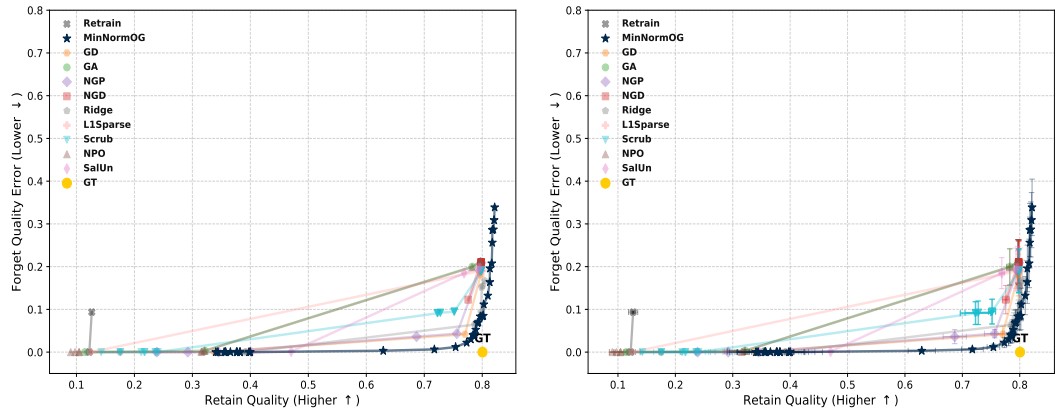

Figure 12: Pareto frontiers for each method across hyperparameter settings in the Multi-Class Label Erasure task on CIFAR-10 with $p_{\text{color}} = .001$, $p_{\text{retain}} = .01$, and $T = 10$. The left panel omits error bars for visual clarity.

Table 15: Hyperparameter values tested for the results in Figure 13 running the Multi-Label Class Erasure experiment on Tiny ImageNet with $p_{\text{color}} = .01$, $p_{\text{retain}} = .001$ and $T = 10$.

| Hyperparameter | Sweep Values |
|---|---|
| $\eta$ | $\{10^{-3}, 10^{-4}, 10^{-5}, 10^{-6}\}$ |
| $\lambda_{\text{GA}}$ | $\{10^{-3}, 10^{-2}, 10^{-1}\}$ |
| $\lambda_{\text{reg}}$ | $\{10^{-3}, 10^{-2}, 5 \times 10^{-2}, 10^{-1}\}$ |
| $\sigma$ | $\{0.1, 1.0, 10.0\}$ |
| $T_{\text{GD}}$ | $\{0, 1, 2\}$ |
| $\gamma_{\text{reg}}$ | $\{0.2, 0.4, 0.8, 0.9\}$ |
| $T_{\text{Proj}}$ | $\{1\}$ |
| $n_{\text{pert}}$ | $\{50\}$ |

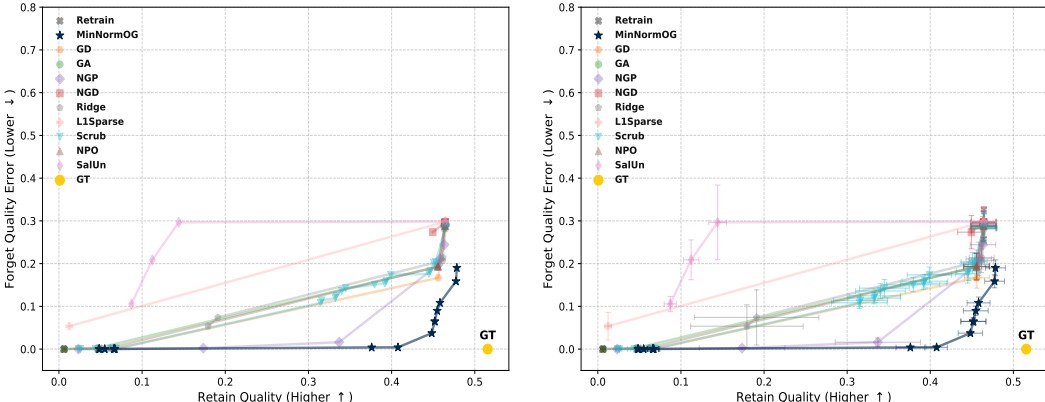

Figure 13: Pareto frontiers for each method across hyperparameter settings in the Multi-Class Label Erasure task on Tiny ImageNet with $p_{\text{color}} = .01$, $p_{\text{retain}} = .001$, and $T = 10$. The left panel omits error bars for visual clarity.

Table 16: Hyperparameter values tested for the results in Figure 14 running the Multi-Label Class Erasure experiment on Tiny ImageNet with $p_{\text{color}} = .01$, $p_{\text{retain}} = .01$ and $T = 10$.

| Hyperparameter | Sweep Values |
|---|---|
| $\eta$ | $\{10^{-3}, 10^{-4}, 10^{-5}\}$ |
| $\lambda_{\text{GA}}$ | $\{10^{-3}, 10^{-2}, 10^{-1}\}$ |
| $\lambda_{\text{reg}}$ | $\{10^{-3}, 10^{-2}, 5 \times 10^{-2}, 10^{-1}\}$ |
| $\sigma$ | $\{0.1, 1.0, 10.0\}$ |
| $T_{\text{GD}}$ | $\{0, 1, 2\}$ |
| $\gamma_{\text{reg}}$ | $\{0.2, 0.4, 0.8, 0.9\}$ |
| $T_{\text{Proj}}$ | $\{1\}$ |
| $n_{\text{pert}}$ | $\{50\}$ |

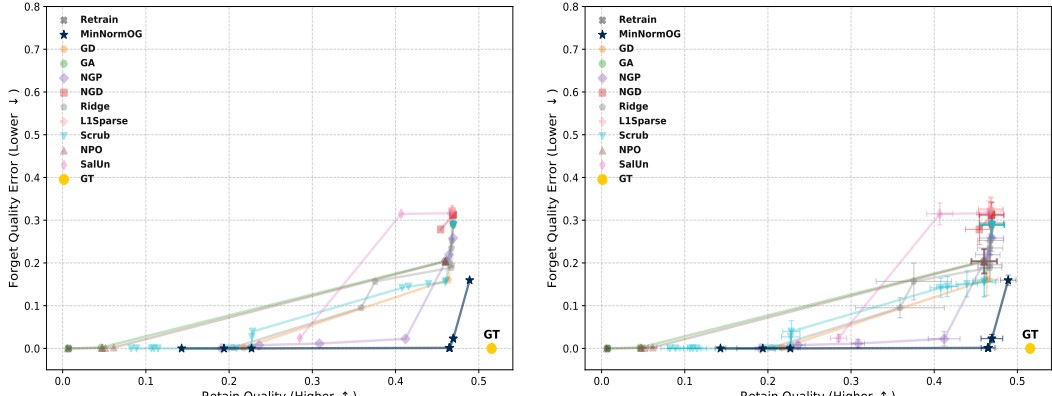

Figure 14: Pareto frontiers for each method across hyperparameter settings in the Multi-Class Label Erasure task on Tiny ImageNet with $p_{\text{color}} = .01$, $p_{\text{retain}} = .01$, and $T = 10$. The left panel omits error bars for visual clarity.

## F.4 Representation Collapse

We again use the CIFAR-10 dataset with a modified ResNet-18 architecture, assigning each of the 10 image classes a unique color. The retain set $\mathcal{D}_r$ consists of images colored according to their assigned class color, while the forget set $\mathcal{D}_f$ contains randomly colored images. The ground-truth unlearned model predicts class labels based solely on color, as color perfectly determines the label and is easier to learn than image content. In contrast, models trained on the full dataset $\mathcal{D} = \mathcal{D}_r \sqcup \mathcal{D}_f$ must rely on content features, since color is no longer fully predictive of the label. Thus, for unlearning evaluation, we label heldout test images by color and assess unlearning via color-label accuracy, testing if the unlearning methods can collapse the original model into just a color classifier.

To construct $\mathcal{D}_f$, we randomly select 1% of the training images per class and assign each a random color chosen from the remaining nine class colors. For the initial model, we first warm-start by training for 20 epochs on gray images (a color not used in the class assignments) to encourage learning general image structure. We then train for 50 epochs on the combined retain and forget sets, up-weighting the loss contribution of off-colored samples to promote fast color-invariant learning. We used the SGD optimizer with an initial learning rate of $3 \times 10^{-2}$, weight decay of $5 \times 10^{-4}$, momentum of $0.9$, a batch size of 256, and a multiplicative learning rate decay of $0.1$ every 20 epochs. The ground-truth unlearned model is trained for 50 epochs on the retain set using the same optimizer settings. All training was performed on a cluster of NVIDIA H200 GPUs.

We report results mean results over 5 trials in Table 17, which presents the same results as Table 2 along with standard errors for each entry. We test each method under different constraints on the number of unlearning epochs and percentage of accessible retain set samples. Specifically, for each random trial we randomly sample a proportion of $p_{\text{retain}} \in [0, 1]$ samples from $\mathcal{D}_r$ which is accessible to each unlearning method. The "Retain %" column Table 17 represents $100 \times p_{\text{retain}}$. Just as in the Multi-Class Label Erasure experiment, during each unlearning epoch the algorithms iterate over

Table 17: Representation Collapse experiment results across constraints on the number of unlearning epochs and percentage of accessible retain set samples. Models are trained on colored images where color perfectly predicts the label in the retain set but not in the full dataset $\mathcal{D}$. Reported values are mean test accuracies (%) on test images labeled by color (higher is better), averaged over 5 trials $\pm$ standard error.

| Retain % | Epochs | Retrain | MinNorm-OG | GD | GA | NGP | NGD | Ridge | $\ell_1$-Sparse | Scrub | NPO | SalUn |
|---|---|---|---|---|---|---|---|---|---|---|---|---|
| 0.1 | 5 | 79.1±4.5 | **49.1**±6.1 | 24.0±3.2 | 34.5±4.4 | 45.9±3.4 | 12.6±0.2 | 20.9±3.7 | 12.4±0.3 | 37.3±3.7 | 40.6±3.7 | 29.4±5.5 |
|  | 10 | 95.5±1.1 | **78.7**±3.4 | 55.2±4.3 | 38.3±5.4 | 73.7±3.8 | 33.0±2.2 | 60.0±3.7 | 23.0±6.5 | 61.9±3.4 | 43.2±6.9 | 53.4±7.9 |
|  | 15 | 96.3±0.8 | **93.2**±1.3 | 78.2±3.3 | 39.8±5.1 | 81.5±1.6 | 46.9±3.4 | 81.3±2.7 | 25.1±4.7 | 74.1±3.9 | 44.6±7.6 | 76.3±5.3 |
| 1 | 5 | 92.8±2.6 | **59.5**±6.1 | 40.7±2.4 | 34.2±4.7 | 58.3±6.0 | 32.9±3.6 | 33.5±6.1 | 12.5±0.2 | 47.8±4.9 | 42.5±4.1 | 42.5±2.7 |
|  | 10 | 98.5±0.1 | **94.7**±1.1 | 73.6±3.0 | 38.2±5.4 | 92.4±2.5 | 70.5±3.2 | 63.4±5.0 | 31.4±6.6 | 75.8±2.8 | 43.5±6.9 | 68.7±4.0 |

batches from the forget set and sample a corresponding batch of the same size from the available retained data. The epoch ends once all forget set batches have been processed, regardless of whether there are unused retain set samples remaining. Any unused retain batches are not discarded—they will be sampled in subsequent epochs. Once all available retain set batches have been used at least once, the sampling process begins again from the start of the available retain set samples.

Retrain achieves much higher color classification accuracy than any unlearning method, as image color is an easy feature to learn from scratch. This result is specific to this setup, as our prior experiments demonstrate that Retrain is not a viable unlearning strategy in general. For this reason, we report the Retrain results separately in the leftmost column and highlight the MinNorm-OG results in bold, as they represent the best performance among the unlearning methods in each row.

For each constraint setting on the number of epochs and the Retain %, we tested different hyperparameters before reporting the best performance for each method. Tables 18, 19, 20, 21, and 22 report the set of hyperparameter values we tested for each row of Table 17.

Table 18: Hyperparameter values considered for the Representation Collapse Experiment with $T = 5$ and $p_{\text{retain}} = 0.001$.

| Hyperparameter | Values |
|---|---|
| $\eta$ | $\{10^{-3},\ 5 \times 10^{-4},\ 10^{-4}\}$ |
| $\lambda_{\text{GA}}$ | $\{10^{-3},\ 10^{-2},\ 10^{-1}\}$ |
| $\lambda_{\text{reg}}$ | $\{0.1,\ 0.2,\ 0.5,\ 0.9\}$ |
| $\sigma$ | $\{0.1,\ 1.0,\ 10.0\}$ |
| $T_{\text{GD}}$ | $\{1,\ 2,\ 3\}$ |
| $\gamma_{\text{reg}}$ | $\{0.2,\ 0.4,\ 0.8\}$ |
| $T_{\text{Proj}}$ | $\{1\}$ |
| $n_{\text{pert}}$ | $\{50\}$ |

Table 19: Hyperparameter values considered for the Representation Collapse Experiment with $T = 10$ and $p_{\text{retain}} = 0.001$.

| Hyperparameter | Values |
|---|---|
| $\eta$ | $\{5 \times 10^{-3},\ 10^{-3},\ 5 \times 10^{-4},\ 10^{-4}\}$ |
| $\lambda_{\text{GA}}$ | $\{10^{-3},\ 10^{-2},\ 10^{-1}\}$ |
| $\lambda_{\text{reg}}$ | $\{0.01,\ 0.1,\ 0.5,\ 0.9\}$ |
| $\sigma$ | $\{0.1,\ 1.0,\ 10.0\}$ |
| $T_{\text{GD}}$ | $\{3,\ 5,\ 7\}$ |
| $\gamma_{\text{reg}}$ | $\{0.2,\ 0.4,\ 0.8\}$ |
| $T_{\text{Proj}}$ | $\{1\}$ |
| $n_{\text{pert}}$ | $\{50\}$ |

Table 20: Hyperparameter values considered for the Representation Collapse Experiment with $T = 15$ and $p_{\text{retain}} = 0.001$.

| Hyperparameter | Values |
|---|---|
| $\eta$ | $\{5 \times 10^{-3},\ 10^{-3},\ 5 \times 10^{-4},\ 10^{-4}\}$ |
| $\lambda_{\text{GA}}$ | $\{10^{-3},\ 10^{-2},\ 10^{-1}\}$ |
| $\lambda_{\text{reg}}$ | $\{0.01,\ 0.1,\ 0.5,\ 0.9\}$ |
| $\sigma$ | $\{0.1,\ 1.0,\ 10.0\}$ |
| $T_{\text{GD}}$ | $\{3,\ 5,\ 7\}$ |
| $\gamma_{\text{reg}}$ | $\{0.2,\ 0.4,\ 0.8\}$ |
| $T_{\text{Proj}}$ | $\{1\}$ |
| $n_{\text{pert}}$ | $\{50\}$ |

Table 21: Hyperparameter values considered for the Representation Collapse Experiment with $T = 5$ and $p_{\text{retain}} = 0.01$.

| Hyperparameter | Values |
|---|---|
| $\eta$ | $\{5 \times 10^{-3},\ 10^{-3},\ 5 \times 10^{-4},\ 10^{-4}\}$ |
| $\lambda_{\text{GA}}$ | $\{10^{-3},\ 10^{-2},\ 10^{-1}\}$ |
| $\lambda_{\text{reg}}$ | $\{0.01,\ 0.1,\ 0.5,\ 0.9\}$ |
| $\sigma$ | $\{0.1,\ 1.0,\ 10.0\}$ |
| $T_{\text{GD}}$ | $\{1,\ 2,\ 3\}$ |
| $\gamma_{\text{reg}}$ | $\{0.2,\ 0.4,\ 0.8\}$ |
| $T_{\text{Proj}}$ | $\{1\}$ |
| $n_{\text{pert}}$ | $\{50\}$ |

Table 22: Hyperparameter values considered for the Representation Collapse Experiment with $T = 10$ and $p_{\text{retain}} = 0.01$.

| Hyperparameter | Values |
|---|---|
| $\eta$ | $\{5 \times 10^{-3},\ 10^{-3},\ 5 \times 10^{-4},\ 10^{-4}\}$ |
| $\lambda_{\text{GA}}$ | $\{10^{-3},\ 10^{-2},\ 10^{-1}\}$ |
| $\lambda_{\text{reg}}$ | $\{0.01,\ 0.1,\ 0.5,\ 0.9\}$ |
| $\sigma$ | $\{0.1,\ 1.0,\ 10.0\}$ |
| $T_{\text{GD}}$ | $\{3,\ 5,\ 7\}$ |
| $\gamma_{\text{reg}}$ | $\{0.2,\ 0.4,\ 0.8\}$ |
| $T_{\text{Proj}}$ | $\{1\}$ |
| $n_{\text{pert}}$ | $\{50\}$ |

### F.5 Asset Information

We use the CIFAR-10 [30] and Tiny ImageNet [31] datasets in our experiments. Both datasets are publicly available but do not specify an explicit license. Additionally, we use the ResNet-18 and ResNet-50 [32] architectures and pretrained weights from PyTorch's `torchvision` library, which are licensed under the BSD 3-Clause License.

