# OpenReview forum: "Machine Unlearning under Overparameterization"
_NeurIPS.cc/2025/Conference — NeurIPS 2025 poster_

### Official Review · Reviewer_oxow · 2025-06-05

**Clarity:** 3
**Significance:** 3
**Originality:** 4
**Rating:** 5
**Confidence:** 3

**Summary:**

The authors present a forget set free unlearning method for overparameterized settings and provide guarantees for a relaxed problem formulation. The theoretical contributions are backed up with three new benchmarks.

**Questions:**

- Why did the authors leave out any established benchmarks/metrics? Additional results here would make me increase my score on this otherwise excellent paper

**Ethical Concerns:**

["NO or VERY MINOR ethics concerns only"]

**Final Justification:**

I remain with an accept score based on the points listed in strengths. The authors provided answers to all of my clarifying questions, which did not raise any additional significant concerns and explained why they left out certain aspects, such as MIA scores and explained why. Machine unlearning is still a new field and this work adds new nuggets of knowledge. Especially simple clearly interpretable experiments such as the ones shown in Fig. 1 are helpful in a field that still figures out how to actually measure unlearning performance (the MIA ULIRA etc struggle.)

**Limitations:**

yes

**Quality:**

3

**Strengths And Weaknesses:**

- Clarity and Quality: The paper is well written with clear notation and easy to interpret visualizations.
- Originality: The work is original with theoretical insights and method contributions.
- Significance: This work provides significant insights for future work to build upon.

*Strengths:*
- Strong theoretical insights/evaluations
- MinNorm-OG is intuitive, well explained, and the reasons for the design decisions are well motivated
- The new benchmarks are easy to interpret and clearly show shortcomings in existing methods

*Weaknesses:*
- Results on established metrics such as MIA for privacy are missing. The authors rightfully argue that established metrics can be opaque but knowing how this new method stacks up to established methods on this benchmark would be of interest. Overall, the authors did not benchmark on any established benchmarks with all three experimental benchmarks created for this paper. While I trust that the method is strong and will hold up, it would avoid any doubts of cherry picked benchmarks by including standard unlearning benchmarks too. If the authors prefer to stay away from MIA etc., other works such as Goel et al. “Corrective Unlearning” perform poison unlearning too with an objective measure of accuracy.
- The work only investigates shallow networks (acknowledged by the authors). Seeing how this method performs on larger models and different architectures would be of interest (e.g. transformers)

Nitpick: The color red in Figure 1 is hard to see and likely vanishes when printed.

---

> ### Author Rebuttal · Authors · 2025-07-31
>
> Dear Reviewer oxow, thank you for recognizing the originality, impact, and clarity of our work. We are happy to address your remaining questions below.
>
> **(Q1): Results on established metrics such as MIA for privacy are missing. The authors rightfully argue that established metrics can be opaque but knowing how this new method stacks up to established methods on this benchmark would be of interest. Overall, the authors did not benchmark on any established benchmarks with all three experimental benchmarks created for this paper. While I trust that the method is strong and will hold up, it would avoid any doubts of cherry picked benchmarks by including standard unlearning benchmarks too. If the authors prefer to stay away from MIA etc., other works such as Goel et al. “Corrective Unlearning” perform poison unlearning too with an objective measure of accuracy.**
>
> (A1) Thank you for raising this thoughtful concern about our evaluation choices.
>
> We did not use membership inference attacks (MIA) to evaluate unlearning performance because they are (i) general-purpose rather than task-specific, (ii) only provide guarantees relative to the sophistication of the attack method, and (iii) offer limited insight into the mechanism of forgetting. MIA results typically reduce to a confidence score indicating whether a model remembers a given sample, without revealing why unlearning failed or how it might be improved. Instead, we design task-specific evaluation metrics that relate to the structure of each unlearning experiment. For instance, in the Multi-Class Label Erasure setting, the forget and retain sets differ only in color: the forget set includes red and green digits while the retain set includes gray digits. This enables precise measurement of retain quality (classification accuracy on gray images) and forget quality (mean squared error between the model’s predicted probability of "gray" and the ideal of 1.0).
>
> We appreciate the reviewer highlighting Goel et al. (``Corrective Unlearning"), whose evaluation avoids MIA and proposes other meaningful objective metrics. Although their setting differs slightly, as they focus on adversarially manipulated forget sets and partial access to the manipulated samples, their Unlearning and Utility metrics closely align with our own. In their framework, Unlearning measures expected accuracy over the corrected, de-manipulated samples from the forget data domain, and Utility measures accuracy on samples from the retained data domain. In fact, we can reinterpret our metrics as analogs of these:
>
>
> - In the Data Poisoning experiment, the forget set consists of label-manipulated data (from $\sin(x)$ to the constant $1.5$.) We evaluate the sup-norm error relative to the true sin(x) function, which directly reflects both Unlearning and Utility, as the two metrics coincide since the input domain is shared.
> - In the Multi-Class Erasure setting, we can view the forget set as consisting of adversarially recolored images. Here, both the retain and forget sets are drawn from the same underlying domain of images, so the Unlearning and Utility metrics both correspond to the model's accuracy at predicting both content and color of gray test images. This is exactly our retain accuracy metric. However, this metric only reflects whether the predicted class is correct and does not compare the confidence of the model's predictions. To more closely evaluate whether the model has unlearned the color information associated with the forget set, we also compute our forget metric which measures the mean squared error between the model's predicted probability for the ``gray" class and the target probability of 1.0, as the ground truth unlearned model always predicts gray with full confidence. This metric can be interpreted as a softmax-level analog of the Unlearning metric, capturing not just whether the model predicts gray, but how confidently it does so, offering a more sensitive measure of unlearning effectiveness on the color prediction task.
> - In the Representation Collapse experiment, forget set labels are based on image content, while retain set labels depend solely on color. This can be seen as an adversarial manipulation where the forget labels are misaligned with the true color-based labels. Our metric measures the model’s accuracy on color-labeled test data, which directly reflects both Unlearning and Utility, since the input domain is shared and the true labels are color-based.
>
> Thus, while our benchmarks are novel, they remain deeply connected to established evaluation principles. We appreciate the reviewer’s suggestion and will revise the paper to cite this related work and explicitly draw these connections.
>
> **(Q2): The work only investigates shallow networks (acknowledged by the authors). Seeing how this method performs on larger models and different architectures would be of interest (e.g. transformers)**
>
> While our experiments focus on model and dataset sizes up to ResNet-18 on CIFAR-10, our primary goal is to present a set of interpretable experiments supported by unambiguous metrics. These metrics precisely quantify whether our proposed algorithm MinNorm-OG, developed as an extension of our theoretical framework, offers consistent improvements over existing baseline methods. We show clear failure cases for existing methods and demonstrate the stronger performance of MinNorm-OG, supporting the practical relevance of our framework and beyond the settings where we can provide formal guarantees.
>
> We agree that testing on larger models and more diverse architectures, especially transformer-based LLMs, is a natural extension. However, developing robust unlearning benchmark datasets at large scales, such as those involving LLMs, with clear and meaningful evaluation metrics remains an active area of research [1–2]. Existing benchmarks [3-4] often rely on indirect and sometimes ambiguous metrics such as membership inference attacks, accuracy comparisons between $D_r$ and $D_f$, and statistical divergence tests. In contrast, our controlled experiments offer unambiguous evaluation criteria. For example, in the Multi-Class Label Erasure experiment, the retain and forget sets differ only in color distribution, as the retain set contains only gray images while the forget set contains red and green images. This allows us to precisely measure retain quality and forget quality, since we know the ground truth unlearned model retrained from scratch on $D_r$ demonstrates two key behaviors: (i) it is a strong classifier of gray inputs and (ii) it always predicts the color of an image to be gray with probability 1, no matter its actual color. This allows us to clearly measure retain quality through classification accuracy on gray test images, and forget quality through the mean squared error between the model's predicted confidence that each input image is gray and the ideal value of 1. These clear metrics enable meaningful comparison across methods and highlight the core contributions of our framework. We recognize that scaling to larger models and different architectures is an important direction, which we plan to explore in future work.
>
> **(Q3): Why did the authors leave out any established benchmarks/metrics? Additional results here would make me increase my score on this otherwise excellent paper**
>
> (A3) We hope the above responses help to address your question. In summary, we appreciate the reference to established evaluation metrics such as those from Goel et al., and we note that our evaluation metrics can be viewed as natural analogs to those proposed in that work. Regarding our focus on novel unlearning tasks rather than standard benchmarks, our goal was to design clear and interpretable settings that enable unambiguous and insightful measurements of unlearning performance. Many existing benchmarks come with limitations related to robustness and interpretability. We will add this rationale and discussion to the revised paper, as we agree this design choice is important and merits further explanation.
>
> **Concluding Remark:**
>
> We hope that the clarifications above adequately address your concerns, and we are committed to address any additional questions you may have during the discussion phase.
>
> [1] Feng, Z., et al. "Existing Large Language Model Unlearning Evaluations Are Inconclusive."
>
> [2] Hu, S., et al. "BLUR: A Benchmark for LLM Unlearning Robust to Forget-Retain Overlap."
>
> [3] Shi, W., et al. ``MUSE: Machine Unlearning Six-Way Evaluation for Language Models."
>
> [4] Maini, P., et al. ``TOFU: A Task of Fictitious Unlearning for LLMs."

---

> > ### Comment · Reviewer_oxow · 2025-08-01
> >
> > Thank you for the detailed clarifications. No further questions from my side.

---

### Official Review · Reviewer_ditX · 2025-06-26

**Clarity:** 3
**Significance:** 2
**Originality:** 3
**Rating:** 4
**Confidence:** 4

**Summary:**

The paper considers the problem of unlearning for overparameterized models. In such cases, even the definition of what we mean by an unlearned model is unclear since the original model may interpolate the retain set data and hence trivially minimize the loss. To resolve this, the paper defines the ideal unlearning solution as the one which minimizes some notion of complexity (e.g. parameter norm) while interpolating the retain set.

To approximately find such a solution, the paper proposes an algorithm that roughly does the following: find an update to the original parameters that minimize the complexity measure, while ensuring that the update is small and is orthogonal to gradients of retain set points. The orthogonality condition is a first order relaxation of the interpolation condition and the update being small ensures that the first order relaxation is accurate. The theory results that follow establish that the approximate solution hence found is reasonable: it is exact in the case of a linear model, and for a 2-layer neural network, it finds an interpolating solution while keeping the number of active neurons small.

The paper then proposes a more practically applicable version of the algorithm focusing on $\ell_2$ norm as the complexity measure. The algorithm roughly does the following: (i) Do (mini-batch) gradient descent on the retain set, (ii) Every once in a while, update the parameter to minimize its norm, while ensuring that the update is small and is orthogonal to retain set gradients. For efficiency, this update is done using a small subset of retain set minibatch.

Finally, the paper designs several clever unlearning evals (altered versions of MNIST/CIFAR datasets) and shows that the proposed method performs better than many existing baselines.

**Questions:**

See the strengths and weaknesses discussion.

**Ethical Concerns:**

["NO or VERY MINOR ethics concerns only"]

**Final Justification:**

I thank the authors for their detailed response. I am increasing the score to borderline accept.

The proposed method is definitely very interesting and well-motivated. However, in the wall-clock time comparison, it seems to do similar to retraining from scratch in one case and running GD in another case (both are standard baselines). It is possible that with optimized implementations, it can outperform these standard baselines. However, without clear evidence, I feel hesitant to recommend a clear acceptance. It would make the paper much stonger if the authors could demonstrate the supriority using optimized implementations.

Also, I think the comparison with the ridge baseline is worth building on more. It seems somewhat surprising to me that constraining the updates to be orthogonal to retain set minibatch can be significantly more effective than plain ridge baseline, given that the mini-batch sizes are typically very small compared to the retain set size. However, the justification authors gave for this definietly seems plausible. While this is nor strictly necessary for accepatance, I think the paper will become much stronger (maybe in the future version) with some more analysis and intuitions comparing the two methods.

**Limitations:**

Yes.

**Quality:**

3

**Strengths And Weaknesses:**

**Strengths**

- The paper is nicely written and was a fun read.

- The proposed method is natural and easy to integrate in existing SGD pipelines (at least in terms of implementation), and seems to beat many existing methods.

- I liked the evals which could be of independent interest. While they may not be sufficient to show that an unlearning method is good, they are certainly necessary and easier to compute compared to some of the existing evals in the unlearning literature.


**Weaknesses**

- The most exciting part of the paper for me was that the proposed method is very simple, and beats many existing methods. However, I don't fully understand why. The theory seems a bit disconnected from the actual algorithm used and doesn't provide good justification of why the method works so well. In particular, it was unclear why the method outperforms plain $\ell_2$ regularization. It seems the proposed method is essentially doing $\ell_2$ regularization subject to the constraint that the effect of this regularization remains in space orthogonal to retain set minibatch subset. But if this subset is small, then one still ends up shrinking the parameters in the rest of the subspace spanned by the retain set but not captured by this subset. In that sense, it doesn't seem that different from pain $\ell_2$ regularization. It would be good if the authors can clarify this in detail.

- The evaluation needs some more work to make them convincing:
(i) The current cost comparison is in terms of number of epochs. But since the per epoch cost differs across methods (e.g. there is expensive orthogonalization in the proposed method), it would be good to compare the wall clock runtime as well.
(ii) While the proposed evals are nice and seem like a necessary test any proposed unlearning method should pass, the paper would become much stronger if it could compare the method on existing evaluation criterion as well such as KLoM or ULIRA (see [1] for a discussion). Having said that, these evaluations are expensive and it is understandable if a paper omits them.

[1] Georgiev, Kristian, et al. "Attribute-to-delete: Machine unlearning via datamodel matching." arXiv preprint arXiv:2410.23232 (2024).

---

> ### Author Rebuttal · Authors · 2025-07-31
>
> Dear Reviewer ditX, we are glad to see you enjoyed reading the paper. We answer your remaining questions below.
>
> **(Q1): The most exciting part...**
>
> (A1): Thank you for this question on plain L2 regularization, which we refer to as Ridge. We note that Ridge is not a previously proposed unlearning method; rather, we include specifically it as a baseline to contrast our proposed algorithm.
>
> As you mention, the Ridge update modifies the model parameters by descending the loss over a mini-batch while simultaneously shrinking the parameter vector towards the origin. In contrast, MinNorm-OG alternates between two decoupled steps: (i) descending the loss, and (ii) shrinking the parameter norm only in the subspace orthogonal to the gradient of the model's output over the retain samples. This orthogonal projection ensures that the complexity reduction step does not interfere with the model performance on the retain set. Even though MinNorm-OG approximates the true orthogonal subspace using a random subset of each batch, this is often sufficient in practice for preserving the critical directions for retain-set performance.
>
> In contrast, the Ridge shrinks the parameter vector in directions which can interfere with the loss minimization over the retain set and lead to suboptimal updates. While Ridge can eventually converge to a decent solution, it requires more epochs and greater access to the retain set. In Table 1 (Data Poisoning), Ridge performance improves as the number of epochs increases (the last column reflects Ridge -- the column header ``RegL2” is a typo that will be corrected.) Similarly, in Table 2 (Representation Collapse), Ridge performs better as both retain set access and epochs increase. Thus, Ridge is a reasonable, yet suboptimal, unlearning method as it is not specifically tailored to the bi-level formulation of unlearning in Eqn. (4).
>
> We will make sure to include this discussion in the revised version of the paper.
>
> **(Q2): The evaluation...if a paper omits them.**
>
> (A2)
>
> (i): Regarding our experimental constraint, we note that it is standard in prior work [1-3] to fix the number of epochs across methods for each experiment. Further, it is increasingly common to evaluate optimization algorithms based on the number of samples processed, especially for LLMs where efficiency is often measured per token. That said, we agree that adding runtime comparisons would strengthen our results.
>
> We use the Multi-Class Label Erasure experiment for the runtime analysis as it represents our largest scale experiment with the most fine-grained evaluation metrics. We create copies of either MNIST or CIFAR-10 colored gray, red, and green, where the retain set comprises only gray images while the forget set includes red and green samples. We train a model to predict both the image content class and the image color. To evaluate unlearning, we measure retain accuracy as the model’s accuracy on gray test images (higher is better), and forget error as the mean squared error between the model’s predicted probability that each input image is gray and the ideal value of 1 (lower is better).
>
> We first present the run-times of each method for unlearning with 5 epochs in the same setting as Figure 2. For MinNorm-OG, we present the runtime using the hyperparameters which result in the best performance for the fixed epoch constraint.
>
> **Mean Runtime (ms)**
> |Method      |MNIST |CIFAR-10|
> |------------|------|--------|
> |Retrain     |83.71 |120.33  |
> |GD          |81.94 |112.97  |
> |GA          |87.19 |256.11  |
> |NGD         |84.66 |115.84  |
> |NGP         |116.63|273.85  |
> |Scrub       |98.80 |176.28  |
> |NPO         |100.58|334.79  |
> |Ridge       |82.39 |116.85  |
> |MinNorm-OG  |574.49|525.78  |
> |SalUn       |270.62|784.31  |
> |KL-Unif     |114.70|274.02  |
>
> MinNorm-OG incurs a relatively large per-epoch cost relative to the other methods as expected. Note that SalUn, Retrain, and KL-Unif were added to address the questions of Reviewers Tqz5 and enfq.
>
> We run additional experiments for unlearning given a fixed runtime budget rather than a fixed epoch budget. We sweep hyperparameters for each unlearning method and record sets of Pareto optimal points for each method. When possible, we select 3-4 points per method. We format the best performing methods in bold. ``Original Model" denotes the initial model before any unlearning. The range of the retain accuracies is [0,1] as it computes accuracy on gray test images, while the forget error ranges from .67 (the worst possible value when there are 3 color classes) to 0 (perfect forgetting).
>
> **Multi-Class Label Erasure Results on MNIST, 200 ms**
> |Method       |Ret. Acc. (%)↑|Forget Err.↓|
> |-------------|----------|--------|
> |Original     |98.6      |.67     |
> |**Retrain**      |84.0      |.00     |
> |GD           |95.6      |.28     |
> |GA           |40.0      |.48     |
> |GA           |10.3      |.00     |
> |NGD          |98.6      |.67     |
> |**NGP**          |95.1      |.25     |
> |             |84.1      |.01     |
> |Scrub        |95.1      |.39     |
> |             |94.2      |.22     |
> |NPO          |50.2      |.44     |
> |             |18.3      |0.0     |
> |Ridge        |95.6      |.27     |
> |**MinNorm-OG**   |94.1      |.18     |
> |             |89.1      |.10     |
> |SalUn        |96.8      |.58     |
> |KL-Unif      |95.0      |.37     |
>
> **Multi-Class Label Erasure Results on CIFAR-10, 700 ms**
>
> |Method      |Ret. Acc. (%)↑|Forget Err.↓|
> |------------|----------|-------|
> |Original    |79.0      |.67    |
> |Retrain     |14.0      |.00    |
> |**GD**          |72.5      |.55    |
> |            |61.0      |.05    |
> |            |23.6      |.00    |
> |GA          |65.0      |.48    |
> |            |10.1      |.45    |
> |NGD         |73.0      |.65    |
> |            |57.8      |.45    |
> |NGP         |73.4      |.54    |
> |            |61.8      |.21    |
> |            |18.0      |.00    |
> |Scrub       |73.0      |.64    |
> |            |61.0      |.10    |
> |            |23.9      |.04    |
> |NPO         |67.0      |.55    |
> |            |20.2      |.36    |
> |**Ridge**       |72.6      |.55    |
> |            |61.4      |.06    |
> |            |23.6      |.00    |
> |**MinNorm-OG**  |79.4      |.66    |
> |            |74.8      |.55    |
> |            |62.3      |.17    |
> |            |61.1      |.07    |
> |SalUn       |73.0      |.55    |
> |            |65.1      |.37    |
> |            |51.1      |.31    |
> |            |23.6      |.02    |
> |KL-Unif     |73.3      |.54    |
> |            |61.2      |.30    |
> |            |18.8      |.00    |
>
>
> Retrain, NGP, and MinNorm-OG perform best on MNIST. Retraining from scratch on the accessible subset of the retain set can achieve some success since MNIST is not as complex as CIFAR-10. For CIFAR-10, GD, Ridge, and MinNorm-OG perform best. In both experiments, methods with cheaper per-epoch updates perform relatively better under a fixed time budget, as they complete more optimization cycles. Despite this, MinNorm-OG remains a strong performer even though we have not optimized its implementation. Further improvements are likely possible through:
> - Caching gradients between loss descent and projection steps
> - Using lower-level GPU libraries (e.g., cuSOLVER) instead of `torch.linalg.qr`
>
> (ii) Regarding evaluation metrics: while general-purpose metrics like KLoM and ULIRA are valid, we design task-specific metrics tailored to each experiment. This improves clarity and interpretability, allowing for a more precise understanding of unlearning performance. We agree that the paper would benefit from additional discussion connecting our metrics to existing criteria, which we will include in the revised version.
>
> As noted by Reviewer oxow, our metrics can be seen as direct analogs to prior work [4]. While our metrics are not exact analogs to KLoM for example, they share the same underlying motivation but offer more refined, task-specific insights.
>
> KLoM computes the KL divergence between a model’s predicted class distribution and that of models sampled from a “safe” distribution (e.g., retrained from scratch). In the Multi-Class Label Erasure task, the ground truth unlearned models always classify gray inputs correctly and predict all input images as gray with probability 1. This enables direct evaluation via (i) accuracy on gray test images and (ii) squared error between the predicted gray probability and 1 on colored test inputs. These metrics quantify distributional distance to the true unlearned model in a task-aligned way, rather than relying on general-purpose KL divergence. Similarly, in the Representation Collapse experiment, the ground-truth unlearned models base predictions on color, not content, allowing evaluation via color-based accuracy. These metrics provide a more grounded, data-driven view of distributional change. We will include this discussion of the relationship between our evaluation methods to existing criteria such as KLoM in the updated paper.
>
> We appreciate the reviewer’s recognition that our metrics are necessary tests unlearning methods should pass. We note that many baseline algorithms struggle even in simple settings like the Data Poisoning experiment. We present these results not to diminish prior work but to motivate the value of our analysis and its practical insights.
>
> **Concluding Remark:**
>
> We hope that our clarifications and additional results have adequately addressed your concerns. If so, we kindly request you to consider revising your score. We are happy to address any additional questions you may have during the discussion phase.
>
> [1] Maini, P., et al. ``TOFU: A Task of Fictitious Unlearning for LLMs."
>
> [2] Shi, W., et al. ``MUSE: Machine Unlearning Six-Way Evaluation for Language Models."
>
> [3] Zhang, R., et al. "Negative Preference Optimization: From Catastrophic Collapse to Effective Unlearning."
>
> [4] Goel, Shashwat, et al. "Corrective Machine Unlearning."

---

> > ### Comment · Reviewer_ditX · 2025-08-04
> >
> > I thank the authors for their detailed response. I am increasing the score to borderline accept.
> >
> > The proposed method is definitely very interesting and well-moitivated. However, in the wall-clock time comparison, it seems to do similar to retraining from scratch in one case and running GD in another case (both are standard baselines). It is possible that with optimized implementations, it can outperform these standard baselines. However, without clear evidence, I feel hesitant to recommend a clear acceptance. It would make the paper much stonger if the authors could demonstrate the supriority using optimized implementations.
> >
> > Also, I think the comparison with the ridge baseline is worth building on more. It seems somewhat surprising to me that constraining the updates to be orthogonal to retain set minibatch can be significantly more effective than plain ridge baseline, given that the mini-batch sizes are typically very small compared to the retain set size. However, the justification authors gave for this definietly seems plausible. While this is nor strictly necessary for accepatance, I think the paper will become much stronger (maybe in the future version) with some more analysis and intuitions comparing the two methods.

---

> > > ### Author Response · Authors · 2025-08-05
> > >
> > > Dear Reviewer ditX,
> > >
> > > Thank you for your thoughtful comment. We greatly appreciate that you took our responses into account in updating your score.
> > >
> > > We acknowledge that, under a fixed runtime budget, the performance gap between our method and the strongest baseline in each experiment is smaller than under a fixed epoch budget. However, we would like to highlight a few points:
> > >
> > > - MinNorm-OG (ours) is the only consistently strong performer across both datasets
> > > - The more sophisticated baselines (NGP, Scrub, SalUn) are outperformed by our method and, in some cases, are even outperformed by the standard baselines (Retrain, GD).
> > >
> > > These results suggest that MinNorm-OG remains robust and efficient even under the specific constraint of a fixed runtime budget.
> > >
> > > To provide additional intuition on the difference between Ridge and MinNorm-OG, consider the simple case of an overparameterized linear model, as in our theoretical analysis, where the ground truth unlearning solution is the minimum $\ell_2$ norm interpolator over the retain set. This minimum norm solution naturally arises as the limiting point when running gradient descent on the squared loss when initialized at the origin. Further, we can compute this exact unlearning solution as the projection of the original model parameters $\theta^\ast$ onto the span of the retain set input vectors. Thus, given full access to the retain set, MinNorm-OG can recover this solution in a single epoch, as the unlearning subproblem exactly computes this projection. In contrast, Ridge must gradually reduce the parameter norm over many epochs while correcting the resulting error on the retain set training loss, since it cannot shrink the parameter norm without also affecting retain set performance. While this example only applies for a simple model and does not directly reflect the batched regime, we hope it helps provide some intuition on why Ridge may be less effective in general. We agree that the performance differences between MinNorm-OG and Ridge are worth highlighting and will add this discussion to the updated paper. More broadly, a formal analysis of the convergence behavior for the different unlearning methods under mini-batching is a compelling direction for future work.
> > >
> > >
> > > Overall, we believe our work offers a novel and principled framework for unlearning in the overparameterized regime, with provable guarantees and practical utility. Our proposed algorithm, MinNorm-OG, consistently performs well across both fixed-epoch and fixed-runtime settings, offering a new perspective on unlearning along with actionable insights. Thank you again for your thoughtful comments — we appreciate your engagement and are happy to address any further questions.

---

### Official Review · Reviewer_enfq · 2025-06-30

**Clarity:** 4
**Significance:** 3
**Originality:** 3
**Rating:** 3
**Confidence:** 5

**Summary:**

The paper addresses the new problem of machine unlearning in overparameterized models, where more than one solutions can fit all training data perfectly. The authors argue that traditional unlearning definitions are insufficient in such settings and proposes a new formulation that accounts for this non-uniqueness. The key contribution is a formal approach to define and achieve unlearning by considering the geometry of the solution space, rather than focusing solely on model outputs or loss values. The paper also introduces an unlearning algorithm aligned with this new definition and demonstrates their effectiveness through theoretical justification and empirical results.

**Questions:**

See weaknesses

**Ethical Concerns:**

["NO or VERY MINOR ethics concerns only"]

**Final Justification:**

I think the question of "what unlearning means in the overparameterized regime" is an excellent one, and the authors provide an interesting and thoughtful answer. However, I remain skeptical about the practical applicability of the proposed method. The core motivation for unlearning lies in its practicality (otherwise, one could simply retrain the model to achieve full removal guarantees). Unlearning becomes relevant when dealing with large-scale models for which retraining is prohibitively expensive. In this context, I am not convinced that such large models can be highly overparameterized in the way assumed by the authors, nor that the proposed method would be applicable or effective in such settings. Therefore, I am maintaining my original score.

**Limitations:**

yes

**Quality:**

3

**Strengths And Weaknesses:**

Strengths:
- The paper presents a novel perspective on machine unlearning in overparameterized models, where multiple solutions can achieve zero training loss. The authors make a compelling case for redefining unlearning under this regime and clearly articulate the motivation and implications.
- The paper is generally well-written, with clear exposition and sound mathematical formalism, making it accessible and rigorous.


Weaknesses:
1. The overparameterized setup considered in the paper aligns closely with the assumptions of the Neural Tangent Kernel (NTK) regime, where solutions that interpolate training data lie near the initialization. Prior unlearning work leveraging NTK (e.g., [1]) should be acknowledged and discussed to clarify the novelty and positioning of the current approach.

2. The practical benefit of the proposed unlearning approach is unclear without a runtime comparison to retraining from scratch, particularly in the overparameterized regime where training converges quickly. Since the proposed method relies on gradients over the retain set, the computational savings may be marginal unless the retain set is substantially smaller.

3. The authors should clarify whether unlearning Algorithm 1 with a subset of the retain set is reasonable.
If so, the authors should compare it with retrain with the subset of the retain set.

4. The motivation for unlearning in overparameterized models would be stronger if grounded in realistic scenarios, especially those involving large models (e.g., LLMs) and massive datasets where retraining is costly. It remains unclear whether the setting proposed in the paper addresses such high-stakes unlearning use cases.

5. The paper focuses primarily on gradient ascent-based methods, but there are other possible approaches to address unlearning in overparameterized models. For instance, minimizing the KL divergence between the model's output logits and a uniform distribution over the label (i.e., minimizing $D_{KL}(f(x) || u)$ for $x\in D_f$) could serve as a simple and effective alternative. Discussion of such methods would provide a broader context for the proposed solution.

6. Typo: Missing period at the end of line 167.

7. Clarification needed: The use of the delta function in Equations (9) and (12) is unclear. Further justification is necessary.

[1] Golatkar, Aditya, Alessandro Achille, and Stefano Soatto. "Forgetting outside the box: Scrubbing deep networks of information accessible from input-output observations." ECCV 2020.

---

> ### Author Rebuttal · Authors · 2025-07-31
>
> Dear Reviewer enfq, thank you for highlighting the novelty of our analysis and the rigor of our arguments. We answer your questions below.
>
> **(Q1): ... NTK regime should be acknowledged...**
>
> (A1): Thank you for highlighting Golatkar et al. which studies unlearning for fine-tuning a pretrained network. Since the pretrained network is often overparameterized for the fine-tuning task, they assume fine-tuning causes only a small change in weights. Thus, both the original model and the ideal unlearned model (fine-tuned only on $D_r$) remain close to the pretrained initialization. This enables the use of NTK theory, which approximates the result of fine-tuning by linearizing the network around the pretrained weights. We clarify the distinction to our approach below.
>
> We consider the general problem of unlearning under overparameterization, not the specific problem of unlearning for fine-tuning. We make no assumption about the magnitude of the weight updates, as unlearning in general may require substantial weight updates which are not well captured by the NTK approximation.
>
> Although our framework applies a linearization, it is fundamentally different from the linearization used in the NTK regime. We apply a linearization to handle the complex interpolation constraint in Eqn. (4) which defines the ideal unlearning solution as the minimum-complexity interpolator of the retain set. Since the original model is such an interpolator, we approximate the constraint by its linearization around the original weights. This leads to the tractable objective in Eqn. (7) which approximates the exact unlearning problem in Eqn. (4).
>
> Further, our practical method MinNorm-OG accounts for unlearning settings which require large weight updates by iteratively solving a version of Eqn. (7) and applying loss descent steps to restore feasibility for Eqn. (4), since in general the linearization of Eqn. (7) is only locally accurate. Unlike the NTK approach, our method handles large weight updates explicitly.
>
> We will revise the paper to include a discussion of this related work.
>
> **(Q2) Unlearning with retain set...**
>
> (A2) In practice, empirical unlearning methods typically assume some access to retain set samples. All baseline methods we compare against rely on this access, except for GA which performs very poorly. Without access to the retain set, developing a robust and general unlearning algorithm becomes extremely challenging.
>
> Specific settings, such as unlearning within our framework using the basic model classes considered Section 3 or unlearning for fine-tuning using NTK theory, make unlearning feasible with access only to model gradients. However, these rely on strong assumptions which fail to hold in general.
>
> **(Q3) Compare to retrain... minimizing the KL divergence to uniform...**
>
> (A3) We define your proposed method KL-Unif which minimizes a weighted sum of the retain set loss and the KL divergence between the model’s output and the uniform class distribution on the forget set.
>
> **We include comparisons to retraining from scratch on the accessible subset of the retain set and KL-Unif**. We run these methods on the Multi-Class Label Erasure experiment for the same settings shown in Figure 2. Due to space constraints we present the results in our response (A3) to Reviewer tqz5 above.
>
> Retraining struggles to achieve strong retain accuracy with limited epochs but always attains 0 forget error. MinNorm-OG achieves near-zero forget error with much better retain accuracy. On MNIST KL-Unif maintains decent accuracy at moderate forget error but cannot reduce forget error further, but it performs poorly on CIFAR-10, failing to lower forget error without harming retain accuracy. MinNorm-OG consistently offers better tradeoffs on both datasets.
>
> We will include these comparisons in the updated paper.
>
> **(Q4) Runtime comparison...**
>
> (A4) We compare mean runtimes over 5 epochs in the Multi-Class Label Erasure experiment. For MinNorm-OG, we report the runtime for the hyperparameters that gave the best performance in Figure 2. For other methods see our response (A2) to Reviewer ditX.
>
>  **Mean Runtime (ms)**
>
> |Method      |MNIST |CIFAR-10|
> |------------|------|--------|
> |Retrain     |83.71 |120.33  |
> |MinNorm-OG  |574.49|525.78  |
> |KL-Unif     |114.70|274.02  |
>
> Retraining is one of the cheapest methods, while KL-Unif is somewhat more expensive as the loss function is more involved. Lastly, MinNorm-OG incurs a relatively large per-epoch cost as expected. Due to this variation, we also compare the performance of each method under a fixed wall-clock runtime budget. For the full table see our response (A2) to Reviewer ditX below.
>
> **Multi-Class Label Erasure Results on MNIST, 200 ms**
>
>
> |Method       |Ret. Acc. (%)↑|Forget Err.↓|
> |-------------|----------|--------|
> |Original     |98.6      |.67     |
> |Retrain      |84.0      |.00     |
> |MinNorm-OG   |94.1      |.18     |
> |             |89.1      |.10     |
> |KL-Unif      |95.0      |.37     |
>
>
> On MNIST, retraining quickly learns a decent classifier with limited data. MinNorm-OG improves retain quality (89.1% vs. 84% accuracy) with a small increase in forget error (0.1 vs. 0, lower is better). KL-Unif fails to achieve low forget error, while MinNorm-OG offers a better tradeoff, reducing forget error while preserving retain accuracy.
>
> **Multi-Class Label Erasure Results on CIFAR-10, 700 ms**
>
> |Method      |Ret. Acc. (%)↑|Forget Err.↓|
> |------------|----------|-------|
> |Original    |79.0      |.67    |
> |Retrain     |14.0      |.00    |
> |MinNorm-OG  |79.4      |.66    |
> |            |74.8      |.55    |
> |            |62.3      |.17    |
> |            |61.1      |.07    |
> |KL-Unif     |73.3      |.54    |
> |            |61.2      |.30    |
> |            |18.8      |.00    |
>
> On CIFAR-10, retraining fails as the dataset cannot be learned quickly from a small subset. MinNorm-OG achieves 61.1% accuracy with a low forget error of 0.07. Additionally, MinNorm-OG outperforms KL-Unif at intermediate points, reaching 62.3% retain accuracy at 0.17 forget error, while KL-Unif achieves 61.2% accuracy with a worse 0.30 forget error.
>
> KL-Unif may struggle to reduce forget error because its KL term forces the forget set predictions toward a uniform distribution. This may work when the retain and forget sets are completely unrelated, but in realistic settings with shared structure, like our experiments, it struggles to recover the correct unlearned model.
>
> **(Q5): The motivation...use cases.**
>
> (A5): Thank you for raising this point. While our experiments focus on model and dataset sizes up to ResNet-18 on CIFAR-10, our primary goal is to present a set of interpretable experiments supported by unambiguous metrics. These metrics precisely quantify whether our proposed algorithm MinNorm-OG, developed as an extension of our theoretical framework, offers consistent improvements over existing baseline methods. We show clear failure cases for existing methods and demonstrate the stronger performance of MinNorm-OG, supporting the practical relevance of our framework and beyond the settings where we can provide formal guarantees.
>
> We agree that scaling evaluation to larger models, especially LLMs, is a natural and important next step. However, developing robust large-scale unlearning benchmarks with clear evaluation metrics remains an active research area [1–2]. Existing benchmarks [3–4] often use ambiguous metrics such as membership inference attacks and statistical divergence tests. In contrast, our experiments provide unambiguous evaluations. For example, in the Multi-Class Label Erasure experiment, the retain and forget sets differ only in color distribution, as the retain set contains only gray images while the forget set contains red and green images. This allows us to precisely measure retain quality as classification accuracy on gray test images, and forget quality through the mean squared error between the model's predicted confidence that each input image is gray and the ideal value of 1, since the ground truth unlearned model always predicts input images to be gray with full confidence. These clear metrics enable meaningful comparison across methods and highlight the core contributions of our framework.
>
> **(Q6): The use of the delta function...**
>
> (A6): To derive our surrogate objective in Eqn. (7) for the exact unlearning problem in Eqn. (4), we relax the interpolation constraint in Eqn. (4) via a linearization around the original model. We then introduce a drift regularization term, denoted $\hat{R}$, to control the error introduced by this linearization for different basic model classes. Thus, we minimize $\tilde{R}$ defined as $R + \hat{R}$. In Eqns. (9) and (12), we specify this $\tilde{R}$ function for (i) linear networks where $R$ measures predictor norm and (ii) 2-layer perceptrons where $R$ measures network width. In both cases, the delta functions appear as the definition of the regularizer $\hat{R}$ within this full objective $\tilde{R}$. $\hat{R}$ ensures that the minimizing Eqn. (7) over the relaxed constraints either recovers the exact solution to Eqn. (4), or returns a feasible interpolating model with tightly controlled complexity. Without $\hat{R}$, minimizing only $R$ over the linearized constraints would fail to unlearn, as shown in Appendix D.4.1.
>
> We will correct the typo Line 167.
>
> **Concluding Remark:**
>
> We hope that our responses and additional results have adequately addressed your concerns. If so, we kindly request you to consider revising your score. We are committed to address any additional points you may have during the discussion phase.
>
> [1] Feng, Z., et al. "Existing Large Language Model Unlearning Evaluations Are Inconclusive."
>
> [2] Hu, S., et al. "BLUR: A Benchmark for LLM Unlearning Robust to Forget-Retain Overlap."
>
> [3] Shi, W., et al. ``MUSE: Machine Unlearning Six-Way Evaluation for Language Models."
>
> [4] Maini, P., et al. ``TOFU: A Task of Fictitious Unlearning for LLMs."

---

> > ### Comment · Reviewer_enfq · 2025-08-06
> >
> > I personally value rigorous theoretical work, even when it relies on idealized assumptions (such as the NTK regime) or focuses on small-scale scenarios as a proof of concept. The question of "what unlearning means in the overparameterized regime" is a genuinely interesting one, and I commend the authors for presenting a thoughtful and well-structured perspective.
> >
> > However, I remain concerned about the practical applicability of the proposed approach. Unlearning is fundamentally an application-driven problem, its significance lies in scenarios where retraining large-scale models is computationally prohibitive. The motivation for unlearning becomes less compelling if the proposed method does not scale to such settings. In this context, I am not fully convinced that the assumptions made (e.g., highly overparameterized models) reflect the conditions in which unlearning is most relevant.
> >
> > While I appreciate the theoretical contributions of this work, these concerns regarding practical relevance lead me to maintain my original score.

---

> ### Author Response · Authors · 2025-08-06
>
> Dear Reviewer enfq,
>
> Thank you for your response. We completely agree that the importance of unlearning lies in settings where it is infeasible to retrain from scratch. We would like to emphasize that to specifically address this concern, in our initial rebuttal we performed additional experiments comparing each method to retraining from scratch under two different computational constraints: (i) a fixed number of unlearning epochs, and (ii) a fixed wall clock runtime. We show these results in the tables below for our Multi-Class Label Erasure experiment using ResNet-18 on CIFAR-10. Retain Accuracy measures classification accuracy on the retain set (higher is better). Forget Error quantifies how well the model removes knowledge of the non-gray forget set, computed as the mean squared error between the predicted probability that each test image is gray and the ideal value of 1.0, as the ideal unlearned model always predicts gray with full confidence, regardless of the true color. Forget Error ranges from 0.0 (perfect forgetting) to 0.67 (no forgetting).
>
> **Multi-Class Label Erasure on CIFAR-10 given 5 Unlearning Epochs**
>
> | Method       | Retain Accuracy (%) ↑ | Forget Error↓ |
> |--------------|--------------|---------------|
> | Original     | 79.0         | 0.67          |
> | Retrain      | 10.0         | 0.00          |
> | MinNorm-OG   | 53.0         | 0.11          |
>
> **Multi-Class Label Erasure on CIFAR-10 given 700 ms**
> | Method       | Retain Accuracy (%) ↑ | Forget Error ↓ |
> |--------------|----------------|---------------|
> | Original     | 79.0           | 0.67          |
> | Retrain      | 14.0           | 0.00          |
> | MinNorm-OG    | 61.1           | 0.07   |
>
> We see that for both computational constraints, retraining from scratch on the retain set completely fails, as given a 5 unlearning epochs the retrained model has an accuracy of 10.0% which is no better than random labeling, while given a fixed runtime of 700ms Retraining only achieves an accuracy of 14.0% In contrast, our method MinNorm-OG given 5 unlearning epochs achieves a retain accuracy of 53.0% with forget error of .11, and given 700ms for unlearning we achieve a retain accuracy of 61.1% with a forget error of .07. Thus,  **our experiments do reflect the regime where retraining from scratch is computationally prohibitive**, as given the same computational constraints retraining from scratch barely achieves a better than random classifier, while our method achieves much stronger performance with small forget error.
>
> We note that in the established work (Golatkar et. al.) which you mention, they consider a similar experimental setup where CIFAR-10 images are unlearned from a ResNet-18 model. While we agree that continuing to scale to larger datasets and models is a natural extension, we hope our additional results which compare the effectiveness of retraining to our method clarify your concerns. Thank you again for your comments.

---

> > ### Comment · Reviewer_enfq · 2025-08-07
> >
> > Thank you for the additional clarification and detailed results.
> >
> > I had already reviewed these experiments and understand the motivation: retraining from scratch can be time-consuming and often infeasible (particularly when the goal is to achieve full memorization), even for relatively small models like ResNet-18. I also appreciate that your method demonstrates improved performance under fixed runtime and epoch constraints.
> >
> > That said, I still find the overall problem setup somewhat self-contradictory, especially given that unlearning is fundamentally a practical problem. Its relevance arises in large-scale real-world settings where retraining is prohibitively expensive. Yet such models are unlikely to memorize individual examples to the extent assumed in your formulation. For instance, your experiments show that the original model achieves only 79% accuracy, indicating that the model does not fully memorize the training data.
> >
> > This creates a tension in the setup: the problem scale must be small enough to allow full memorization (to justify the problem formulation), yet large enough that retraining is impractical (to justify the need for unlearning). These two conditions are rarely satisfied simultaneously in realistic scenarios. As a result, the setting feels disconnected from the practical cases where unlearning is most urgently needed.
> >
> > While I value the theoretical contributions and the clarity of exposition, this underlying mismatch limits the practical relevance of the work. I therefore maintain my score, though I appreciate the thoughtfulness of the approach.

---

> ### Author Response · Authors · 2025-08-07
>
> Dear reviewer enfq,
>
> Thank you for the response. We believe there was a misunderstanding: **Retain Accuracy denotes the accuracy of the model on held out test images drawn from the same distribution as the retain set (i.e., gray test images), not the training set itself**, as it is meant to measure the model utility after unlearning. We apologize if that was unclear in our previous response. When training ResNet-18 on CIFAR-10 we do see that the model nearly interpolates the entire training data, achieving 99-100% training accuracy over all trials. While we are unable to fully complete larger scale experiments in the short discussion period, we have already observed in preliminary results training ResNet50 on TinyImageNet the same behavior that the training images are predicted with almost 100% accuracy. In many additional realistic scenarios, especially in the context of LLMs, it has been observed that models memorize training samples, as LLMs output exact training text verbatim [1] and can be prompted to output specific samples or knowledge [2-3]. Further, larger models have been observed to memorize samples to a larger degree [4]. Thus, we believe the problem of memorization is especially important at large scales where retraining is infeasible, and this is the setting we aim to capture in our framework.
>
> [1] "Extracting training data from large language models." (Carlini et al.)
>
> [2] "Muse: Machine unlearning six-way evaluation for language models." (Shi, et al.)
>
> [3] "Are large pre-trained language models leaking your personal information?." (Huang et al.)
>
> [4]  "Quantifying memorization across neural language models."  (Carlini, et al.)

---

> > ### Comment · Reviewer_enfq · 2025-08-08
> >
> > Thank you for the clarification regarding Retain Accuracy. I now realize my earlier concern was based on a misunderstanding (I had interpreted it as the accuracy on the training portion of the retain set, rather than on held-out test images from the same distribution). This resolves part of my confusion, and I appreciate you pointing it out.
> >
> > I also agree it is reasonable to observe near-zero training loss (and thus effective interpolation) with models of moderate size such as ResNet-18, and your additional results and references make a convincing case that memorization is a real phenomenon in both vision and language domains.
> >
> > That said, I still believe that memorization in larger-scale models such as LLMs often corresponds to a small fraction of the dataset, and is more about memorizing specific subsets rather than uniformly interpolating across the entire training set. This difference in the nature of memorization could have implications for how unlearning is formulated and evaluated.
> >
> > Nevertheless, based on your clarification, I now think the setup is more relevant than I initially judged, and I agree that memorization, even if partial, is indeed a key reason why unlearning remains important in practice. I will keep this in mind and adjust my score accordingly during the final discussion period if needed.

---

### Official Review · Reviewer_Tqz5 · 2025-07-01

**Clarity:** 3
**Significance:** 2
**Originality:** 3
**Rating:** 3
**Confidence:** 4

**Summary:**

The paper focuses on unlearning in over-parameterized settings, where data is interpolated by the model – a setting where related work in unlearning is still scarce. It provides solid theoretical underpinnings for this setting, focusing on showing why gradient-based unlearning is vacuous once the model interpolates, and proposing a principled alternative (algorithm MinNorm-OG) alongside provable guarantees for linear and shallow-network model classes. The paper also provides empirical evaluations of the proposed method against several (rather simple) baselines.

**Questions:**

To improve my score(s) the above comments must be taken into account.

**Ethical Concerns:**

["NO or VERY MINOR ethics concerns only"]

**Final Justification:**

After reading all rebuttals and reviews, I fail to see the practicality of the setting studied in the paper.
I especially find it very hard to believe that all training examples will be interpolated - to the best of my knowledge, this is never the case in real-life scenarios. So, I will keep my score at 3.

**Limitations:**

I was happy to see that the conclusions section is forthcoming about limitations. Recognizing additional limitations (such as simple/easy baselines, etc.) would help here.

**Paper Formatting Concerns:**

None.

**Quality:**

3

**Strengths And Weaknesses:**

Strengths:

1.	The unlearning problem in overparameterized settings where data is interpolated is indeed an interesting (perhaps corner) case that deserves shedding light into.

2.	The theoretical argumentation and development is very nicely done and informative – especially the discussion around single versus multiple existing solutions and the impact on unlearning beyond loss optimizations.

3.	The effort to provide empirical results (substantiating the usefulness of the theoretical work) is noteworthy (albeit rather falling short of true practicality in real-world settings).

Weaknesses:

Quality: 3

The paper’s technical depth and rigour is high but the caveats below reduce the overall mark.

1.	The paper’s claim that prior works (in the under-parameterized setting) define the solution as “any loss minimizer over the retained set’ is inaccurate. Loss functions are much more complex in SOTA methods (involving retain sets, forget sets, influence functions and noising ops on forget data, etc.). Hence, the paper’s motivation and problem formulation (looking as the only solution the minimizer on the retain data, D_r) should be re-examined. (By the way, line 113 should have D_r, not D). I urge the authors to explain this in more detail, as this (or I) may (have) be misunderstood.

2.	The paper targets very simple network architectures – I realize this is an obvious criticism, but an important one, nonetheless.

3.	The paper’s “from theory to practice argumentation” is rather overselling its contents – see the criticism on simplistic models used above. In addition, the datasets used are also rather simplistic.

4.	The baselines used are rather rudimentary, perhaps with the exception of SCRUB and NGP. More recently many new unlearning methods have emerged as SOTA. I urge the authors to consider, for instance, algorithms like SalUn [Fan et al, ICLR24] and the framework RUM [Zhao et al, Neurips24].

5.	Speaking of RUM, which focuses on memorization and its affect on unlearning, it would be instructive (and extremely interesting IMHO) to investigate the parallels between interpolation and memorization.  In the RUM paper, for example,  authors found (empirically) a similar result (that unlearning becomes vacuous for low-memorized data).

6.	Speaking of the SalUn method, which first identifies the “salient” weights and only updates them (instead of all parameters) during unlearning, a question relevant to this paper is how this affects the existence of multiple solutions in the overparameterized setting – are there fewer differences compared to strong solutions in the under-parameterized setting?

Clarity: 3.

1.	The paper is nicely written, with formal/crisp language. Perhaps it is overloaded early on with symbols etc which may alienate not very-well-versed readers?


Significance:  2.

Overall, the practical significance is rather limited – as argued in the quality section.


Originality: 4.

Originality is high – to my knowledge this is the first formal grounding of unlearning within an over-parameterized setting and the first principled solution for this.

---

> ### Author Rebuttal · Authors · 2025-07-31
>
> Dear Reviewer Tqz5, thank you for highlighting the depth and originality of our contributions. Thank you for appreciating the rigor of our arguments, and we are glad you found the writing clear. We address your questions below.
>
> **(Q1): The paper’s claim...misunderstood.**
>
> (A1): Thank you for the question. We believe there was a misunderstanding due to our word choice which we aim to clarify.
>
> When we say the unlearning solution has been previously defined as "any loss minimizer over the retained set," we refer to the definition of the ideal, ground truth unlearned model as the best performing model on the retain set, meaning it is a minimizer of the *training loss* over the retain set. We distinguish the training loss from the *unlearning objectives*, which are the objective functions minimized by the different unlearning methods, which as you mention can involve more complicated structure.
>
> We discuss this definition of the ideal unlearned model used in prior work, which defines it as any minimizer of the training loss over $D_r$, in order to highlight its limitations in the overparameterized setting. In this regime, the original model already interpolates the retain set and achieves zero training loss. As a result, the prior definition would classify the original model as a valid unlearned model, despite the fact that it completely memorizes $D_f$. This motivates our improved definition in Eqn. (2) of the ideal unlearned model as not just any minimizer to the training loss over the retain set, but the one with minimal model complexity.
>
> We will update the paper’s phrasing to clearly distinguish the training loss from the unlearning objective. Thank you for pointing out the typo in Line 113, we will correct this in the revised paper.
>
> **(Q2): The paper targets...simplistic.**
>
> The goal of our theory is to establish a principled framework with strong guarantees relating the tractability of solving the exact unlearning problem in Eqn. (4) using the relaxed surrogate proposed in Eqn. (7). To obtain rigorous guarantees, we focused on analyzable yet natural model classes. Extending the theoretical framework to handle deep non-linear networks remains an exciting and challenging direction for future work.
>
> While our experiments focus scales up to ResNet-18 on CIFAR-10, our goal is to enable clear, unambiguous unlearning evaluation. Our metrics precisely quantify whether MinNorm-OG, derived from our theoretical framework, consistently improves over baselines. We highlight clear failure cases of prior methods and show that MinNorm-OG performs reliably in practice.
>
> We agree that scaling to larger models like LLMs is an important next step. However, building robust unlearning benchmarks at this scale with clear, meaningful metrics remains an open challenge [1–2]. Current LLM benchmarks [3–4] often rely on indirect metrics like membership inference or statistical divergence tests. In contrast, our controlled experiments allow for precise evaluation. For example, in the Multi-Class Label Erasure experiment, the retain and forget sets differ only in color distribution, as the retain set contains only gray images while the forget set contains red and green images. This allows us to precisely measure retain quality as classification accuracy on gray test images, and forget quality through the mean squared error between the model's predicted confidence that each input image is gray and the ideal value of 1, since the ground truth unlearned model always predicts input images to be gray with full confidence. These clear metrics enable meaningful comparison across methods and highlight the core contributions of our framework.
>
> In our section titled “From Theory to Practice,” our goal is to show how our theoretical framework can be translated into an implementable algorithm that can be compared against baselines which perform iterative updates using access to a subset of the retain set. While this does not fully bridge the gap to extremely large-scale settings such as multi-billion-parameter LLMs, we believe it represents a concrete step toward making theoretically grounded unlearning methods applicable in empirical settings. If the reviewer feels that the current title is misleading, we are happy to revise it to something more appropriate, such as “Implementation Details.”
>
> **(Q3): The baselines...RUM.**
>
> (A3): Thank you for bringing up this additional method SalUN as well as the framework RUM. RUM is an interesting framework that proposes a meta-algorithm that applies different existing unlearning algorithms to subsets of the forget set based on a memorization score. We discuss the connection to RUM in our response (A4).
>
> To address the suggestion to incorporate SalUn, we have added comparisons to SalUN as well as to two additional baselines: (i) retraining from scratch on the retain subset, and (ii) KL-Unif, the method proposed by Reviewer enfq, which minimizes the KL divergence between the model's output distribution on the forget set and the uniform distribution over classes. (See our response to Reviewer enfq for further discussion.)
>
> We test on the Multi-Class Label Erasure experiment for the same settings shown in Figure 2. In this experiment, we create copies of either MNIST or CIFAR-10 colored gray, red, and green, where the retain set comprises only gray images while the forget set includes red and green samples. We train a model to predict both the image content class and the image color. To evaluate unlearning, we measure retain accuracy as the model’s accuracy on gray test images (higher is better), and forget error as the mean squared error between the model’s predicted probability that each input image is gray and the ideal value of 1 (lower is better). We sweep hyperparameters for each method and record the Pareto frontier of these two metrics. We include a table of Pareto optimal point(s) for each method. "Original Model" denotes the performance of the initial model before any unlearning.
>
> **MNIST**
>
> |Method      |Ret. Acc(%)↑|Forget Err.↓|
> |------------|----------|-------|
> |Original    |98.6      |0.67    |
> |SalUn       |98.3      |0.42   |
> |KL-Unif     |98.5      |0.27   |
> |Retrain     |66.1      |0.00   |
> |MinNorm-OG  |97.0      |0.03   |
>
> **CIFAR-10**
>
> |Method     |Ret. Acc(%)↑|Forget Err.↓|
> |-----------|----------|-------|
> |Original   |79.0      |0.67    |
> |SalUn      |63.0      |0.43   |
> |           |49.6      |0.32   |
> |KL-Unif    |58.3      |0.47   |
> |           |18.2      |0.21   |
> |Retrain    |0.10      |0.00   |
> |MinNorm-OG |67.5      |0.48   |
> |           |60.1      |0.20   |
> |           |53.0      |0.11   |
>
> On MNIST, SalUn slightly reduces forget error with minimal impact on retain accuracy but quickly plateaus, never achieving forget error below 0.42. MinNorm-OG, by contrast, approaches the ideal of 100% retain accuracy and 0.0 forget error. A similar pattern holds on CIFAR-10, where SalUn trades large drops in retain accuracy for modest forget gains, while MinNorm-OG achieves better forget error with far less retain loss. We will add this comparison to the paper.
>
> In our responses to Reviewers enfq and ditX, we include runtime-constrained experiments across all methods. SalUn consistently showed a poor tradeoff between retain and forget metrics and was relatively costly due to its initial computation of the "salient weights". See our response to Reviewer ditX for details.
>
> SalUn may struggle to unlearn for two main reasons. First, with random labels on the forget set, SalUn still trains with non-gray color labels, making it hard to lower forget error. Second, in the overparameterized regime, SalUn freezes parameters based on loss gradients at the original model, which may be near zero and mostly noise, making its salient weight selection unreliable.
>
> **(Q4): RUM...**
>
> (A4): Thank you for bringing up RUM and the relationship between memorization and interpolation. Given a randomized learning algorithm $A(D)$ that returns a distribution over models $f$, the memorization score of a sample $(x_i, y_i)$ is defined as the difference of:
>
> (i) the probability that a model $f \sim A(D)$ correctly classifies $x_i$ and
> (ii) the probability that a model $f \sim A(D \setminus {i})$ correctly classifies $x_i$.
>
> The memorization score from RUM measures how much a sample's inclusion in training improves the model's prediction. In the overparameterized regime, we assume the model class is rich enough to interpolate the entire dataset, so the first term in the memorization score is essentially 1 for every sample. This allows for large memorization scores under overparameterization. We will include a discussion of this relationship in the revised paper.
>
> **(Q5): Speaking SalUn method...**
>
> (A5): SalUn restricts updates to a subset of model parameters so it can reduce the space of minimizers to its unlearning objective. However, we emphasize that the multiple minimizers of the *training loss* under overparameterization was what motivated our definition of unlearning Section 2.1, not the minimizers of the *unlearning objective*.
>
> **Concluding Remark:**
>
> We hope that the clarifications and additional results above adequately address your concerns. If so, we kindly request you to consider revising your score. We are committed to address any additional points you may have during the discussion phase.
>
> As a side note, we also noticed that the originality score mentioned in the text of your review was a 4, while the official score was recorded as a 3. If the higher score was your intended assessment, we would appreciate your consideration in updating it.
>
> [1] Feng, Z., et al. "Existing Large Language Model Unlearning Evaluations Are Inconclusive."
>
> [2] Hu, S., et al. "BLUR: A Benchmark for LLM Unlearning Robust to Forget-Retain Overlap."
>
> [3] Shi, W., et al. ``MUSE: Machine Unlearning Six-Way Evaluation for Language Models."
>
> [4] Maini, P., et al. ``TOFU: A Task of Fictitious Unlearning for LLMs."

---

> > ### Comment · Reviewer_Tqz5 · 2025-08-04
> >
> > Thank you for your detailed, meaningful explanations.
> >
> > **1. You write: "While our experiments focus scales up to ResNet-18 on CIFAR-10, ...".**
> >
> > I understand the focus on theoretical/foundational arguments and I appreciate your experimental efforts, as I had said explicitly. But scalling to ResNet18/CIFAR-10 is simply not enough. You do not have to try LLMs - you could try ResNet50 or 100 on TinyImageNet-1k, for example.
> > I am aware that this is a tall ask! But an important one.
> > Failing to do this: Can you explain what you would expect to see when moving up the ladder towards real-world models/tasks? Can you argue about fundamental issues that would render the results useful/useless in those settings?
> > Are the reasons simply the high compute- and time-costs?
> > How would you defend that the fundamentals discovered via this work are relevant?
> >
> > **2. Interpolated data**
> >
> > I would strongly discuss more on interpolation. What if the dataset is very noisy (eg scraped from the web)? And perhaps augment your related discussions.
> >
> > Also, please explain what happens to the generalization of the model - accuracy on test sets?
> >
> >
> > **3. Forget loss**
> >
> > **3.a.** Please clarify this issue: In most unlearning literature minimizing the forget loss is **not** the goal - instead, the goal is to have a comparable forget loss to the Retrain model, no?
> >
> > **3.b.** In most real-world cases, some examples in the forget set may be interpolated and some not. Is this not the case? If so, please discuss this in detail.

---

> > > ### Author Response · Authors · 2025-08-04
> > >
> > > **Interpolated Data**
> > >
> > > Thank you for bringing up these points. We consider overparameterized training, where the original model exactly fits each sample. If the data corresponds to some signal plus noise, then overparameterized training will interpolate the noisy observations, and unlearning corresponds to finding the model which we would have gotten if we had trained on $D_r$ alone. This ground truth unlearned model will also interpolate $D_r$, but has no knowledge of $D_f$. Since there are many candidate models which interpolate $D_r$, including the original model which interpolates both $D_r$ and $D_f$, our framework searches for the least complex interpolator of $D_r$ (Eqn. 4), which we define as the minimizer to a complexity measure $R$ subject to the constraint of interpolating $D_r$.
> > >
> > > We note that solving Eqn. (4) for many choices of $R$, which may have nothing to with model complexity, can guarantee that the returned model has no dependence on $D_f$. For example, $R$ could promote the parameter vector $\theta$ to align with some fixed direction: $R(\theta) = -(v^\top \theta)^2$ for some $v \in \mathbb{R}^d$. However, this may not lead to a model that performs well given the information in $D_r$. This leads to your question about generalization. We design $R$ to measure model complexity so that our framework returns a model which achieves strong generalization performance given only $D_r$. This is because in overparameterized settings, model simplicity is closely tied to generalization. Although our framework makes no assumptions about the data distribution, it aims to remove all information unique to $D_f$ while preserving the best generalization possible from $D_r$ alone. Since unlearning aims to simulate retraining from scratch on $D_r$, if $D_r$ is noisy, then strong generalization may be impossible. However, our framework uses model simplicity to extract the most reliable signal available, aiming for the best performance achievable from $D_r$ alone.
> > >
> > > **Forget Loss**
> > >
> > > (a) Thank you for the opportunity to clarify. Our framework does **not** minimize training loss on the forget set, nor do we compute this in evaluation. Our framework aims to minimize model complexity subject to interpolation of $D_r$ and does not consider $D_f$. In terms of our evaluation metrics, in the Multi-Class Label Erasure experiment, we report two metrics: (i) retain quality (accuracy on gray test images), and (ii) forget quality (squared error between the model's prediction that images of all colors are gray and the ideal value of 1.0 achieved by the true unlearned model trained only on $D_r$). Thus, forget quality reflects divergence from this behavior of certainty of gray predictions, not forget set loss. In the results table in our rebuttal we use “Retain Accuracy” and “Forget Error” to refer to these metrics to emphasize that the retain quality metric should be maximized while the forget quality metric should be minimized.
> > >
> > > Comparing training loss on the forget and retain sets can be a valid metric when $D_r$ and $D_f$ are independent and identically distributed. However, our experiments introduce structural differences between $D_r$ and $D_f$ to enable fine-grained, interpretable evaluation. In such non-i.i.d. settings, this loss comparison may be misleading, as even the true unlearned model trained from scratch on $D_r$ will naturally exhibit different losses on the two sets.
> > >
> > > (b) We appreciate the reviewer highlighting that the ground truth unlearned model trained on $D_r$ may still perform well on some $D_f$ examples. This is a core motivation behind our framework: rather than forcing poor performance on $D_f$ (as done by GA, NGP, Scrub, NPO, SalUn), we search for the minimum complexity interpolator over $D_r$. This ensures the model generalizes as well as possible using only the information in $D_r$. If $D_f$ shares structure with $D_r$, the model may still perform well on it. If $D_f$ contains unique information, it will not. This aligns with the intended behavior of unlearning, unlike methods that penalize $D_f$ regardless of its relation to $D_r$.
> > >
> > > To illustrate this, consider when $D_f$ is very similar to $D_r$. In the linear model setting, suppose the inputs in $D_f$ lie close to the subspace $S_r$ spanned by the inputs from $D_r$. When the complexity measure $R$ is the $\ell_2$ norm, the true unlearned model is found by projecting the original parameter $\theta^\ast$ onto $S_r$. The error of this true unlearned model in predicting the examples in $D_f$ depends only on the distance of each $D_f$ input from $S_r$. In the edge case where the inputs from $D_f$ lie on $S_r$, the true unlearned model then interpolates $D_f$ exactly. This highlights that our framework recovers the best generalization possible from $D_r$ alone—if $D_f$ is redundant with $D_r$, we provably maintain strong performance in this example setting.

---

> ### Author Response · Authors · 2025-08-04
>
> Thank you for your additional comments. We include our responses below in two sections due to the space constraints. We are happy to respond to any additional concerns or further questions you may have during the discussion period.
>
> **Experiments**
>
> Thank you for recognizing our theoretical and experimental contributions. On larger models and datasets, we expect similar trends, with some caveats. Larger models are more prone to memorization, so retaining performance on $D_r$ likely requires access to more—if not all—of the retain set. For diverse datasets like TinyImageNet-1k, small subsets may not capture the full data distribution, making it harder for any method to preserve retain set accuracy while effectively removing influence of the forget set. While unlearning for a small number of epochs using all of $D_r$ may be more efficient than retraining since the retraining cost is also larger at scale, we still aim to study unlearning in settings where the full retain set is not accessible. This may lead to a more difficult evaluation setting which further stresses the impact of limited sample access relative to the complexity of the learning task.
>
> However, besides the need for more samples or unlearning epochs at scale, the fundamental limitations of existing unlearning methods remain. Our theory highlights a fundamental issue: vanishing gradients make descent-based methods like GD, NGD, and Ridge less effective, as the loss gradient over $D_r$ vanishes at the original model (Theorem 1). This hinders their ability to meaningfully perturb the model and slows unlearning. We observe this consistently across experiments: in Figure 1 (Data Poisoning), these methods stay close to the original model; in Figure 2 (Multi-Class Label Erasure), their Pareto frontiers fail to move away from the upper-right corner which represents the original model performance; and in Table 2 (Representation Collapse), they struggle to recover the correct decision rule based on image color rather than image content. These limitations stem from their formulations and remain regardless of model or dataset scale.
>
> Secondly, methods that enforce specific behavior on the forget set, such as GA, NGP, Scrub, and NPO which aim to maximize training loss, or SalUn and KL-Unif which maximize uncertainty, do not accurately reflect the behavior of the true unlearned model retrained from scratch on $D_r$ alone. The performance of the true unlearned model on $D_f$ depends on how similar $D_r$ is to $D_f$. If they are similar, the true unlearned model likely performs well on $D_f$, and if not, it likely achieves poor performance. Methods that rigidly enforce poor performance or high uncertainty fail to capture this nuance. As a result, they perform poorly in our experiments. For example, GA returns a model which completely diverges in the Data Poisoning experiment. Also, in the Multi-Class Label Erasure experiment these methods struggle to achieve low forget quality error, as this requires predicting all colored images to be gray with high confidence, reflecting the behavior of the true unlearned model trained only on gray images. Their objective functions only push the model to avoid predicting the actual color of these forget set images, which does not directly lead to forgetting the non-gray labels, since simply performing badly on $D_f$ does not always correspond to unlearning.
>
> In contrast, our framework which searches for the simplest model that interpolates the retain set (i) does not rely on loss descent over $D_r$ to perturb the original model, and (ii) does not prescribe specific behavior on $D_f$. Instead, it uses the fact that the simplest model explaining $D_r$ will generalize to $D_f$ only to the extent that $D_f$ is inferable from $D_r$ alone. This more closely reflects the behavior of the true unlearned model, which is the best performing model trained solely on $D_r$. Our method is the only one that captures this principle. We formalize this difference in our theory and show that it leads to consistent performance improvements across experiments.
>
> While we cannot fully explore larger-scale results within the remaining 2 days of the discussion period, we believe the theoretical and empirical differences between our framework and existing methods strongly support the value of our contributions. Verifying these effects at larger scale remains an important direction, but we believe our work identifies the root causes of failure in existing methods and develops a principled framework and algorithm for addressing them.

---

> > ### Comment · Reviewer_Tqz5 · 2025-08-05
> >
> > Thank you for the additional explanations.
> >
> > I fail to see the practicality of this setting, especially find it very hard to believe that all training examples will be interpolated - to the best of my knowledge, this is never the case in real-life scenarios.
> > So, unless you can convince me that this belief is incorrect, I will keep my score.

---

> ### Author Response · Authors · 2025-08-05
>
> Thank you for your continued engagement and for clarifying your concern that exact interpolation rarely holds in practice. We fully agree with this claim and appreciate the opportunity to clarify how **our framework remains applicable and effective even in this more realistic setting.**
>
> In our framework we consider the overparameterized setting, meaning the model class has the capacity to interpolate the dataset, and for our analysis, we make the common theoretical assumption that the original model is an exact minimizer of the training loss (Eqn. 1), meaning it is an exact interpolator. **This is a technical assumption for the theoretical overparameterized setting.** It allows us to cleanly analyze unlearning for different model classes in Section 4, where we define the exact unlearning solution in Eqn. (4) as the specific interpolator of the retain set which minimizes the model complexity measure $R$, and aim to recover this exact solution using the relaxed, surrogate problem in Eqn. (7). We use the assumption that the original model $\theta^\ast$ interpolates the retain set in order to replace the general interpolation constraint with the local linear approximation in Eqn. (7), which enforces that the unlearned model's predictions on the retain set remain unchanged relative to the original model up to first order error. Thus, **when the original model only interpolates the retain set up to $\epsilon$ error, our framework searches for the least complex model which interpolates the retain set up to the same $\epsilon$ error**, since we effectively enforce that the returned model achieves the same predictions as the original model over the retain set. In these cases where the original model is not an exact interpolator, our framework aims to simply preserve the original model performance over the retain set.
>
> Regarding the performance of the existing baselines which perturb the initial model according to loss gradients, we show in Theorem 1 that when the original model exactly interpolates the training data, these methods fail to provide a meaningful update since the loss gradients vanish to 0 over all samples. In the practical case where the initial model does not exactly fit each sample, these methods can still struggle, as for the majority of samples which are accurately predicted by the model we still observe this vanishing gradient behavior.
>
> Importantly, our approach does not rely on perfect interpolation to remain effective. We recognize that in practice we may only achieve approximate interpolation, especially when the model is not largely overparameterized relative to a diverse, potentially noisy dataset. We emphasize that in our Multi-Class Label Erasure experiment using ResNet18 on CIFAR-10 (Figure 2, right), the original model does not exactly interpolate the training data. Even in this setting, we still see significant benefits to our practical method MinNorm-OG relative to the existing baselines. For example in a sample trial we see that the norm of the loss gradient evaluated at the initial model is 8452.23 for a parameter vector which lies in a 11.1-million-dimensional space. Thus, while the original model does not achieve exact interpolation and the loss gradients are not exactly 0 at the initial model, we still see that our framework provides practically relevant insights and leads to improved performance.
>
> While we agree it is a natural next step to test using larger, more diverse datasets, such as TinyImageNet, we anticipate that this behavior will largely remain the same, since our existing experiments capture the realistic case you point out where we fail to achieve exact interpolation. In summary, we agree with your point that exact interpolation is rarely satisfied in practice, and we hope this response clarifies that our framework remains meaningful in this more realistic regime.

---

> > ### Author Response · Authors · 2025-08-08
> >
> > Dear Reviewer tqz5,
> >
> > We wanted to follow up on our previous response given some preliminary additional experimental results. As we mention above, in realistic overparameterized settings we do not expect the model to *exactly* interpolate the training data, but instead achieve approximate interpolation up to some small error. We see this in our CIFAR-10 experiment, where the initial model ResNet-18 model achieves 99-100% accuracy at predicting the training set images, but does not exactly interpolate. Further, while we are unable to fully complete larger scale experiments in the short discussion period, we have obtained preliminary results in the setting you suggested training ResNet50 on TinyImageNet. We similarly observe that the original model predicts the training images with almost 100% accuracy.
> >
> > Lastly, we note that in many realistic scenarios, especially for LLMs, models have been observed to memorize training samples, demonstrating that they nearly interpolate specific pieces of text from their training data. For example, LLMs can output exact training text verbatim [1] and can be prompted to output specific samples or knowledge [2-3]. Further, larger models have been observed to memorize samples to a larger degree [4]. Thus, we believe that models do demonstrate memorization and near-interpolation at realistic large-scales, which is the setting we aim to capture in our framework.
> >
> > [1] "Extracting training data from large language models." (Carlini et al.)
> >
> > [2] "Muse: Machine unlearning six-way evaluation for language models." (Shi, et al.)
> >
> > [3] "Are large pre-trained language models leaking your personal information?." (Huang et al.)
> >
> > [4]  "Quantifying memorization across neural language models."  (Carlini, et al.)

---

### Official Review · Reviewer_nR8A · 2025-07-04

**Clarity:** 3
**Significance:** 3
**Originality:** 4
**Rating:** 5
**Confidence:** 4

**Summary:**

This paper addresses the machine unlearning problem, with a particular emphasis on overparameterized models. The authors formulate exact unlearning in this setting as a bilevel optimization problem, which not only minimizes loss optimality but also incorporates an additional complexity objective. Building on this novel definition, they propose a solution framework supported by rigorous theoretical analysis. Additionally, they develop an iterative algorithm, MinNorm-OG, and demonstrate its effectiveness through three comprehensive experiments with interpretable quantitative metrics. Overall, this is a solid work that effectively fills a gap in machine unlearning research.

**Questions:**

Q1: In Line 47, the authors state that their analysis “ … focus on settings where the loss is minimized by any interpolating model”.

How does the theoretical framework account for cases where the model converges to a local minimum while still satisfying the interpolation constraint?

Q2: Following Q1, what are the theoretical and empirical consequences if the overparameterized model only reaches an approximate optimum but fails to achieve exact interpolation (e.g., due to optimization limitations or early stopping)? Since real-world training may not guarantee convergence to a perfect interpolating solution, how does this affect the proposed unlearning framework’s validity?

Q3: Figure 1 shows an unusual trend: both the original model (red curve) and most baseline unlearning methods (except GA) exhibit a sharp initial peak in their fitting curves. What explains this consistent phenomenon across multiple methods? Why does MinNorm-OG uniquely avoid this issue? Could the authors provide empirical observations or theoretical insights to clarify the mechanism behind this improvement?

**Ethical Concerns:**

["NO or VERY MINOR ethics concerns only"]

**Limitations:**

A deeper discussion on the assumption regarding the interpolating models would strengthen the work.

**Quality:**

4

**Strengths And Weaknesses:**

Clarity
- S1. The paper is well-organized and clearly written, making it accessible to readers.

Significance
- S2. While machine unlearning has been extensively studied, this work tackles a novel and previously unexplored setting, overparameterized models. This unique focus makes it a valuable contribution to the machine unlearning community.

Originality
- S3. The paper offers a fresh perspective by distinguishing between underparameterized and overparameterized regimes, thereby justifying the need for a new definition of unlearning tailored to the latter case.

Quality
- S4. The theoretical foundations are rigorous, and the experimental results in both the main manuscript and the appendices provide sufficient support for the paper’s claims.
- W1. The major concern is the assumption regarding the interpolating models, whose practical applicability remains unclear. A deeper discussion on this aspect (see Q1-Q2) would strengthen the work.

---

> ### Author Rebuttal · Authors · 2025-07-31
>
> Dear Reviewer nR8A, thank you for highlighting the novelty of our analysis of unlearning within the overparameterized setting along with the rigor of our results. We address your questions and concerns below.
>
> **(Q1): In Line 47, the authors state that their analysis ``focus on settings where the loss is minimized by any interpolating model.” How does the theoretical framework account for cases where the model converges to a local minimum while still satisfying the interpolation constraint?**
>
> (A1): Thank you for raising this question about our theoretical framework and the relationship between data interpolation and loss minimization. For our analysis, we assume that the loss function has the property that any model that interpolates the data must be a global minimizer of the loss. For example, this is the case in $\ell_p$-norm regression or 0-1 classification (Line 103). In these settings, it is not possible to recover an interpolating model which is not also a global loss minimizer. Because we work in the overparameterized setting, we know such interpolating models exist. Thus, using these assumptions we can rewrite Eqn. (2), which defines the ground truth unlearned model as the least complex loss minimizer, into Eqn. (4), which expresses the ground truth unlearned model as the least complex data interpolator. Our assumption about the loss structure allows this conversion without loss of generality.
>
> However, as you mention, there are cases where interpolation and loss minimization are not strictly equivalent. This notably arises in classification tasks, where the true goal is to minimize the expected 0-1 loss over the data, but since this is known to be NP-hard, we instead look to minimize a tractable surrogate like the cross-entropy loss or hinge loss. Even in these settings, our framework of aiming to solve Eqn. (4) through the relaxation in Eqn. (7) remains meaningful, as it **aims to preserve optimality with respect to the true target loss, rather than the training surrogate**. For example, in the classification setting using a surrogate loss like the ones mentioned above, Eqn. (4) searches for the least complex model which correctly predicts the label for each sample in the retain set, therefore minimizing the 0-1 loss over these samples, rather than searching for a model which minimizes the training surrogate. Although 0-1 loss is difficult to directly minimize, we use the fact that the original overparameterized model interpolates the training data and thus a minimizes the 0-1 loss on the retain set. To ensure the unlearned model preserves this property, we only require that the unlearned model's label predictions over the retain set match those of the original model. This condition is tractably approximated using the constraint relaxation in Eqn. (7) which enforces that the model predictions on the retain set remain unchanged up to first order error.
>
> Therefore, in these cases where surrogate losses are used to approximate a more difficult target loss like 0-1, our framework effectively aims to preserve optimality with respect to the true target loss on the retain set, rather than the surrogate, throughout unlearning. We also empirically validate this in our Multi-Class Label Erasure and Representation Collapse experiments, where we use cross-entropy loss as a surrogate for 0-1, and our method continues to perform strongly.
>
> We appreciate the reviewer’s thoughtful observation and will revise the paper to clarify that even when interpolation does not imply global loss minimization, our framework seeks to preserve optimality with respect to the true target loss, not the training surrogate.
>
> **(Q2): What are the theoretical and empirical consequences if the overparameterized model only reaches an approximate optimum but fails to achieve exact interpolation (e.g., due to optimization limitations or early stopping)? Since real-world training may not guarantee convergence to a perfect interpolating solution, how does this affect the proposed unlearning framework’s validity?**
>
> (A2): This question raises an important point about the validity of our unlearning framework in practical settings. For our analysis, we make the common theoretical assumption that the original model is an exact minimizer of the training loss (Eqn. 1). However, as you point out, in practice we can only hope to achieve approximate minimization of the training objective. In this case, we cannot assume in Eqn. (4) that the original model exactly interpolates the retain set, but instead achieves approximate interpolation up to some error $\epsilon$. In this practical setting, our framework of aiming to solve Eqn. (4) through the relaxation in Eqn. (7) then aims to find the least complex model which **preserves the same error $\epsilon$ on the retain set**.
>
> To illustrate this point, note that a key characteristic of our framework is to exploit the fact that the original model interpolates the retain set and is thus feasible for the problem in Eqn. (4). This allows us to replace the general interpolation constraint with the local linear approximation in Eqn. (7) which enforces that the unlearned model's predictions on the retain set remain unchanged relative to the original model up to first order error. Thus, when the original model only interpolates the retain set up to $\epsilon$ error, our framework searches for the least complex model which interpolates the retain set up to the same $\epsilon$ error. Since we assume the original model is overparameterized and trained to completion, this $\epsilon$ should be small in practice so that our framework leads to a strong (approximate) candidate unlearning solution.
>
> We note that this is the case in our Data Poisoning experiment. The initial trained model, shown in light red in Figure 1, does not perfectly fit each of the retain set or forget set samples. However, our framework still achieves strong performance.
>
> **(Q3): Figure 1 shows an unusual trend: both the original model (red curve) and most baseline unlearning methods (except GA) exhibit a sharp initial peak in their fitting curves. What explains this consistent phenomenon across multiple methods? Why does MinNorm-OG uniquely avoid this issue? Could the authors provide empirical observations or theoretical insights to clarify the mechanism behind this improvement?**
>
> Thank you for mentioning this observation, as it sheds light on key characteristics of overparameterized learning and the failure of the other baseline unlearning methods within this setting. The sharp peak observed in the original model just before the first forget point (ordered left to right) arises because the original model is first trained to fit the entire dataset before unlearning. There is little separation in input space between this forget point and the preceding retain points, but a large difference in their corresponding output values. To interpolate the dataset, the model must sharply transition its predictions in this region, resulting in the observed peak.
>
> The reason the other unlearning methods (besides GA) fail to remove this peak directly relates to the issue of vanishing loss gradients discussed in Section 2.2 (Theorem 1). Descent-based methods such as GD, NGD, and Ridge attempt to unlearn primarily by continuing to minimize the loss on the retain set. However, at the original model, the loss gradients are nearly zero, making it extremely slow, or even impossible, for these methods to remove unnecessary artifacts embedded in the initial model. In contrast, GA, which only ascends the loss on the forget set, successfully removes the peak, but entirely ignores the retain set, as the resulting model diverges significantly from the retained data. NGP, which optimizes a weighted combination of GA and GD objectives, also fails to remove the artifact in Figure 1. This is because Figure 1 shows results which correspond to the best hyperparameters we found for each method in terms of the unlearning error metric. For NGP, we found that the best performance occurred when the GA term was given a very small weight (Appendix F.2, Table 5), since increasing this weight caused the model to diverge and behave more like GA. Therefore, in this setting, NGP behaves similarly to GD and inherits its failure to produce a meaningful unlearning update.
>
> Our method avoids this issue because MinNorm-OG neither relies on loss gradients nor incorporates any form of loss ascent. Instead, it focuses on minimizing model complexity while approximately preserving optimality on the retain set. As a result, it is not affected by the vanishing gradient problem and is able to effectively remove unnecessary peaks and artifacts in the original model that are irrelevant to retain set interpolation.
>
> We again thank the reviewer for this insightful observation, as it highlights a central claim of our paper: that loss-gradient-based unlearning methods are fundamentally ill-suited to the overparameterized regime, whereas minimizing model complexity subject to interpolation constraints offers a tractable and effective alternative.
>
> **Concluding Remark:**
>
> We hope that the clarifications above adequately address your concerns, and we are committed to address any additional questions you may have during the discussion phase.

---

> > ### Comment · Reviewer_nR8A · 2025-08-04
> >
> > Dear authors,
> >
> > Thanks for your explanations and they’ve addressed most of my questions. But I still have one point of confusion regarding Q3:
> >
> > The authors have clarified why most unlearning methods (except GA) fail to eliminate the sharp peak between the first and second forget points, i.e., attributing it to the issue of vanishing loss gradients. That makes sense. However, I’m curious why this same issue doesn’t seem to affect the other forget points. In Figure 1 (b), (d), and (e), there’s a noticeable trend: the blue curve fits the second and third peaks of the sin(x) function more accurately than the first peak. What accounts for this discrepancy?

---

> > > ### Author Response · Authors · 2025-08-04
> > >
> > > Thank you for your question and for highlighting this interesting pattern. One reason the first peak may be harder to remove is that the poisoned target label of the first forget set point is far from the true sine curve value. The model has thus learned a much larger value at this point, embedding a strong “peak” within the model weights that is more difficult to erase. In contrast, the second and third peaks correspond to forget points where the poisoned labels are closer to the true sine values, making them potentially easier to unlearn. While formally analyzing the dynamics of each unlearning method is an interesting extension, we hope this offers some intuition for the observed behavior, and we are happy to answer any follow-up questions you may have.

---

> > > > ### Comment · Reviewer_nR8A · 2025-08-05
> > > >
> > > > Dear authors, thank you for your answers. That's a very interesting finding. I have no further questions.

---

### Comment · Area_Chair_cHxz · 2025-08-02

Dear all,

Thanks for your engagement in the review process.

***Please read all other reviews and the author responses carefully, and provide your response as soon as possible***. The Author-Reviewer Discussions (July 31 - Aug 6) are crucial for ensuring review quality and providing authors with an opportunity to address questions and potential misunderstandings from reviewers. Please treat it seriously.

Again, thanks for your work so far, and ***please do participate in the Author-Reviewer Discussions ASAP***.

Best,

AC

---

### Decision · Program_Chairs · 2025-09-17

**Decision:**

Accept (poster)

**Comment:**

The work addresses an important problem, i.e., how to define and perform machine unlearning in overparameterized regimes where models interpolate the training data. The authors propose a new definition of the unlearning solution as the minimum-complexity interpolator on the retain set and introduce a corresponding algorithm, MinNorm-OG, which shows consistent empirical improvements over several baselines across thoughtfully designed benchmarks.

The paper makes a valuable contribution by providing a well-justified theoretical foundation for unlearning in this setting. The connection between theory and practice is clear, and the experiments are interpretable and relevant. Meanwhile, some limitations are also noted, particularly regarding the scalability of the method to very large models and the strong assumption of exact interpolation. The authors are encouraged to consider these aspects in future work, including more extensive empirical validation on larger-scale datasets and architectures. Additionally, further analysis of the relationship between their proposed metrics and existing unlearning evaluation frameworks would strengthen the applicability of their approach.

Overall, the paper introduces a novel and theoretically sound perspective that effectively addresses a clear gap in the literature, making it a good fit for acceptance.